# A bistable inhibitory optoGPCR for multiplexed optogenetic control of neural circuits

Jonas Wietek [1,2,19] ✉, Adrianna Nozownik[3,20], Mauro Pulin [3,21], Inbar Saraf-Sinik[1,2], Noa Matosevich[4], Raajaram Gowrishankar[5,6], Asaf Gat [1,2], Daniela Malan[7], Bobbie J. Brown[8], Julien Dine[1,2,22], Bibi Nusreen Imambocus[9], Rivka Levy[1,2], Kathrin Sauter[3], Anna Litvin [1,2], Noa Regev[10], Suraj Subramaniam[1,2], Khalid Abrera[5], Dustin Summarli [5], Eva Madeline Goren[4,23], Gili Mizrachi[1,2], Eyal Bitton[1,2], Asaf Benjamin [1,2], Bryan A. Copits [8], Philipp Sasse[7], Benjamin R. Rost [11,12], Dietmar Schmitz[11,12,13,14,15], Michael R. Bruchas [5,6,16], Peter Soba [9,17], Meital Oren-Suissa [1,2], Yuval Nir [4,10,18], J. Simon Wiegert [3,24] & Ofer Yizhar [1,2] ✉

Information is transmitted between brain regions through the release of neurotransmitters from long-range projecting axons. Understanding how the activity of such long-range connections contributes to behavior requires efficient methods for reversibly manipulating their function. Chemogenetic and optogenetic tools, acting through endogenous G-protein-coupled receptor pathways, can be used to modulate synaptic transmission, but existing tools are limited in sensitivity, spatiotemporal precision or spectral multiplexing capabilities. Here we systematically evaluated multiple bistable opsins for optogenetic applications and found that the *Platynereis dumerilii* ciliary opsin (*Pd*CO) is an efficient, versatile, light-activated bistable G-protein-coupled receptor that can suppress synaptic transmission in mammalian neurons with high temporal precision in vivo. *Pd*CO has useful biophysical properties that enable spectral multiplexing with other optogenetic actuators and reporters. We demonstrate that *Pd*CO can be used to conduct reversible loss-of-function experiments in long-range projections of behaving animals, thereby enabling detailed synapse-specific functional circuit mapping.

Even the simplest behaviors are coordinated by neural ensembles spanning multiple brain regions. Understanding the roles of long-range connections linking these ensembles requires techniques that allow selective manipulation of their function. While optogenetic tools allow the manipulation of neural firing with high temporal and spatial precision[1], projection neurons often target several downstream regions via branched axonal collaterals[2,3]. Thus, manipulating the neuronal soma may result in a partial or even misleading picture of their contribution to circuit function. Instead, directly targeting the synaptic terminals of long-range connections can provide fine-grained insight into the role of specific neuronal pathways. However, direct suppression of synaptic terminal function poses challenges. Inhibitory optogenetic tools, such as the microbial light-driven ion pumps, have traditionally been used to silence synaptic transmission at axonal terminals[4–8]. Their inhibitory effect on synaptic release, however, is not only partial and short-lived, but it also can induce unintended paradoxical

effects, such as an increase of spontaneous neurotransmitter release[9]. Chloride-conducting channelrhodopsins also proved unsuitable for synaptic silencing, as they depolarize the axon and can trigger antidromic firing due to the high chloride reversal potential in this subcellular compartment[9–11]. In contrast, targeting inhibitory G-protein-coupled receptor (GPCR) pathways with bistable rhodopsins is effective for attenuating synaptic release in a projection-specific manner[12–14].

Exogenously expressed light-activated animal rhodopsins can transiently inhibit synaptic transmission, by coupling to endogenous inhibitory G proteins. While visual rhodopsins can be expressed in neurons and used to suppress synaptic release[12], these photoreceptors can undergo bleaching (that is, they lose their light-sensitive chromophore retinal), which reduces their efficacy under sustained illumination[15]. In contrast, bleaching-resistant nonvisual rhodopsins have gained attention as light-activated tools for suppression of presynaptic release[16]. Like endogenous inhibitory GPCRs, these light-activated GPCRs (optoGPCRs) trigger the opening of G-protein-coupled inwardly rectifying potassium (GIRK) channels, activate the $G\alpha_{i/o}$ signaling pathway (Fig. 1a) and efficiently suppress synaptic transmission through the inhibition of voltage-gated calcium channels (VGCCs) and the SNARE (soluble N-ethylmaleimide-sensitive-factor attachment receptor)-mediated fusion of synaptic vesicles with the presynaptic membrane[17,18]. Activated $G\alpha_i$ subunits can also reduce cAMP production by adenylate cyclases (ACs), indirectly decreasing cAMP-dependent neurotransmission[19–21].

Although progress has been made in developing inhibitory optoGPCRs, the existing tools are limited in either their spectral or their temporal features. Two bistable rhodopsins, the trafficking-enhanced eOPN3 derived from *Anopheles stephensi*[13] (referred to herewith as *As*OPN3; for further species and protein abbreviations, see Extended Data Fig. 1) and the parapinopsin[14] from the Japanese lamprey *Lethenteron camtschaticum* (*Lc*PPO), have been utilized as inhibitory optoGPCRs for presynaptic inhibition. The highly light-sensitive *As*OPN3 has a broad action spectrum that spans the entire ultraviolet (UV)-visible range[13]. However, *As*OPN3 activity cannot be rapidly reverted to the inactive (dark-adapted) state and takes minutes to spontaneously recover to its non-signaling state[13,22]. *Lc*PPO can undergo photoswitching between its active and inactive states by different wavelengths, thus allowing better temporal control[23,24]. However, *Lc*PPO's limitation lies in its UV maximum activation wavelength (~370 nm) and its broad inactivation spectrum[23,24]. These spectral properties restrict the wavelength range available for multiplexed applications with additional optogenetic actuators or fluorescence-based sensors. Especially for single-photon fiber photometry and miniature microscopy techniques, spectral multiplexing can be challenging with the current tools.

To improve and expand the capabilities of inhibitory optoGPCRs, we aimed for a tool that retains the advantages of *As*OPN3 and *Lc*PPO but overcomes their limitations. We systematically screened a range of bistable opsins and evaluated their potential use as optoGPCRs based on their cellular biodistribution, spectral features and kinetic properties. Our screen revealed that the ciliary opsin 1 from *Platynereis dumerilii* (*Pd*CO)[25,26] is a highly light-sensitive, bidirectionally switchable, versatile inhibitory optoGPCR. *Pd*CO expresses well in mammalian neurons and allows robust, high-efficiency and rapidly switchable presynaptic silencing across various cell types and preparations. With its red-shifted activation wavelength and a narrow inactivation spectrum, *Pd*CO is well suited for multiplexing with other optogenetic actuators and genetically encoded sensors.

## Results

### optoGPCR literature mining and functional benchmarking

We conducted a comprehensive literature search and identified a list of suitable optoGPCR candidates that could enable light-controlled inhibition of synaptic transmission. We collected information on retinal binding, spectral properties, switchability, G-protein coupling specificity and activation of GIRK channels (Extended Data Fig. 1). Of the 32 described rhodopsins we selected for analysis, we identified 11 switchable variants that were promising due to their coupling to $G\alpha_{i/o}$ or activation of GIRK channels (Fig. 1b). We designed optoGPCR constructs as previously reported[13] with a C-terminal rhodopsin 1d4 epitope tag, a Golgi trafficking signal (TS) and an endoplasmic reticulum (ER) export signal to enhance membrane localization[10] (Fig. 1c). Including the *As*OPN3 and *Lc*PPO for comparison, we conducted a three-part benchmark to characterize the functional properties of these optoGPCRs (Fig. 1d–i and Extended Data Figs. 2 and 3). Using the chimeric Gα bioluminescence assay (GsX[27]; Fig. 1d,e), we found that *Ol*TMT1A, *Lc*PPO, the PPOs from pufferfish (*Tr*PPO1/*Tr*PPO2), *Pd*CO, *As*OPN3 and *Dr*PPO1 couple to the inhibitory $G_{i/o}$-protein family ($G_{i/o/t/z}$; Fig. 1f). With the exception of *Tr*PPO2, which displayed additional coupling to $G_{q/15}$ and $G_{12/13}$, and *Bb*OPN, which showed nonselective G-protein coupling, we could not detect any G-protein activation other than for the $G_{i/o}$ pathway (Extended Data Fig. 2).

Next, we recorded Gβγ-activated GIRK currents in the human embryonic kidney (HEK) 293 cell line (Fig. 1g,h). *Lc*PPO, *Pd*CO, *Tr*PPO1 and *As*OPN3 produced the largest GIRK current amplitudes (>700 pA), while the other variants induced currents smaller than 270 pA (Fig. 1i). *Bb*OPN did not produce detectable light-induced currents, and *Gg*OPN5l1 displayed a small inhibition of GIRK currents, which is consistent with its reported dark activity[28]. We also measured optoGPCR-evoked GIRK activation in cultured neurons, where all optoGPCR except *Bb*OPN and the OPN5 variants showed light-evoked GIRK conductance (Extended Data Fig. 3).

In line with these electrophysiological recordings, *Lc*PPO, *Pd*CO, *Tr*PPO1 and *As*OPN3 showed strong expression and membrane targeting (Extended Data Fig. 3). Based on this initial screen, we selected the seven best-performing variants and next tested their ability to attenuate synaptic transmission.

### Benchmarking of bistable optoGPCRs in autaptic neurons

We expressed each of the selected optoGPCRs via recombinant adeno-associated virus serotype 2/1 (rAAV2/1) transduction in autaptic neurons (Fig. 2a) and measured paired-pulse-induced excitatory postsynaptic currents (EPSCs) over three sequential periods: in the dark, after light application (one 500-ms pulse, 390 nm) and after administration of a 4.5-s-long 560-nm light pulse to inactivate the optoGPCR (Fig. 2b and Extended Data Fig. 4). We applied the same protocol to non-expressing controls, to correct for spontaneous EPSC rundown over time (Fig. 2c,d and Extended Data Fig. 4). Light activation of the PPOs from pufferfish (*Tr*PPO1/*Tr*PPO2) and zebrafish (*Dr*PPO1) had no effect on synaptic transmission, while *Ol*TMT1A, *Pd*CO, *Lc*PPO and *As*OPN3 substantially attenuated the action potential (AP)-evoked EPSCs (Fig. 2e and Extended Data Fig. 4). *Ol*TMT1A and *Lc*PPO could only attenuate transmission by 66% ± 5% and 61% ± 5% (mean ± s.d.), respectively. Activation of *Pd*CO and *As*OPN3 yielded the strongest EPSC reduction, by 89% ± 3% and 84% ± 5%, respectively (mean ± s.d.; Fig. 2e). As reported[13], *As*OPN3-mediated inhibition was long-lasting and could not be recovered by light application at different wavelengths. Green light-induced EPSC recovery was only partially possible for *Ol*TMT1A, consistent with reported spectra[29]. However, for *Pd*CO and *Lc*PPO, the green light pulse reliably induced recovery of synaptic transmission (Fig. 2e). Expression of the different optoGPCRs did not alter the intrinsic properties of expressing neurons (membrane resistance and cell capacitance) when compared to non-expressing control cells (Extended Data Fig. 5).

In line with presynaptic inhibition, the frequency but not the amplitude of miniature EPSCs (mEPSCs) was reduced for the four optoGPCRs that showed light-induced EPSC attenuation, while paired-pulse ratios were increased (Extended Data Fig. 5). Baseline paired-pulse ratios were not affected by optoGPCR expression (Extended Data Fig. 5), further indicating the absence of dark activity of these optoGPCRs.

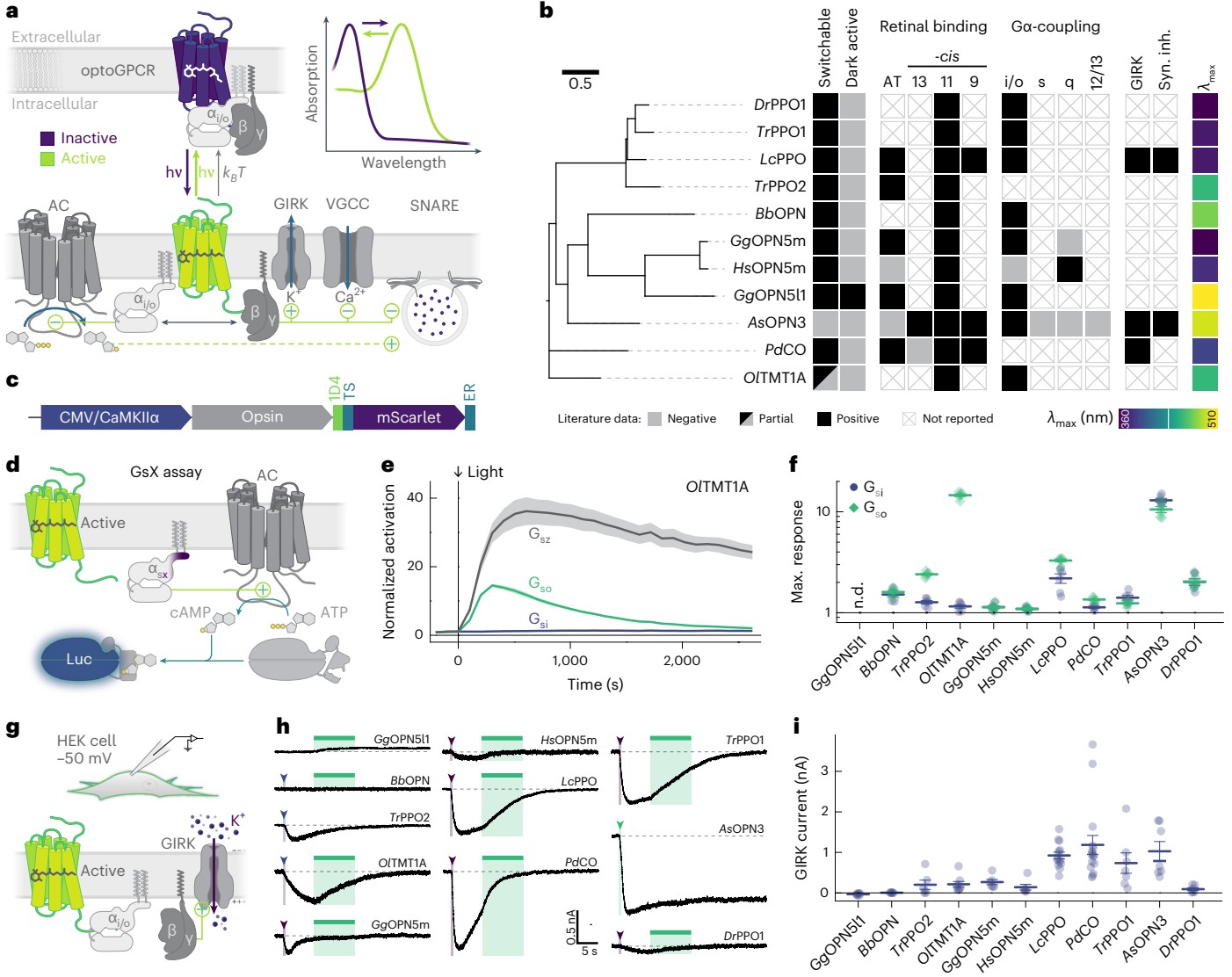

**Fig. 1 | Benchmarking of inhibitory optoGPCR candidates. a**, Scheme of inhibitory optoGPCRs that couple via the $G_{i/o}$ pathway. A dark, inactive optoGPCR bound to the heterotrimeric Gαβγ protein is shown. Once the optoGPCR is activated by light, the heterotrimeric G protein separates into the active Gα and Gβγ subunits. Gβγ activates GIRK channels, inhibits VGCCs and may interfere with the SNARE vesicle fusion apparatus. The Gα subunit inhibits ACs, thus reducing production of cAMP. OptoGPCRs can relax thermally ($k_BT$) to the non-signaling ground state. If their active state is spectrally separated from the ground state (see inset: absorption), absorption of a second photon with longer wavelength ($\lambda$) can terminate the signaling activity. hv, light energy. **b**, Overview of optoGPCR candidates investigated in this study. Left, phylogenetic tree of optoGPCRs. Right, properties of optoGPCRs as reported in available literature. AT, all-*trans*. **c**, Design of DNA constructs used for initial characterization. **d**, Characterization of each optoGPCR's Gα-protein signaling profile using the GsX assay.

Photoactivation of the optoGPCR activates a chimeric Gα_s subunit harboring the C terminus from one other Gα protein. This chimeric Gα protein activates an AC, which generates cAMP from ATP. cAMP activates a cAMP-dependent luciferase (Luc). **e**, Time course of averaged bioluminescence reads for Gα_sz, Gα_si and Gα_so activation by *Ol*TMT1A. The bioluminescence signal was normalized to the signal of cells not expressing optoGPCR and to pre-illumination baseline. A 1-s 470-nm light pulse was used for activation. $n = 6$. **f**, Maximum bioluminescence response for light-activated optoGPCRs coupling to Gα_si (blue circles) and Gα_so (green diamonds). n.d., not determined. $n = 6$. **g**, HEK cell experiments to measure optoGPCR-evoked GIRK currents with whole-cell voltage-clamp recordings. **h**, Representative GIRK current traces recorded in HEK cells expressing the indicated optoGPCRs. Arrowheads and narrow bars indicate light application of 0.5 s, while wide bars indicate 10-s light activation. **i**, Quantification of optoGPCR-evoked peak GIRK currents ($n = 6–16$). All data are shown as the mean ± s.e.m.

## Biophysical properties of *Pd*CO and G-protein specificity

Given its promising performance in autaptic neurons and its photochromic properties, we characterized *Pd*CO's biophysical properties in further detail in the context of synaptic inhibition and compared it with *Lc*PPO and *As*OPN3. First, we varied the wavelength of the activating 500-ms light pulse to generate action spectra for opsin activation, quantified from the average EPSC inhibition over 35 s after illumination (Fig. 2f). The wavelength needed for half-maximal EPSC inhibition of *Pd*CO was red-shifted by 40 nm compared to *Lc*PPO (Fig. 2g

and Extended Data Fig. 6). In addition, synaptic transmission at this wavelength range was more effectively reduced by *Pd*CO compared to *Lc*PPO (Fig. 2h). *Pd*CO activation with blue light (470 nm) showed transient inhibition that recovered with a time constant $\tau_{rec}$ of 3.4 ± 0.6 s (Fig. 2i). We next tested if continuous blue light illumination of *Pd*CO can evoke sustained inhibition that would similarly recover in the dark, without the need for a green pulse for inactivation. Indeed, continuous 470-nm illumination (2.83 mW mm$^{-2}$) for 60 s reduced EPSCs by 85% ± 1% (mean ± s.e.m., used hereinafter) in *Pd*CO-expressing neurons.

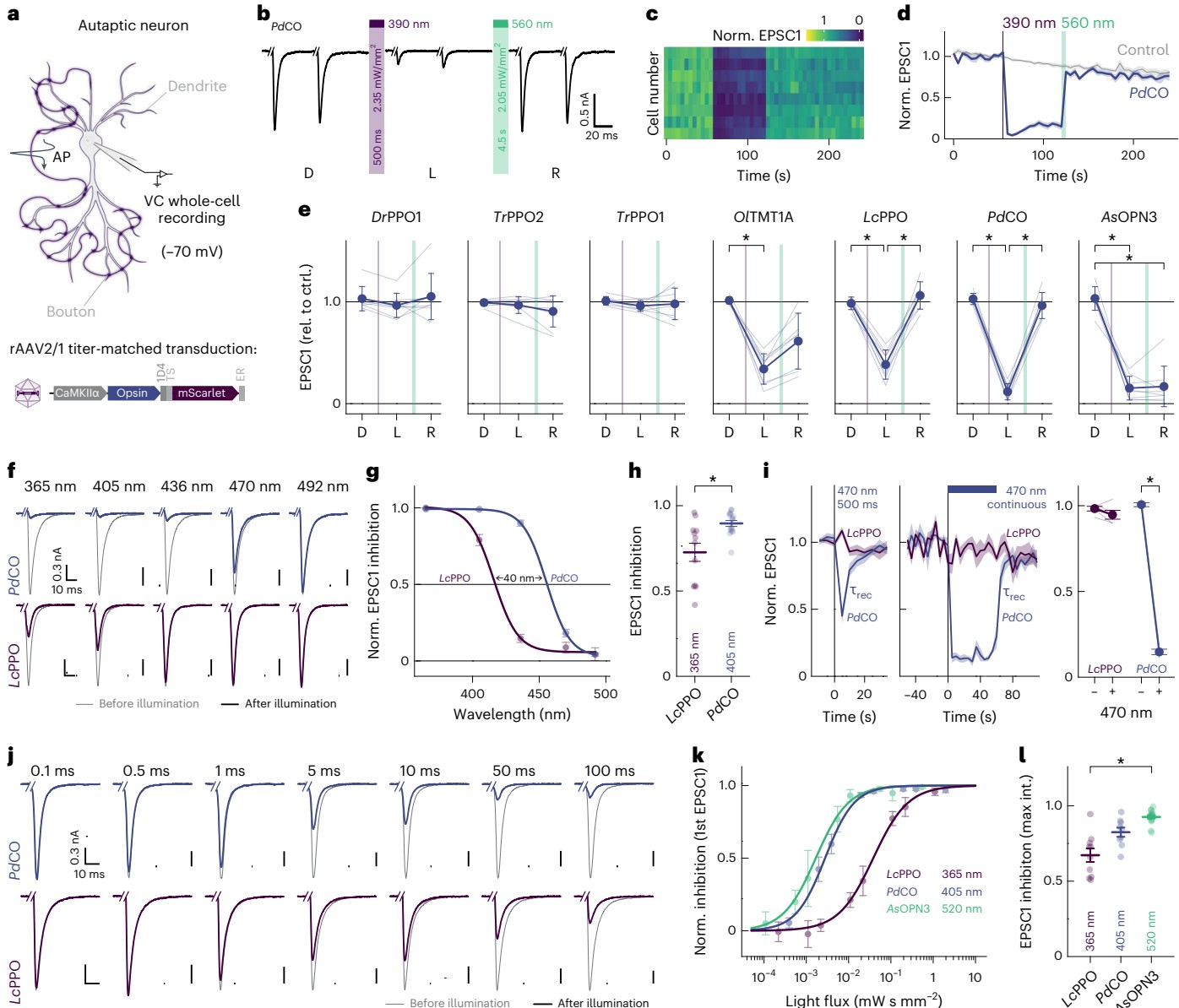

**Fig. 2 | Comparison of optoGPCR performance and biophysical characterization in autaptic neurons. a**, Schematic of an autaptic neuron recorded in the whole-cell configuration. Autaptic neurons were depolarized to fire unclamped APs, which triggered EPSCs. Neurons were transduced with rAAVs (bottom) encoding different optoGPCRs. **b**, Representative EPSC traces evoked by a pair of 1-ms depolarizing current injections (50-ms interstimulus interval, every 5 s) in a *Pd*CO-expressing autaptic neuron before (left; dark, D) and after (middle; light, L) UV-light illumination, followed by green light-induced recovery (right; recovery, R). Traces show five averaged sweeps. Current injection transients were removed for visualization. **c**, Contour plot representing EPSC amplitudes in eight neurons expressing *Pd*CO, activated with 390 nm of light for 500 ms and recovered by 560 nm of light for 4.5 s. EPSCs were normalized to the average amplitudes of five EPSCs before 390 nm of illumination. **d**, Averaged EPSC data (across replicates, *n* = 8) as shown in **c**, together with control EPSC recordings from non-expressing autaptic cultures measured with the same protocol. **e**, Quantification of EPSC inhibition for all optoGPCR candidates as shown in **b–d**. Data for dark (two EPSCs before UV light), light (two EPSCs after UV light and two EPSCs before green light) and recovery (two EPSCs 20 s after green light) were averaged and normalized for EPSC rundown by the same quantification of non-expressing control cells from matching autaptic cultures as shown in **c**. *$P < 0.05$; Friedman test followed by the Dunn–Sidak multiple-comparison test (two-sided); *Ol*TMT1A: $P(D, L) = 3.46 \times 10^{-3}$, $P(D, R)$ $1.78 \times 10^{-2}$; *Lc*PPO: $P(D, L) = 3.68 \times 10^{-2}$, $P(L, R) = 1.40 \times 10^{-3}$; *Pd*CO: $P(D, L) = 1.40 \times 10^{-3}$, $P(L, R) = 3.68 \times 10^{-2}$; *As*OPN3: $P(D, L) = 3.46 \times 10^{-3}$, $P(D, R) = 1.78 \times 10^{-2}$. Data are shown as the mean ± s.d., *n* = 8. rel. to ctrl., relative to control. **f**, Example EPSCs (average of seven) before illumination (gray) and after illumination (blue and purple), activated with light of different wavelengths (equal photon flux density) as indicated for *Pd*CO and *Lc*PPO from the same autaptic neuron, respectively. **g**, Quantification of normalized EPSC inhibition for different wavelengths. For each cell, EPSC inhibition at each wavelength was normalized to the maximum inhibition. Lines show the dose–response fit (*n* = 6–15). **h**, Quantification of the absolute EPSC inhibition at indicated wavelengths for experiments shown in **f** and **g** (*n* = 12–15). **i**, Time course of EPSC inhibition following blue light (470 nm) illumination for *Pd*CO and *Lc*PPO. Illumination with blue light for 0.5-s (left, *n* = 10–14) or 60-s (middle, *n* = 5) evoked EPSC inhibition by *Pd*CO but not *Lc*PPO. EPSCs recovered spontaneously for *Pd*CO with a time constant $\tau_{rec}$ (monoexponential fit). Right, quantification of the EPSC reduction by sustained application of blue light, *n* = 5. *$P < 0.05$; two-sided Wilcoxon rank-sum test; $P = 6.27 \times 10^{-3}$. **j**, Representative EPSC traces (average of seven) before (gray) and after illumination with different light pulse durations (blue and purple) recorded in the same autaptic neuron, for *Pd*CO and *Lc*PPO. **k**, Quantification of release inhibition after illumination versus light flux for *Pd*CO, *Lc*PPO and *As*OPN3, normalized to the inhibition for maximum light flux used. Solid lines show sigmoidal fits (*n* = 3–17). **l**, Quantification of the absolute EPSC inhibition over 30 s after illumination at indicated wavelengths and maximum light flux for experiments as shown in **j** and **k** (*n* = 9–17). *$P < 0.05$; Kruskal–Wallis test followed by two-tailed Dunn–Sidak multiple-comparison test; $P(Lc\text{PPO}, As\text{OPN3}) = 1.82 \times 10^{-4}$. Unless stated otherwise, all data are shown as the mean ± s.e.m. int., intensity.

Evoked EPSCs recovered spontaneously after the cessation of light, with a time constant of $2.7 \pm 0.3$ s. In contrast, we were not able to achieve inhibition with 470 nm of light for $Lc$PPO (Fig. 2i).

Next, we varied the light pulse duration at the maximal effective wavelengths to compare the light sensitivity of $Pd$CO, $Lc$PPO and $As$OPN3 (Fig. 2j). When quantifying the first EPSC after light activation, $Pd$CO ($EC_{50} = 3.1 \pm 0.4$ µW s mm$^{-2}$) showed similar sensitivity to $As$OPN3 ($EC_{50} = 1.9 \pm 0.3$ µW s mm$^{-2}$, $P = 0.3217$), whereas $Lc$PPO showed lower sensitivity with an $EC_{50}$ of $30 \pm 2$ µW s mm$^{-2}$ (Fig. 2k and Extended Data Fig. 6). At the maximum pulse duration, $As$OPN3 showed the strongest inhibition of ($93\% \pm 1\%$) followed by $Pd$CO ($82\% \pm 3\%$) and $Lc$PPO ($67\% \pm 4\%$; Fig. 2l).

We next tested whether $Pd$CO can be activated with two-photon excitation in HEK293T cells coexpressing GIRK channels. To obtain the two-photon action spectrum for $Pd$CO, we measured GIRK channel activation in cells expressing $Pd$CO using whole-cell patch-clamp electrophysiology (Extended Data Fig. 6). First, we applied raster scans at different wavelengths ranging from 700 nm to 1,100 nm (3 mW, 20-s raster scanning) while applying a voltage ramp from −120 mV to +40 mV. Maximum GIRK channel activation was achieved with 800 nm at −120 mV (Extended Data Fig. 6), in good agreement with one-photon activation. Next, we titrated the $Pd$CO-coupled GIRK activation at 800 nm by varying light intensity. The half-maximal activation was $0.49 \pm 0.2$ mW (Extended Data Fig. 6).

Our electrophysiological recordings showed stronger and faster GIRK-mediated hyperpolarization in neurons expressing $Pd$CO as compared to $As$OPN3 or $Lc$PPO. Treatment with the GIRK channel blocker SCH23390 reduced light-evoked currents by $77\% \pm 6\%$ in $Pd$CO-expressing neurons but did not affect inhibition of EPSCs (Extended Data Fig. 7), indicating that synaptic inhibition via $Pd$CO is independent of GIRK channel activity. Because $Pd$CO showed weak coupling to $G\alpha_o$ and $G\alpha_z$ in the GsX assay (Extended Data Fig. 2), we speculated that $Pd$CO might have a different G-protein signaling bias that leads to the observed differences in GIRK activation. We therefore used the TRUPATH assay[30] to characterize the G-protein signaling of $Lc$PPO, $As$OPN3 and $Pd$CO in more detail. Both $As$OPN3 and $Lc$PPO showed long-lasting coupling to all members of the inhibitory G-protein family ($G\alpha_{i-z}$). In contrast, $Pd$CO only coupled to $G\alpha_{oA/B}$ and $G\alpha_z$, and not to $G\alpha_{i1-3}$ (Extended Data Figs. 7 and 8). Treatment with pertussis toxin abolished light-induced inhibition of EPSCs for all three opsins, indicating that $G\alpha_z$ does not contribute to presynaptic inhibition by these opsins (Extended Data Figs. 7 and 8). As $G\alpha_i$ proteins are the main inhibitors of ACs, we tested whether $Pd$CO is capable of modulating cAMP production using a cAMP-dependent luciferase assay (GloSensor). As anticipated, $Pd$CO activation did not have any detectable effect on cAMP production in HEK cells only expressing $G\alpha_s$ and $G\alpha_i$, whereas $Lc$PPO and $As$OPN3 activation led to a bioluminescence signal decrease of $63\% \pm 1\%$ and $62\% \pm 1\%$, respectively (Extended Data Fig. 7).

### Presynaptic inhibition in organotypic hippocampal slices

We next aimed to assay the inhibition efficacy of $Pd$CO against $Lc$PPO, the only other photoswitchable optoGPCR using organotypic hippocampal slice cultures. First, we confirmed that the biophysical properties of these two opsins were similar to those characterized in the autaptic culture preparation. Individual CA3 pyramidal neurons were transfected by single-cell electroporation to express either $Pd$CO (Extended Data Fig. 9) or $Lc$PPO. We recorded GIRK-mediated currents evoked by light pulses at varying wavelengths and durations (Fig. 3a). The maximum GIRK current response for $Pd$CO-expressing neurons was between 405 nm and 435 nm (Fig. 3b). Peak GIRK currents evoked by $Pd$CO were higher than the ones induced by $Lc$PPO at all tested wavelengths, even at a tenfold lower light intensity for $Pd$CO. Next, we varied the illumination time at the peak activation wavelengths of both optoGPCRs (365 nm for $Lc$PPO and 405 nm for $Pd$CO). $Pd$CO-evoked GIRK currents showed maximum responses to light pulses with durations

between 50 and 100 ms, and a higher amplitude than those evoked by $Lc$PPO at the same pulse duration (Fig. 3c). We next activated the two optoGPCRs selectively at axonal terminals to compare their ability to suppress synaptic transmission. Presynaptic CA3 neurons were virally co-transduced with $Pd$CO or $Lc$PPO together with a soma-localized BiPOLES (somBiPOLES)[31], to elicit red light (625-nm)-evoked APs in CA3 while avoiding potential cross-activation by $Pd$CO illumination at CA1 (Fig. 3d). Red light pulses applied to the CA3 region reliably evoked EPSCs in CA1 cells, while application of a 100-ms light pulse at 365 nm (10 mW mm$^{-2}$) to $Lc$PPO-expressing terminals reduced EPSCs by $27\% \pm 4\%$ (Fig. 3e,f and Extended Data Fig. 9). Activation of $Pd$CO with tenfold lower light power at 405 nm led to a $78\% \pm 5\%$ reduction in synaptic transmission, while no EPSC reduction was observed when somBiPOLES was expressed alone (Fig. 3e,f and Extended Data Fig. 9). For both optoGPCRs, attenuation of synaptic transmission was reliably recovered with 525 nm of light (Fig. 3e,f).

We next measured the stability of inhibition of synaptic release by $Pd$CO, by stimulating $Pd$CO-expressing Schaffer collaterals with a bipolar electrode at 0.1 Hz, while recording EPSCs in CA1 neurons (Fig. 3g). To exclude any somatic effects of the opsin and to avoid antidromic and recurrent activation of the CA3 network, we dissected out area CA3 before the recordings. Local application of a brief 500-ms light pulse in CA1 reduced evoked postsynaptic currents (PSCs) by $71\% \pm 0.3\%$ and showed no spontaneous recovery over the time course of 25 min (Fig. 3h). This contrasts with $As$OPN3-mediated inhibition, which spontaneously recovers with a time constant of approximately 5 min under identical experimental conditions[13]. In addition, we were able to recover transmission with 525 nm of light and subsequently block synaptic transmission again with a second 405-nm pulse (Fig. 3h). Normalized EPSC amplitudes were not affected in non-expressing control cultures or non-illuminated $Pd$CO cultures (Extended Data Fig. 9).

### Single-photon spectral multiplexing with $Pd$CO

It is often informative to combine an optical readout of neuronal activity with optogenetic manipulations. For example, fiber photometry or miniature microscopes can be combined with light stimulation at a different wavelength in the single-photon domain[32,33]. This requires spectral multiplexing of different optogenetic sensors and actuators, and benefits from minimizing spectral cross-talk[1,34]. To establish whether $Pd$CO can be combined with red-shifted sensors or actuators, we analyzed the wavelength dependence of inactivation by varying the wavelength and irradiance of the inactivating pulse for both $Pd$CO and $Lc$PPO expressed in autaptic neurons. In these experiments, the optoGPCRs were activated at their peak excitation wavelength, and inactivation light was applied 30 s later (Fig. 4a). $Lc$PPO showed a broad wavelength sensitivity that enabled near-complete off-switching between 436 nm and 560 nm, while $Pd$CO's inactivation sensitivity was maximal between 470 nm and 520 nm (Fig. 4b and Extended Data Fig. 2b). We noted that the confined spectral window for inactivating $Pd$CO might present an opportunity for spectral multiplexing with other optogenetic probes that are activated by longer wavelengths. We therefore titrated the light sensitivity for both optoGPCRs at 560 nm and determined that EPSC recovery at this wavelength is sixfold more efficient for $Lc$PPO ($EC_{50} = 61 \pm 2$ µW mm$^{-2}$) compared to $Pd$CO ($EC_{50} = 372 \pm 163$ µW mm$^{-2}$; Fig. 4c), suggesting that $Pd$CO is better suited for multiplexing applications with red-shifted sensors or actuators.

We next explored spectral multiplexing using the red-shifted calcium indicator FR-GECO1c[35] (Fig. 4d). $Pd$CO, fused to EGFP for verification of expression, was coexpressed with FR-GECO1c in cultured neurons. In a tight-seal cell-attached patch-clamp configuration, we evoked APs at 0.2 Hz (Fig. 4d and Extended Data Fig. 9), resulting in reliable calcium transients in the FR-GECO1c signal (Fig. 4e). Blue light ($445 \pm 10$ nm) used to transiently activate $Pd$CO caused a $42\% \pm 12\%$ reduction in the amplitude of evoked calcium events (Fig. 4e,f).

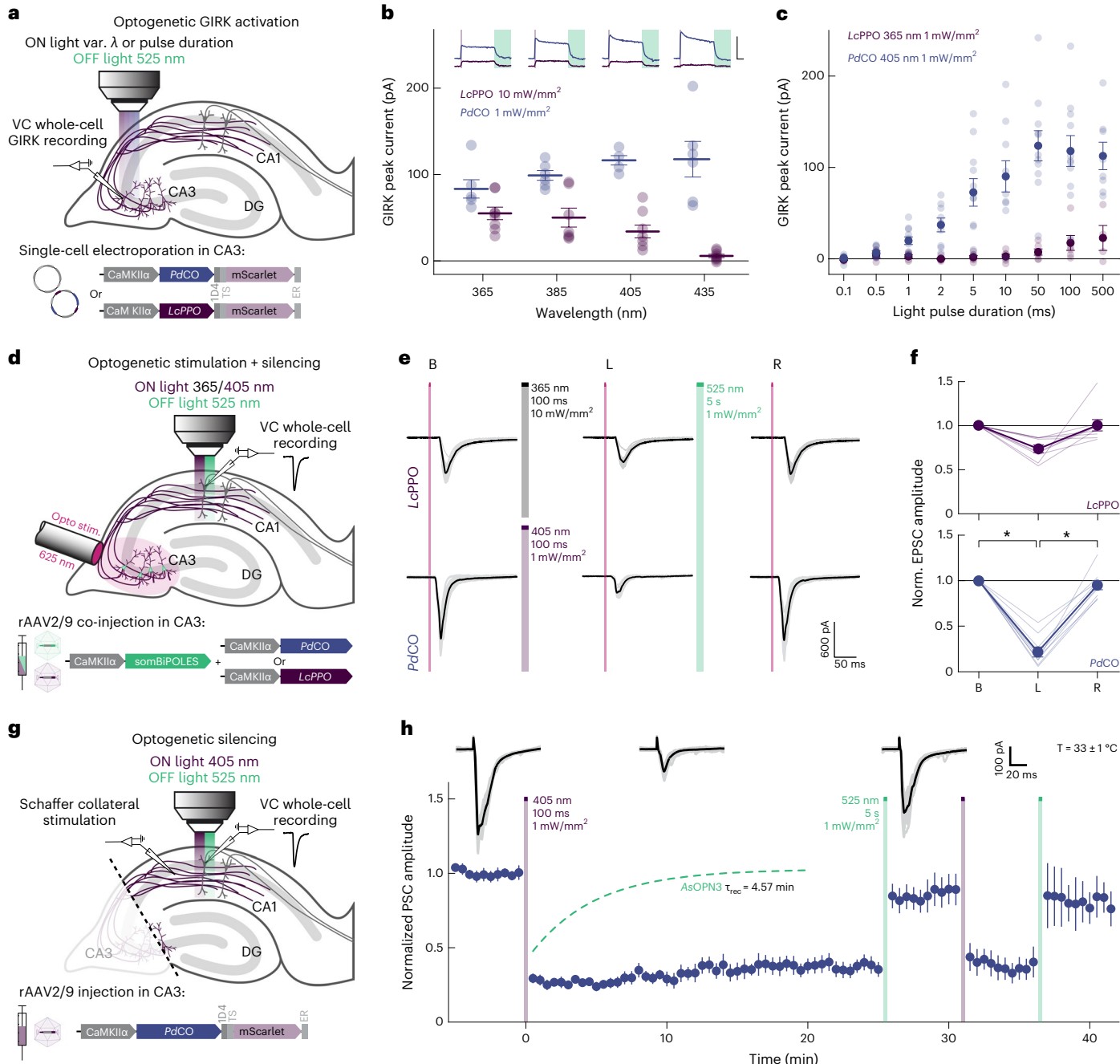

**Fig. 3 | Characterization and performance of optoGPCRs in organotypic hippocampal slice cultures. a**, Illustration of experimental setup in organotypic hippocampal slice cultures (single-cell plasmid electroporation, circles). Activation spectrum and light sensitivity were measured by recording GIRK-mediated currents from CA3 neurons expressing either *Pd*CO or *Lc*PPO in response to optogenetic stimulation through the microscope's objective at varying light parameters. **b**, Quantification of GIRK current amplitudes recorded in cells expressing *Pd*CO and *Lc*PPO at the indicated light wavelengths and intensities. Activation light pulse, 500 ms. Inactivation light pulse, 5 s at 525 nm. Inset scale bar, 100 pA, 2 s. *n* = 6–8. **c**, Quantification of GIRK current amplitudes recorded in cells expressing *Pd*CO and *Lc*PPO at their optimal activation wavelength at varying light durations (*n* = 4–14). **d**, Illustration of experimental design for bidirectional optogenetic control of synaptic transmission. **e**, Representative current traces of patched CA1 neurons in

response to optogenetically induced presynaptic APs (625 nm, 5 ms) under baseline conditions (left), after activation (middle) and after inactivation (right) of the optoGPCRs. Gray, single trials; black, averaged traces. **f**, Normalized EPSC amplitudes from *Lc*PPO and *Pd*CO groups. *n* = 9. *$P < 0.05$; repeated-measures one-way analysis of variance (ANOVA) with Geisser–Greenhouse correction followed by Tukey's comparison; *Pd*CO: $P(B, L) < 1.00 \times 10^{-4}$, $P(L, R) = 1.00 \times 10^{-4}$. **g**, Illustration of experimental setup for electrical stimulation of Schaffer collaterals and optogenetic inhibition of synaptic transmission. **h**, Time course of normalized PSC amplitudes from all the recorded postsynaptic CA1 neurons, before, during and after activation/inactivation of *Pd*CO. Representative voltage-clamp traces are shown on top. Gray, single trials; black, average trials. Light was applied locally in CA1 for activation and inactivation of *Pd*CO (ON light: 500 ms, 405 nm, 1 mW mm$^{-2}$; OFF light: 5 s, 525 nm, 1 mW mm$^{-2}$). *n* = 3–7. All data are shown as the mean ± s.e.m.

Notably, as reported for other GECO variants[36], blue light application alone increased FR-GECO1c fluorescence, for which we corrected in our analysis (Extended Data Fig. 9). As GIRK activation can lead to reduced

excitability or even suppression of AP firing, we blocked GIRK channels using SCH23390. Blue light application still decreased calcium transients by 18% ± 4%, while in control cells only expressing FR-GECO1c,

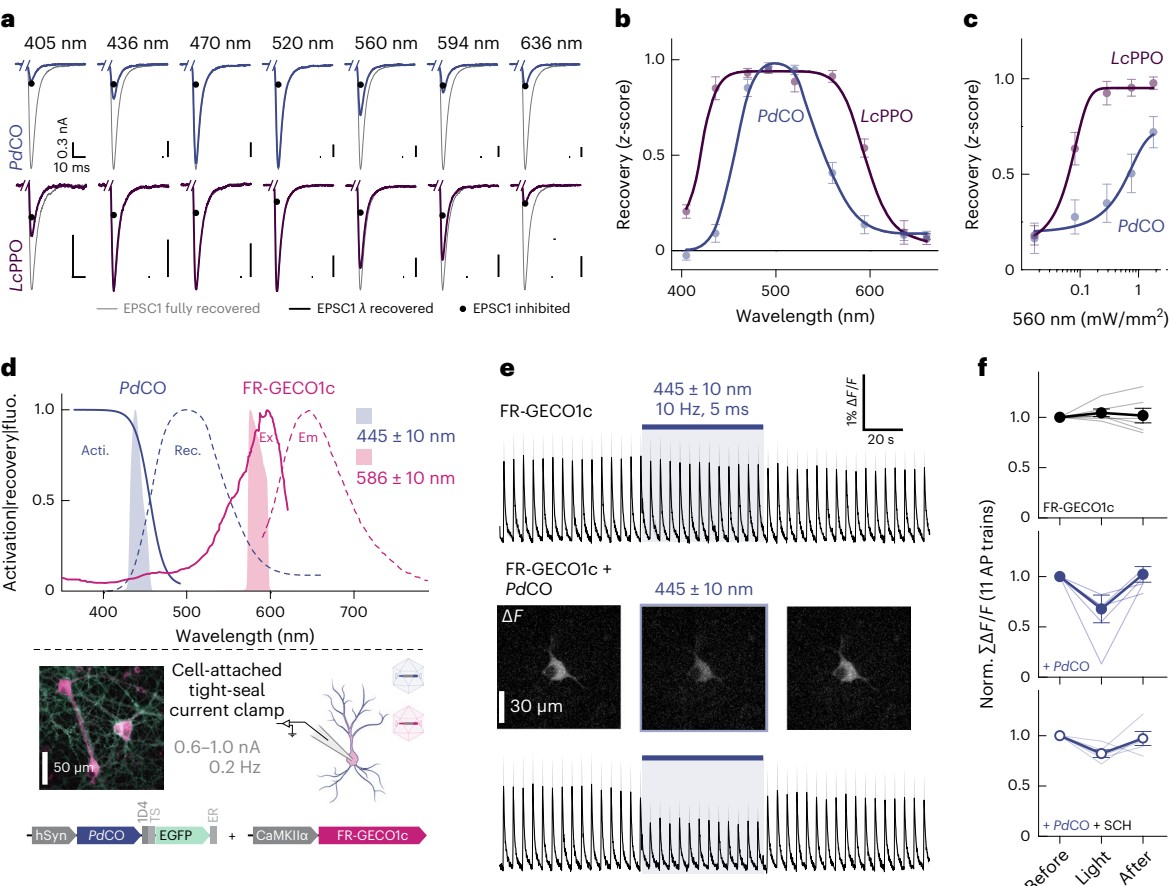

**Fig. 4 | *Pd*CO recovery properties and multiplexing with FR-GECO1c.**
**a**, Example traces of experiments used to determine the spectral features and light sensitivity of optoGPCR inactivation in autaptic neurons. Samples were first illuminated with 390-nm (*Lc*PPO) or 405-nm (*Pd*CO) light for 500 ms, to inhibit EPSCs (black circles), followed by recovery with light delivery at the indicated wavelengths (equal photon flux density) for 4.5 s (colored traces) and finally completely recovered with 520 nm for at least 10 s (gray traces). An average of seven EPSC traces are shown, scaled to the fully recovered EPSCs. **b**, Wavelength sensitivity of light-induced recovery. To correct for potential EPSC rundown, recovery $z$-scores were calculated using the mean of four EPSCs after inhibition, prior recovery light at different wavelengths and the mean of four EPSCs after full recovery with green light. $n = 4$–7. **c**, Light titration of light-induced recovery. Experiments were conducted as in **b** but at a fixed wavelength of 560 nm, while varying the light intensity between trials. $n = 7$–8. **d**, Top, spectra of *Pd*CO

activation (solid blue line) and inactivation (dashed blue line), excitation (solid magenta line) and emission (dashed magenta line) spectra of FR-GECO1c[35] and stimulation light properties used for activation/excitation (blue/magenta shaded areas). Bottom (upper left), representative epifluorescence pseudocolor image of neuronal culture transduced with rAAVs encoding *Pd*CO-EGFP and FR-GECO1c. Bottom (upper right), schematic approach of the cell-attached tight-seal patch-clamp configuration. Bottom, constructs used for spectral multiplexing calcium imaging experiments. **e**, Average (across replicates) FR-GECO1c calcium traces during repetitive current injections (top) and with additionally expressing *Pd*CO (lower trace). Example images from left to right show the averaged signal before, during and after blue light illumination. **f**, Quantification of FR-GECO1c only (top), coexpressed with *Pd*CO (middle) and additionally blocking GIRK currents with SCH23390 (SCH) (bottom). $n = 5$–6. All data are shown as the mean ± s.e.m. $\Delta F/F$, change in fluorescence intensity.

no reduction of calcium transients was detected (Fig. 4e,f). Consistent with previous work[13,14], this indicates that *Pd*CO activation leads to the attenuation of somatodendritic VGCC activity.

Next, we combined *Pd*CO with a soma-targeted variant of the red light-sensitive channelrhodopsin ChrimsonR[37] in a single bicistronic construct to allow the triggering of APs with red light, while simultaneously inhibiting synaptic transmission with the blue light-sensitive *Pd*CO (Fig. 5). In cultured hippocampal neurons expressing this bicistronic construct, red light pulses (5 ms, 632 nm) generated photocurrents above 900 pA that reliably induced APs (Fig. 5b,c). In non-expressing neurons, the same red light pulses caused reliable PSCs (Fig. 5b). When activating *Pd*CO by a brief 390-nm light pulse (100 ms), repeated red light pulsing did not evoke PSCs, indicating effective *Pd*CO-mediated inhibition of synaptic transmission (Fig. 5d–g). Following green light application (512 nm) to recover transmission, red light-evoked PSCs were readily detectable again (Fig. 5d,g). Inhibition of evoked synaptic transmission could be achieved in a repetitive manner, without any detectable switching fatigue of *Pd*CO (Fig. 5d–g).

## *Pd*CO applications in vivo

To establish the efficacy of *Pd*CO in modulating mouse behavior, we used it to unilaterally inhibit dopaminergic projections from the substantia nigra to the dorsomedial striatum, a neural pathway that plays an important role in animal locomotion[38]. We activated *Pd*CO unilaterally in these axons during free locomotion and measured the resulting side bias[13]. We expressed *Pd*CO (or EYFP as control) bilaterally in substantia nigra pars compacta dopaminergic neurons and implanted bilateral optical fibers above the dorsomedial striatum (DMS; Fig. 6a). Unilateral light activation caused an ipsiversive rotational bias in *Pd*CO-expressing mice (Fig. 6b) that accumulated over time and ceased after illumination with green light (Fig. 6c). This effect was consistent across *Pd*CO-expressing mice and absent in the EYFP-expressing control group (Fig. 6c,d).

To further test how *Pd*CO inhibits specific synapses in vivo, we focused on locus coeruleus norepinephrine (LC-NE) modulation of pupil size[39–41], which is largely mediated by disinhibition of the parasympathetic Edinger–Westphal (EW) nucleus[42]. LC neurons form

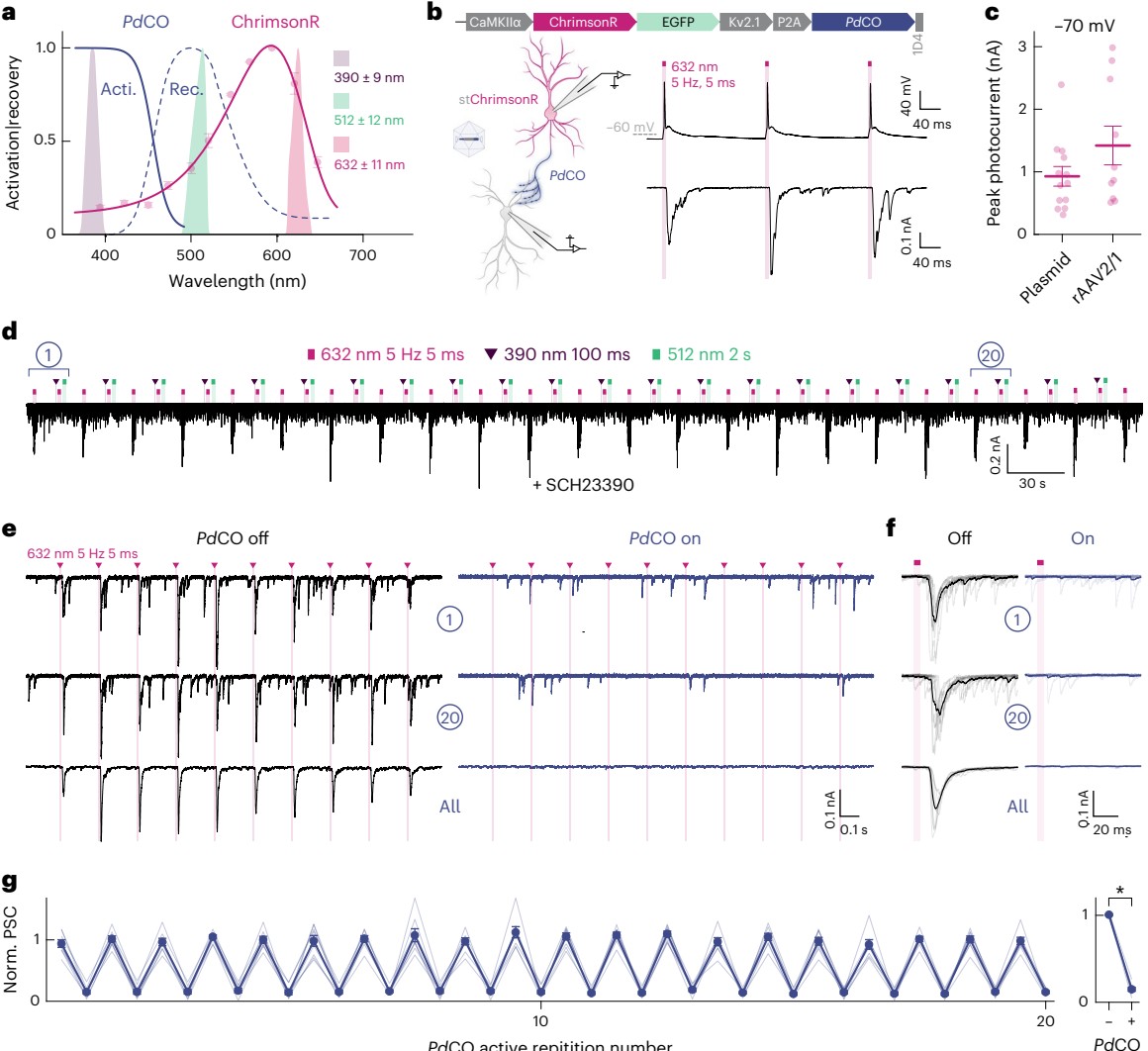

**Fig. 5 | Single-photon spectral multiplexing of *Pd*CO with ChrimsonR.**
**a**, Spectra of *Pd*CO activation (solid blue line) and inactivation (dashed blue line), together with the action spectra recorded from stChrimsonR-EGFP-P2A-*Pd*CO. Expressing neurons measured with tetrodoxin (TTX), cyanquixaline (CNQX) and 2-amino-5-phosphonovalerate (AP-5) at −70 mV holding potential. Action spectra were recorded twice per cell in both directions (UV to red and vice versa) with equal photon flux density (2-ms light pulse) and then averaged. Stimulation light properties (purple, green and magenta shaded areas) are shown. $n = 7$.
**b**, Construct design (top), schematic of expression (bottom left) and experiment (bottom right) are shown. Red light activation (632 nm, 5 ms, 5 Hz) evoked APs in stChrimsonR-EGFP-P2A-*Pd*CO-expressing cells (upper trace), while in non-expressing cells a pronounced PSC could be recorded (lower trace).
**c**, Quantification of maximum photocurrent amplitudes in stChrimsonR-EGFP-

P2A-*Pd*CO-expressing cells mediated by either calcium phosphate transfection or viral transduction. $n = 10$–13. **d**, Representative recording of a postsynaptic non-expressing cell, where red light application evoked reliable PSCs, while activation of *Pd*CO inhibits synaptic transmission. *Pd*CO activity was toggled between on and off 22 times. **e**, Comparison of first (top traces) and 20th *Pd*CO activation (middle traces) as well as the average across all repetitions (bottom traces) from the recording shown in **d**. **f**, Average of the ten red light-induced PSCs with *Pd*CO inactive (left) or active (right) for the first (top) and the 20th (middle) toggling cycle, as well as for the average across all repetitions (bottom) for the traces shown in **e**. **g**, Left, quantification of experiments shown in **d**–**f** for eight biological replicates. Right, average PSCs per neuron, with and without *Pd*CO activation, $n = 8$. *$P < 0.05$; two-sided Wilcoxon rank-sum test; $P = 0.0078$. All data are shown as the mean ± s.e.m.

widespread projections that terminate in multiple brain regions[43], making somatic inhibition nonspecific. To selectively suppress LC axons terminating in the EW nucleus, we conditionally expressed *Pd*CO unilaterally in NE neurons of the LC and implanted optical fibers above the ipsilateral EW, and above the basal forebrain (BF, as control region; Fig. 6e and Extended Data Fig. 10). Blue light application (447 nm) to EW led to robust dose-dependent pupil constriction, whereas identical stimulation of the BF did not (Fig. 6f,g). Compared to mCherry-expressing control mice, a pronounced ipsilateral pupil constriction difference was detected for *Pd*CO-expressing mice at light stimulation frequencies between 10 Hz and 40 Hz (Fig. 6h). Notably, the ipsilateral pupil was substantially more affected by laser stimulation than the contralateral pupil. Given that pupil asymmetry does not occur

under physiological conditions[44], the observed lateralization provides strong evidence for pathway-specific inhibition.

We next tested how *Pd*CO-mediated inhibition of synapses in vivo affects motivated behavior. Photostimulation of nucleus accumbens (NAc)-ventral tegmental area (VTA) D1/dyn terminals with channelrhodopsin negatively impacts feeding behavior, whereas photoinhibition of D1/dyn neurons in the NAc projecting to the VTA enhances it[45,46]. Hence, to demonstrate the utility of *Pd*CO to inhibit peptidergic terminals and impact behavior, we used *Pd*CO to silence dynorphin (dyn) terminals projecting from the NAc to the VTA during cued reward consumption behavior. Following injection of *Pd*CO in the NAc and fiber implantation in the VTA in either *Pdyn-Cre* or wild-type (WT) mice (Fig. 6i,j), we trained food-restricted animals on a cued reward

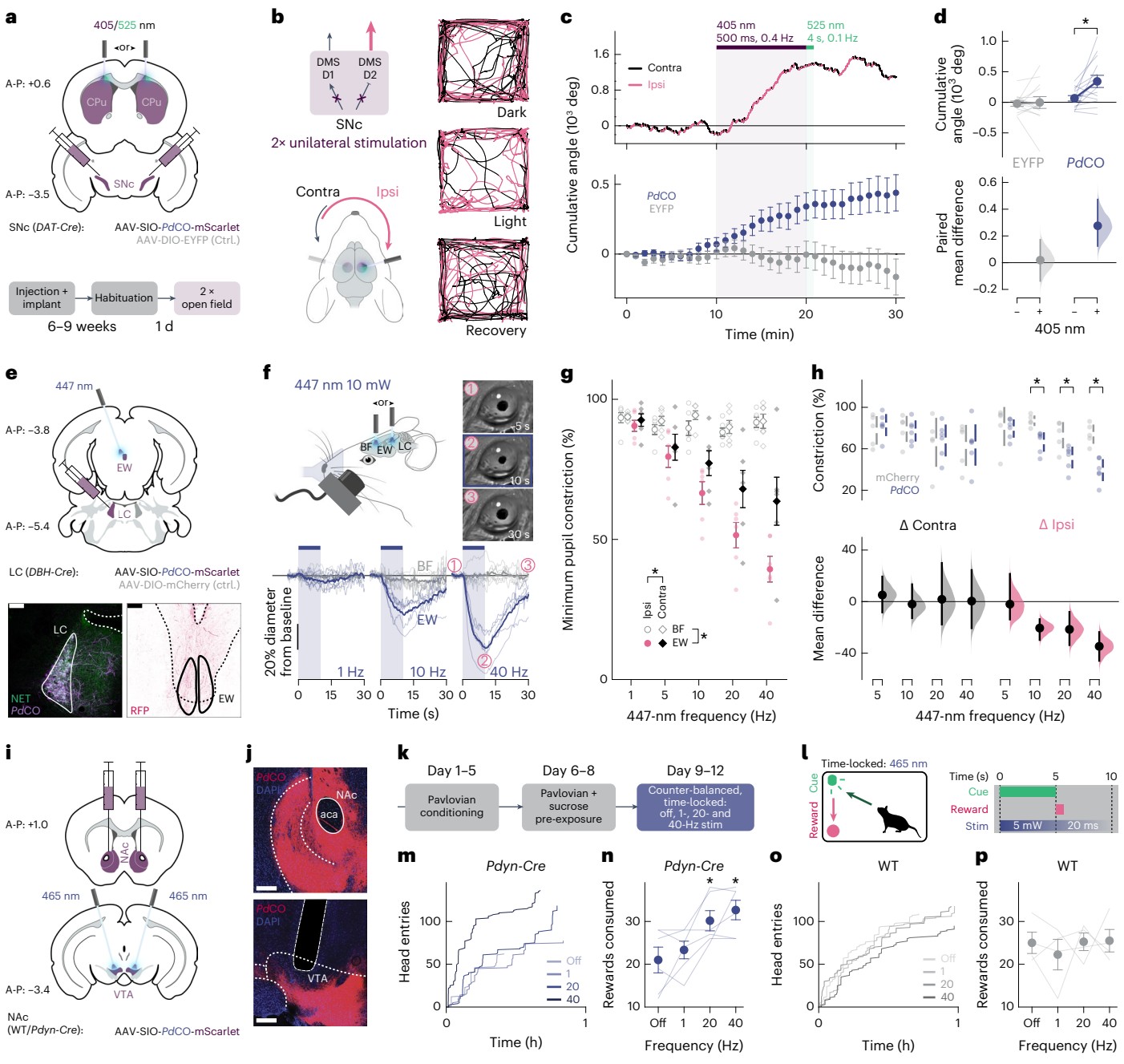

delivery task, where they learned to associate a cue with delivery of sucrose pellets. Once the mice consistently consumed all the reward pellets, mice received 0.25 g of sucrose before the session to ensure that we are able to bidirectionally modulate behavior and received stimulation of 465 nm of light at varying frequencies (off, 1 Hz, 20 Hz and 40 Hz, 20-ms pulse width) time-locked to cue presentation for 10 s across counterbalanced sessions (Fig. 6k,l). *Pdyn-Cre* animals that received 20-Hz or 40-Hz light pulses to the VTA increased their food consumption, relative to the sessions where no light was delivered (Fig. 6m,n). In contrast, in WT controls, light delivery at any frequency did not alter consumption (Fig. 6o,p).

As *Pd*CO worked efficiently in rodent neurons in vitro and vivo, we further tested its application in non-neuronal tissue and in two invertebrate neuronal model systems. We demonstrate that *Pd*CO can suppress spontaneous beating in neonatal atrial cardiomyocytes and suppress behavior in the nematode *Caenorhabditis elegans* and *Drosophila* larvae (Supplementary Information and Supplementary Figs. 1–3).

## Discussion

Efficient presynaptic inhibition offers opportunities to study projection-specific contributions to behavior. Although the absorption maxima of purified *Pd*CO (383 nm)[26] and *Lc*PPO (370 nm)[23] differ only by 13 nm, we were able to activate *Pd*CO with wavelengths up to 470 nm. In line with previous reports[24], *Lc*PPO could only be activated up to 405 nm. *Pd*CO's red-shifted activation spectra can be explained by a low absorption cross-section of the active state that shifts the equilibrium wavelength between activation and light-induced recovery toward the red spectrum. The lower probability of light-induced inactivation is also reflected in the inactivation spectrum of *Pd*CO, which is narrower compared to that of *Lc*PPO. Intriguingly, *Lc*PPO has been reported to be activatable with blue light (around 470 nm) in various experimental settings[14]. However, we were not able to achieve true blue light activation of *Lc*PPO. Instead, we efficiently inhibited *Lc*PPO activity in our experiments using 470 nm as reported previously[24]. This discrepancy might result

**Fig. 6 | *Pd*CO applications in vivo. a**, Experimental setup and timeline for silencing of the nigrostriatal pathway. Top, schematic of injection sites, expression areas and fiber placement. Bottom, experimental timeline. **b**, Left, bilateral expression of *Pd*CO in substantia nigra pars compacta (SNc) dopaminergic neurons and unilateral light-mediated suppression of their striatal projections would induce an ipsiversive side bias during free locomotion. Right, representative locomotion trajectories of *Pd*CO mice, over successive 10-min periods before, during and after light delivery (top to bottom). Magenta and black colors depict ipsilateral and contralateral angle trajectory segments, respectively. **c**, Top, representative cumulative angle traces of the individual *Pd*CO-expressing mice shown in **b**, over 30 min of free locomotion in an open field arena. Magenta and black colors depict ipsilateral and contralateral segments, respectively. Bottom, average cumulative angle across *Pd*CO-expressing (blue) and EYFP-expressing control mice (gray). Each mouse underwent two unilateral stimulations of each hemisphere, respectively, that was then averaged per mouse. $n = 13$–14. **d**, Quantification of the accumulated angle prior illumination (min. 9) compared to post-UV illumination (min. 20) for *Pd*CO-expressing (blue, $n = 14$) and EYFP-expressing control mice (gray, $n = 13$). The paired mean difference for both comparisons is shown in the Cumming estimation plot. Each paired mean difference is plotted as a bootstrap sampling distribution. Mean differences are depicted as dots; 95% confidence intervals are indicated by the ends of the vertical error bars. *$P < 0.05$; two-sided permutation $t$-test; $5.80 \times 10^{-3}$. **e**, Schematic of pupil experiment (top; Methods) and histology of LC (left) showing staining against norepinephrine transporter (NET) and *Pd*CO-mScarlet expression (right) (bottom). Staining against red fluorescent protein (RFP, mScarlet) in the EW nucleus. Scale bars, 100 μm. **f**, Top, schematic representation of the experiment (left) and representative frames (right) from pupil video recording at the indicated time points relative to 40-Hz laser stimulation onset as shown in the plots below. Plots depict the mean pupillometry traces (bold lines, $n = 6$) for 1 Hz (left), 10 Hz (middle) and 40 Hz (right) of laser stimulation. Each plot denotes the median time course of ipsilateral pupil diameter across trials in each subject when illuminating the BF (gray) or the EW nucleus (blue).

The vertical blue shaded area represents laser stimulation interval. Thin traces, individual mice; thick traces, mean ($n = 6$). **g**, Average pupil constriction ($n = 6$ *Pd*CO mice), matching the minimal value in time courses shown in **f** as a function of stimulation frequency ($x$ axis). Magenta dots and black diamonds represent the EW stimulation effect on the ipsilateral pupil and the contralateral pupil, respectively. Bold dots/diamonds, average across mice ($n = 6$). Error bars indicate the s.e.m. across mice. Light dots, average of trials in each mouse. *$P < 0.05$; multiple-way ANOVA; $P$(placement) $= 1.22 \times 10^{-20}$, $P$(frequency) $= 9.62 \times 10^{-12}$, $P$(eye laterality) $= 1.98 \times 10^{-4}$, $P$(placement × frequency) $= 5.42 \times 10^{-10}$, $P$(placement × eye laterality) $= 8.71 \times 10^{-3}$. **h**, The mean difference for eight comparisons (four contralateral, four ipsilateral) between *Pd*CO ($n = 6$) and mCherry control ($n = 5$) is shown for different blue light stimulation frequencies in the Cumming estimation plot. The raw data are plotted on the upper axes; each mean difference is plotted on the lower axes as a bootstrap sampling distribution. Mean differences are depicted as dots; 95% confidence intervals are indicated by the ends of the vertical error bars. *$P < 0.05$; two-sided permutation $t$-test; ipsilateral *Pd*CO versus control: 0.0026 (10 Hz), 0.0250 (20 Hz) and 0.0016 (40 Hz). **i**, Coronal brain schematic of viral injection of *Pd*CO into the NAc and fiber implantation into the VTA of *Pdyn-Cre* mice. **j**, Representative ×20 coronal images showing expression of *Pd*CO-mScarlet (red) and DAPI (blue) in the NAc (top) and projections as well as fiber placement in the VTA (bottom). Scale bar, 200 μm. **k**, Cartoon outlining experimental timeline of training and cued reward delivery testing. **l**, Cartoon outlining experimental procedure of stimulation during cued reward delivery testing. **m**, Representative trace showing head entries across the 60-min session for each experimental condition in *Pdyn-Cre* mice. **n**, Significant increase in reward consumption following NAc-VTA dynorphin terminal stimulation at 20 Hz and 40 Hz in *Pdyn-Cre* mice. *$P < 0.05$; one-way repeated-measures ANOVA followed by multiple comparisons versus off: $P = 0.6958$ (1 Hz), 0.0119 (20 Hz) and 0.0026 (40 Hz). Data are represented as the mean ± s.e.m., $n = 6$. **o**, Representative trace showing head entries across the 60-min session for each experimental condition in WT mice. **p**, No change in reward consumption following stimulation in WT mice. Data are the mean ± s.e.m., $n = 4$. All data points represent individual animals.

from bandwidth-limited light in our experiments, which eliminated low-wavelength photons.

Transient synaptic inhibition observed when *Pd*CO was activated with blue light (for example, 445–470 nm) could indicate that only a small number of the activated G proteins are recruited, consistent with similar effects in chemogenetic actuators[47]. Although we demonstrated long-lasting synaptic inhibition in organotypic slice preparations, care should be taken when using *Pd*CO for long-lasting synaptic silencing experiments following only a brief single light pulse activation. Especially for in vivo experiments, if light delivery and expression levels are below saturation, *Pd*CO-mediated inhibition could be short-lived; this can be overcome by repetitive light application or increased opsin expression. Furthermore, optoGPCR kinetics might vary between cell types, availability of heterotrimeric G-protein subunits and effectors/targets[48], and input-specific AP frequency and membrane depolarization[49–51]. Therefore, the inhibitory effect of presynaptic optoGPCRs should be tested by recording postsynaptic input reduction over time as discussed elsewhere[16]. Such experiments would be facilitated by the bicistronic construct described above, which allows coexpression of the red-shifted ChrimsonR with *Pd*CO in the same neurons.

Testing the same optoGPCR with different established assays of GPCR signaling can lead to vastly different, and even contradictory, outcomes. OPN5 homologs, for example, which did not couple to either $G\alpha_q$ or $G\alpha_{i/o}$ in the GsX assay, generated a GIRK response in our hands. These optoGPCRs have been described to mediate $G\alpha_q$ coupling in various settings[52,53] and have been shown to preferentially couple via $G\alpha_{14}$ (ref. 54). For *Pd*CO, we observed efficient GIRK coupling as shown previously[26] but could only demonstrate very weak $G\alpha_o$ coupling in the GsX assay. The TRUPATH assay, however, revealed that *Pd*CO selectively couples to $G\alpha_o$ but not to $G\alpha_i$, which was confirmed by the demonstration that *Pd*CO activation does not inhibit cAMP production in HEK cells lacking in $G\alpha_o$. Nevertheless, we found that *Pd*CO allows

efficient silencing of presynaptic transmission, indicating that selective activation of the $G\alpha_o$ pathway can strongly suppress presynaptic release in all preparations tested in our study. The lack of *Pd*CO impact on AC activity and the absence of effects on presynaptic cAMP can offer potential benefits as cAMP is involved in various intracellular processes such as proliferation, differentiation, survival, long-term synaptic potentiation, neurogenesis and neuronal plasticity. However, potential modulation of cAMP by *Pd*CO activation should not be completely excluded as a variety of ACs have been reported to be affected by different $G\alpha$ and $G\beta\gamma$ subunits including $G\alpha_o$[55]. It has also been shown that *Pd*CO can transiently recruit $G\alpha_i$ under long-lasting continuous and/or high-intensity illumination[55,56], potentially by depleting available $G\alpha_o$ over time and, therefore, generating a signaling bias toward $G\alpha_i$.

While our primary focus in this study has been to develop optoGPCRs for presynaptic inhibition, *Pd*CO could be used as a tool to reduce neuronal excitability when activated at the soma. However, not all neurons express GIRK channels and thus somatic inhibition might be absent in some cell types (for example, medium spiny neurons in the striatum). Thus, when somatic inhibition is desired, anion-conducting or potassium-conducting channelrhodopsins[10,57,58] might be more suitable, due to their strong inhibitory photocurrents and their millisecond-scale decay kinetics upon light offset. Nonetheless, by blocking GIRK channel activity, we demonstrated that *Pd*CO-mediated synaptic attenuation of transmission is independent of GIRK activity and can, therefore, be applied in neurons lacking these channels.

For *Pd*CO expression across various preparations, we did not observe any discernible modifications of intrinsic neuronal cell parameters or effects on baseline behavior compared to vertebrate control cells or animals and *C. elegans*. In *Drosophila* larvae, an increased behavioral response was noted for functionally expressed *Pd*CO compared to control animals. However, it should be noted that high-level

overexpression of any exogenous protein can lead to impairment in neuronal cell health. We therefore recommend that users test for such alterations at the cellular, circuit and behavioral levels and adhere to the lowest possible expression levels that allow an adequate inhibitory effect of *Pd*CO.

The diversity of genetically encoded actuators and sensors provides a wealth of opportunities for multiplexed experiments, combining two or more such tools in a single experimental setting. As the activation spectrum of *As*OPN3 covers the entire UV-visible range and due to its high light sensitivity, it requires careful handling and cannot be combined with other optical approaches apart from two-photon imaging[13]. In contrast, *Lc*PPO and *Pd*CO are both activated on the high-energy visible spectrum (UV to blue light) and, therefore, do not bear the risk of cross-activation by other wavelengths used for imaging or optogenetic control. The narrow action spectrum of *Pd*CO's light-induced back-reaction to the inactive state is an attractive property for multiplexing with genetically encoded tools that have red-shifted excitation spectra. Whereas one-photon multiplexing is possible with *Lc*PPO[14], we found that application of cyan to red light can cause a stronger inactivation of *Lc*PPO compared to *Pd*CO. For activation of larger brain areas, *As*OPN3 might serve as a more suitable inhibitory optoGPCR due to its red-shifted activation spectrum and high light sensitivity. However, independent *As*OPN3 activation at different brain loci might be less feasible due to potential cross-excitation by scattered photons. In this case, *Lc*PPO and *Pd*CO could serve as an alternative as short-wavelength light is more effectively attenuated in neuronal tissues. Since *As*OPN3 can also be activated by UV to blue light, these wavelengths can be used to excite *As*OPN3 in settings where slow kinetics are desirable and activation by scattered light is a concern. In contrast, *Pd*CO can provide faster onset and termination of inhibitory signaling. However, the rate-limiting steps in signaling kinetics will be determined in all cases by the availability and mobility of the Gα and Gβγ subunits.

Taken together, our results demonstrate that *Pd*CO is a rapid, reversible and versatile optoGPCR that mediates efficient silencing of glutamatergic and neuromodulatory synaptic transmission in diverse cell types in vitro and in vivo that expands and complements the collection of presynaptic optogenetic tools[16,59]. For manipulating the presynapse, *Pd*CO could potentially serve as a suitable template to create optoGPCR chimeras with altered signaling specificity by exchanging the intracellular GPCR interface as previously demonstrated for other rhodopsin GPCRs[60–70]. *Pd*CO's biophysical properties are suitable for one-photon spectral multiplexing approaches, which are becoming more common in the systems neuroscience field. We believe that *Pd*CO, along with existing optogenetic sensors and with future improved, red-shifted indicators of neuronal activity, will serve as a valuable tool that will allow a better understanding of long-range neural communication in the brain.

## Online content

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

[1]Department of Brain Sciences, Weizmann Institute of Science, Rehovot, Israel. [2]Department of Molecular Neuroscience, Weizmann Institute of Science, Rehovot, Israel. [3]Center for Molecular Neurobiology, Hamburg, Germany. [4]Sagol School of Neuroscience, Tel Aviv University, Tel Aviv, Israel. [5]Department of Anesthesiology and Pain Medicine, University of Washington, Seattle, WA, USA. [6]Center for Excellence in the Neurobiology of Addiction, Pain and Emotion, University of Washington, Seattle, WA, USA. [7]Institut für Physiologie I, University of Bonn, Bonn, Germany. [8]Washington University Pain Center, Department of Anesthesiology, Washington University School of Medicine, St. Louis, MO, USA. [9]LIMES-Institute, University of Bonn, Bonn, Germany. [10]Department of Physiology and Pharmacology, Tel Aviv University, Tel Aviv, Israel. [11]German Center for Neurodegenerative Diseases (DZNE), Berlin, Germany. [12]Neuroscience Research Center, Charité – Universitätsmedizin Berlin, Berlin, Germany. [13]Bernstein Center for Computational Neuroscience, Berlin, Germany. [14]Einstein Center for Neurosciences, Berlin, Germany. [15]Max Delbrück Center for Molecular Medicine, Berlin, Germany. [16]Department of Pharmacology, University of Washington, Seattle, WA, USA. [17]Institute of Physiology and Pathophysiology, Friedrich-Alexander-Universität Erlangen-Nürnberg, Erlangen, Germany. [18]Department of Biomedical Engineering, Faculty of Engineering, Tel Aviv University, Tel Aviv, Israel. [19]Present address: Neuroscience Research Center, Charité – Universitätsmedizin Berlin, Berlin, Germany. [20]Present address: Paris Brain Institute, Institut du Cerveau (ICM), CNRS UMR 7225, INSERM U1127, Sorbonne Université, Paris, France. [21]Present address: Laboratory of Sensory Processing, Brain Mind Institute, Faculty of Life Sciences, Ecole Polytechnique Fédérale de Lausanne (EPFL), Lausanne, Switzerland. [22]Present address: Boehringer Ingelheim Pharma GmbH & Co. KG; CNS Diseases, Biberach an der Riss, Germany. [23]Present address: University of Michigan, Ann Arbor, MI, USA. [24]Present address: MCTN, Medical Faculty Mannheim of the University of Heidelberg, Mannheim, Germany. ✉e-mail: jonas.wietek@gmail.com; ofer.yizhar@weizmann.ac.il

## Methods

### Animals

All experiments involving animals were carried out according to the guidelines stated in directive 2010/63/EU of the European Parliament on the protection of animals used for scientific purposes. Animal experiments at the Weizmann Institute of Science were approved by the Institutional Animal Care and Use Committee (IACUC) of the Weizmann Institute; experiments in Berlin were approved by the Berlin local authorities and the animal welfare committee of the Charité – Universitätsmedizin Berlin, Germany. Experiments in Bonn and Hamburg were performed in accordance with the guidelines of local authorities. Experiments in Tel Aviv were approved by the IACUC of Tel Aviv University (approval 01-19-037). Experiments performed at the University of Washington, Seattle, were approved by the IACUC of the University of Washington and conformed to US National Institutes of Health guidelines. For the locomotor behavior experiments, male and female mice (DAT-IRES-Cre; The Jackson Laboratory, strain 006660) were used. Mice were housed in groups, with 2–5 littermates of the same sex per cage. Cagemates underwent surgery on the same day and were assigned to the *Pd*CO or control group such that cages always included mixed groups. The control group included 13 mice (10 males and 3 females, 8–27 weeks old at the time of surgery). The *Pd*CO group included 14 mice (10 males and 4 females, 10–27 weeks old at the time of surgery). For in vivo pupillometry experiments, the control group included 5 mice (3 males and 2 females); the *Pd*CO group included 6 mice (5 males and 1 female); all DBH-Cre (B6.FVB(Cg)-Tg(Dbh-cre) KH212Gsat/Mmucd; 036778-UCD-HEMI) mice were aged 8–12 weeks at the time of surgery. For the cued reward delivery task experiment, adult (25–35 g), group-housed, 12- to 16-week-old *Pdyn-Cre* (PDYN-IRES-Cre, The Jackson Laboratory, strain 027958; 3 male and 3 female) or WT (The Jackson Laboratory, strain 000664; 2 male and 2 female) mice were used. Animals were kept in a sound-attenuated, isolated holding facility in the laboratory 1 week before surgery, after surgery and throughout the duration of the behavioral assays to minimize stress. All mice were kept at 22 ± 2 °C and 55% ± 10% room humidity in a 12-h light–dark cycle with access to food and water ad libitum. Mice were checked daily by animal caretakers.

### Molecular biology, DNA constructs and availability of reagents

Mammalian codon-optimized genes encoding optoGPCRs were synthesized (Twist Bioscience; except for *Lc*PPO, which was generously provided by P. Hegemann, Humboldt-Universität zu Berlin) and fused to a C-terminal rhodopsin 1D4 tag (TETSQVAPA). All genes were further subcloned in-frame with a C-terminal mScarlet into pcDNA3.1 vector under a CMV promoter or into a pAAV vector under the CaMKIIα minimal promotor (CaMKIIα 0.4). Expression vectors additionally contained the Kir2.1 membrane trafficking signal (KSRITSEGEYIPLDQIDINV) and Kir2.1 ER export signal (FCYENEV)[71], N-terminal and C-terminal to mScarlet, respectively, as previously reported for *As*OPN3 (ref. 13).

The following genes were used for expression (NCBI GenBank identifier; modifications if applied): *Gg*Opn5l1 (ref. 72; AB368181; modified N and C termini originating from *Xenopus tropicalis* OPN5m to improve expression as reported elsewhere[28] and further using C-terminal extension of the last 25 amino acids of *As*OPN3), *Bp*OPN[73] (AB050606.1), *Tr*PPO2 (ref. 74; AB626965), *Ol*TMT1A[75] (AGK24990; C terminus truncated by 63 amino acids to improve expression as reported elsewhere[29]), *Gg*OPN5m[72] (AB368182), *Hs*OPN5m[76] (AY377391; C terminus truncated by 37 amino acids to improve expression as reported elsewhere[77]), *Lc*PPO[23] (BAD13381), *Pd*CO[25] (AY692353), *Tr*PPO1 (ref. 74; AB626964), *As*OPN3 (ref. 22; BAN05625; C terminus truncated by 99 amino acids to improve expression as reported elsewhere[13,22]) and *Dr*PPO1 (ref. 74; AB626966). CAG-FR-GECO1c was a gift from R. Campbell (Addgene plasmid, 163682)[44], subcloned into a pAAV vector under the CaMKIIα 0.4 promotor. The pcDNA3.1-CMV-GIRK2.1 plasmid was a gift from E. Reuveny, Weizmann Institute of Science. GsX[27] and TRUPATH[30] plasmids were obtained from Addgene (GsX: 109373, 109375, 109373, 109360, 109359, 109357, 109356, 109355 and 109350; TRUPATH kit, 1000000163).

Further subcloning into other expression vectors and substitution of mScarlet by EGFP were performed by PCR and/or restriction enzyme-based cloning or the Gibson assembly method[78]. For rAAV packaging limitations, the standard woodchuck hepatitis posttranscriptional regulatory element (WPRE) and bovine growth hormone polyadenylation signal were replaced by a size-optimized expression cassette (miniWPRE, CW3SL)[79].

The following constructs are available from Addgene (ID): pcDNA3.1_CMV-PdCO-mScarlet (198507), pAAV_CaMKIIa(0.4)-PdCO-mScarlet-WPRE (198508), pAAV_hSyn-SIO-PdCO-mScarlet-WPRE (198509), pAAV_hSyn-PdCO-mScarlet-WPRE (198510), pAAV_hSyn-DIO-PdCO-mScarlet-WPRE (198511), pcDNA3.1_CMV-PdCO-EGFP (198512), pAAV_hSyn-PdCO-EGFP-WPRE (198513), pAAV_CaMKIIa(0.4)-PdCO-EGFP-WPRE (198514), pAAV_EF1a-fDIO-PdCO-mScarlet-miniWPRE (198515), pAAV_EF1a-DIO-PdCO-mScarlet-ER-miniWPRE (198516), pAAV_CaMKIIa(0.4)-stChrimsonR-EGFP-P2A-PdCO-WPRE (202198) and pAAV_hSyn-SIO-stChrimsonR-EGFP-P2A-PdCO-miniWPRE (202199) (all available at https://www.addgene.org/Ofer_Yizhar/). Other constructs are available from the authors upon request. rAAV2/5 hSyn-SIO-PdCO-mSacrlet-WPRE is available from the ETH Zürich viral vector core facility (ID: v816; https://vvf.ethz.ch/). rAAV2/9 encoding *Pd*CO-mScarlet are available from the Charité viral core facility (https://vcf.charite.de) as follows (ID): hSyn (BA-389), CamKIIa(0.4) (BA-388) and hSyn-DIO (BA-390).

### rAAV vector production

For production of rAAV particles, HEK293T cells were seeded at 30% ± 5% confluency and transfected 1 day after seeding with plasmids encoding AAV *rep*, *cap* genes of AAV2 and AAV1 or AAV9 and a vector plasmid encoding an rAAV cassette expressing the above-described optoGPCRs using the PEI method[80]. Seventy-two hours after transfection, cells were harvested and concentrated by centrifugation at 300g. The resulting cell pellet was resuspended in lysis solution: 150 mM NaCl, 50 mM Tris-HCl (pH 8.5 with NaOH). Cell lysis was performed by three freeze-thaw cycles and treated with 250 U ml$^{-1}$ lysate benzonase (Sigma) at 37 °C for 1.5 h to remove genomic and unpacked DNA, followed by centrifugation at 3,000g for 15 min. Crude virus used for transducing neuronal cultures was filtered with sterile 0.45-μm PVDF filters (Millex-HV, Merck). To produce purified rAAVs, the virus-containing supernatant (crude rAAV) was purified using heparin-agarose columns, eluted with 0.5 M NaCl and washed with PBS. The resulting viral suspension was concentrated with 100-kDa Ultra-15 centrifugal filters (Amicon), aliquoted and stored at −80 °C. Viral titers were quantified by real-time PCR using primers targeting the WPRE sequence: fw: CTTCCCGTATGGCTTTCATTTT, rv: CGGGCCACAACTCCTCATAA. Other high-titer rAAVs were produced at the ETH Zürich viral vector core facility (https://vvf.ethz.ch/): rAAV2/5 hSyn-SIO-PdCO-mSacrlet-WPRE (v816), hEF1α-DIO-mCherry-WPRE (v218) and hSyn1-DIO-mCherry-WPRE (v116).

### G-protein coupling assays using HEK cell culture

For initial testing of Gα signaling specificity, optoGPCR variants expressed in HEK293ΔG7 (lacking GNAS/GNAL/GNAQ/GNA11/GNA12/GNA13/GNAZ; A. Inoue, Tohoku University, Japan)[81] were tested using the GsX live cell assay[27]. In brief, cells were grown at 37 °C, 5% $CO_2$ in DMEM containing 4500 mg l$^{-1}$ glucose, L-glutamine (Sigma-Aldrich) with penicillin–streptomycin (100 U ml$^{-1}$) and 10% FBS. Cells were seeded ($2.5 \times 10^4$ cells per well) into poly-L-lysine (Sigma-Aldrich)-coated solid white 96-well plates (Greiner) and were co-transfected with different optoGPCR variants (pcDNA3.1, 50 ng per well) together with individual G-protein chimera (GsX, 2 ng per

well) and Glo22F luciferase (GloSensor, Promega, 100 ng per well) using Lipofectamine 2000 (Thermo Fisher). Cells were incubated for 24 h at 37 °C, at 5% $CO_2$ and, subsequently, in Leibovitz's L-15 medium (without phenol red, with L-glutamine, 1% FBS, penicillin–streptomycin 100 mg ml$^{-1}$), 9-*cis* retinal (10 µM) and beetle luciferin (2 mM in 10 mM HEPES pH 6.9) for 1 h at room temperature (RT). Cells were kept in the dark, and baseline luminescence was measured over a period of 200 s followed by optoGPCR activation using a 1-s light pulse (collimated CoolLED pE4000) of either 385 nm or 470 nm (for *Tr*PPO2, *Ol*TMT1A and *Bb*OPN). Changes in cAMP levels were measured over time using GloSensor luminescence with a Mithras luminometer (Berthold Technologies). For the assay quantification, each biological replicate was normalized to its pre-light baseline as well as to a non-optoGPCR control.

For the testing of optoGPCR-mediated inhibition of AC activity, HEK293ΔG7 cells were seeded and transfected as described above with (plasmid amount per well: 100 ng GloSensor, 2 ng $G\alpha_s$ and 50 ng optoGPCR). Twenty-four hours after transfection, the medium was changed to PBS (with $Ca^{2+}$ and $Mg^{2+}$), supplemented with 9-*cis* retinal (10 µM) and beetle luciferin (2 mM in 10 mM HEPES pH 6.9). After 1 h of incubation at RT, the baseline bioluminescence was measured, followed by application of forskolin (500 µM final concentration). After 20 min, optoGPCRs were activated with a 2-s light pulse (365 nm), and bioluminescence measurements were continued for another 60 min. Bioluminescence signals were normalized to the average bioluminescence prior forskolin application.

For more detailed Gα-protein profiling, the TRUPATH assay[30] was used and HEK293ΔG$_7$ cells were seeded as described above, co-transfected with RLuc8-Gα, Gβ, Gγ-GFP2 and optoGPCRs in a 1:1:1:1 ratio (100 ng per well total DNA) using Lipofectamine 2000. Cells were incubated for 24 h at 37 °C, at 5% $CO_2$ and, subsequently, in Leibovitz's L-15 medium (without phenol red, with L-glutamine, 1% FBS, penicillin–streptomycin 100 mg ml$^{-1}$) and 9-*cis* retinal (10 µM) and kept in the dark. For performing BRET assays, the medium was changed to HBBS, supplemented with 20 mM HEPES and 10 µM 9-*cis*-retinal + 5 µM Coelenterazine 400a, and incubated for 5 min at RT. optoGPCRs were activated using a 1-s, 385-nm light pulse (collimated CoolLED pE4000). BRET ratio changes were determined from RLuc8-Gα and Gγ-GFP2 signals, integrated over 3 min, directly after light application and 17–20 min after optoGPCR activation. All bioluminescence data were acquired using MikroWin2010 (Mikrotek Laborsysteme).

## optoGPCR-mediated GIRK current recordings from HEK cells

For the initial comparison of optoGPCR-evoked GIRK currents, optoGPCRs were transiently expressed in HEK293 cells stably expressing GIRK1/GIRK2 subunits (kindly provided by A. Tinker, Queen Mary's School of Medicine and Dentistry). Briefly, cells were maintained at 37 °C and 5% $CO_2$ in high-glucose DMEM supplemented with Geneticin (G418, Gibco), 10% FBS (Biological Industries) and penicillin–streptomycin (100 U ml$^{-1}$) and seeded onto poly-D-lysine-coated coverslips in 24-well plates (Corning) and were additionally supplemented with 1 µM 9-*cis* retinal (Sigma). One day after seeding, pcDNA3.1-CMV-optoGPCR-mScarlet plasmids were transiently transfected using FuGENE HD (Promega; 0.75 µl per well, plasmid DNA, 250 ng per well) in serum-free DMEM (50 µl per well).

Currents from HEK293 cells stably expressing GIRK were recorded under visual guidance using a SliceScope II (Scientifica) with an Olympus LUMPlanFL N ×40/0.80 W objective under IR-DIC. A Lumencor SpectraX light engine was used to identify expressing cells via mScarlet fluorescence and for light application to toggle optoGPCR activation. In the case of non-switchable or slow-cycling optoGPCRs (*As*OPN3, *Ol*TMT1A and *Gg*OPN5l1), expressing cells were identified first and patched only after an additional 25 min in darkness. HEK cells were perfused with extracellular solution: 20 mM NaCl, 120 mM KCl, 2 mM $CaCl_2$, 1 mM $MgCl_2$, 10 mM HEPES, pH 7.3 (KOH), 320 mOsm (with D-glucose).

Glass microelectrodes (1.5–2.5 MΩ) were pulled from thin-walled glass capillaries and filled with 5 mM NaCl, 40 mM KCl, 2 mM $MgCl_2$, 10 mM HEPES, 100 mM κ-aspartate, 5 mM MgATP, 0.1 mM $Na_2GTP$ and 2 mM EGTA, with pH 7.3 (KOH) and 300 mOsm (with D-glucose). GIRK currents were recorded in whole-cell voltage-clamp mode at a holding potential of −50 mV. A Multiclamp 700B amplifier and Digidata 1440A digitizer were used to control and acquire electrophysiological recordings using Clampex 10.7 (all Molecular Devices) at 10 kHz and filtered at 3 kHz.

The different optoGPCRs were activated with a 500-ms (5 s in case of *Gg*OPN5l1) light pulse close to their reported activation maximum with 10-nm narrow bandpass filters (Edmund Optics). Light intensities for each wavelength were calibrated to the same photon flux corresponding to 0.92 mW mm$^{-2}$ at 520 nm. The center wavelengths used were 520 nm, *As*OPN3; 450 nm, *Ol*TMT1A; 473 nm, *Tr*PPO2; 450 nm, *Bb*OPN; 546 nm, *Gg*OPN5l1; and 394 nm for all other optoGPCRs. Light-induced recovery was induced by application of a 10-s 568 ± 10-nm light pulse. Experiments were performed at 22 ± 1 °C. Maximum GIRK current amplitudes were determined using Clampfit 10.7 (Molecular Devices). Light intensities were measured with a calibrated S170C power sensor (Thorlabs).

For two-photon activation of *Pd*CO, electrophysiological recordings were performed on HEK293T cells (HEK293T/17, American Type Culture Collection, CRL-1573) as described previously[14]. In brief, pcDNA3.1-CMV-PdCO-mScarlet was co-transfected (in a 1:3 ratio) together with pCAG-GIRK2/1-myc[82] using Lipofectamine 2000 (Invitrogen) according to the manufacturer's instructions and supplemented with 1 µM 9-*cis* retinal (Sigma). GIRK currents evoked by activation of *Pd*CO were recorded under visual guidance using a Fluoview FVMPE-RS multiphoton imaging system using an XLPLN25XWMP2 objective (both Olympus). The extracellular solution contained 140 mM NaCl, 20 mM KCl, 0.5 mM $CaCl_2$, 2 mM $MgCl_2$, 10 mM glucose, 10 mM HEPES, pH 7.3 with NaOH and 315 ± 5 mOsm. Cells were patched with microelectrodes pulled from thin-walled glass capillaries (1–5 MΩ) and filled with 120 mM potassium gluconate, 5 mM NaCl, 0.1 mM $CaCl_2$, 2 mM $MgCl_2$, 1.1 mM EGTA, 10 mM HEPES, 4 mM $Na_2ATP$, 0.4 mM $Na_2GTP$ and 15 mM $Na_2$-phosphocreatine, with pH 7.28 and 290 mOsm. Whole-cell voltage-clamp current recordings in response to 200-ms ramp depolarizations from −120 to 40 mV every 2 s with a holding potential of −40mV were amplified, digitized (20 kHz) using a HEKA EPC10 (filtered at 3 kHz) and recorded using Patchmaster software (HEKA). Whole-cell and pipette capacitance transients were minimized, and series resistance was compensated by 70%.

Two-photon excitation for *Pd*CO was carried out using MaiTai and Insight tunable titanium/sapphire lasers (Spectra Physics). For photostimulation, cells were centered in a pixel square of 90 × 90 nm (0.4792 mm per pixel) and scanned at a speed of 8 µs per pixel. The spectral characterization was performed using a 20-s two-photon stimulation at 700 nm, 800 nm, 900 nm, 1,000 nm or 1,100 nm at 3-mW laser power. For each cell, three to five different wavelengths were applied randomly. For a dose–response titration at 800 nm, a 10-s stimulation was performed across 0.1 mW, 0.3 mW, 1 mW, 3 mW and 10 mW from low to high intensity. Two-photon intensities were calibrated at the sample focal plane using a thermal power sensor (S175C, Thorlabs) and power meter (PM100D, Thorlabs). Data were analyzed using IgorPro (WaveMetrics) and NeuroMatic[83] using custom macros. For each voltage ramp, the GIRK-mediated inward current was averaged over 5 ms at −120 mV holding potential.

## Primary dissociated hippocampal neuron culture and gene delivery

Primary cultured hippocampal neurons were isolated from CA1 and CA3 hippocampal regions of postnatal day (P) 0 Sprague-Dawley rat pups of either sex (Envigo). Neurons were digested with 0.4 mg ml$^{-1}$ papain (Worthington) and seeded on Matrigel (1:30 dilution;

Corning)-coated glass coverslips in 24-well plates at a density of 65,000 cells per well. Neurons were maintained in a 5% $CO_2$ humidified incubator in Neurobasal-A medium (Invitrogen) supplemented with 1.25% FBS, 4% B27 supplement (Gibco) and 2 mM Glutamax (Gibco). For inhibition of glial overgrowth, 200 mM fluorodeoxyuridine (Sigma) was added at day in vitro (DIV) 4.

For confocal imaging or initial electrophysiological recordings of opsin-expressing cultured primary neurons, opsin and cell-filling plasmids encoding EYFP or GIRK2.1 were co-transfected at DIV 5 using a modified $Ca^{2+}$-phosphate method[84]. Briefly, the neuronal cultured medium of a 24-well plate was collected and replaced with 400 ml serum-free MEM (Thermo Fisher Scientific). Then, 30 µl transfection mix (2 µg plasmid DNA and 250 mM $CaCl_2$ in HBS at pH 7.05) was added per well, and cells were incubated for 1 h to allow for transfection. Neurons were washed twice with MEM, and the medium was changed back to the collected original medium. Cultured neurons were used between DIVs 14 and 17 for experiments.

For $Ca^{2+}$ imaging experiments, cultured neurons were co-transduced with rAAV2/1.hSyn-PdCO-EGFP-WPRE and rAAV2/1.CaMKIIα(0.4)-FR-GECO1c-WPRE at DIV 1. Experiments were carried out between DIVs 14 and 21.

Autaptic primary hippocampal neuronal cultures on glial cell micro-islands were prepared from P0 mice (C57BL/6NHsd; Envigo, 044) of either sex as previously described[85]. First, 300-µm-diameter spots of growth-permissive substrate consisting of 0.7 mg ml$^{-1}$ collagen and 0.1 mg ml$^{-1}$ poly-D-lysine were applied with a custom-made stamp on agarose-coated coverslips. Second, astrocytes were seeded and were allowed to proliferate in DMEM supplemented with 10% FCS and 0.2% penicillin–streptomycin (Invitrogen) for 1 week to form glia micro-islands. Third, after changing the medium to Neurobasal-A supplemented with 2% B27 and 0.2% penicillin–streptomycin, hippocampal neurons prepared from P0 mice were added at a density of 370 cells cm$^{-2}$. Neurons were transduced with rAAVs ($1.5 \times 10^8$ viral genomes (vg) per well, matched by titer for all constructs) at DIV 1 and were recorded between DIVs 14 and 21.

### Hippocampal organotypic slice culture and gene delivery
Organotypic hippocampal slices were prepared from Wistar rats at P5–7 as described[86]. Briefly, dissected hippocampi were cut into 400-µm slices with a tissue chopper and placed on a porous membrane (Millicell CM, Millipore). Cultures were maintained at 37 °C, 5% $CO_2$ in a medium containing 80% MEM (Sigma, M7278), 20% heat-inactivated horse serum (Sigma, H1138) supplemented with 1 mM L-glutamine, 0.00125% ascorbic acid, 0.01 mg ml$^{-1}$ insulin, 1.44 mM $CaCl_2$, 2 mM $MgSO_4$ and 13 mM D-glucose. No antibiotics were added to the culture medium.

Transgene delivery in individual CA3 pyramidal cells was performed by single-cell electroporation between DIVs 15 and 20 as previously described[87]. The plasmids pAAV-CaMKIIα(0.4)-PdCO-mScarlet and pAAV-CaMKIIα(0.4)-LcPPO-mScarlet were each diluted to 50 ng ml$^{-1}$ in potassium gluconate-based solution consisting of 135 mM potassium gluconate, 10 mM HEPES, 0.2 mM EGTA, 4 mM $Na_2$ATP, 0.4 mM Na-GTP, 4 mM $MgCl_2$, 3 mM ascorbate, 10 mM $Na_2$-phosphocreatine, pH 7.2 and 295 mOsm kg$^{-1}$. An Axoporator 800A (Molecular Devices) was used to deliver 25 hyperpolarizing pulses (−12 V, 0.5 ms) at 50 Hz. During electroporation, slices were maintained in pre-warmed (37 °C) HEPES-buffered solution consisting of 145 mM NaCl, 10 mM HEPES, 25 mM D-glucose, 2.5 mM KCl, 1 mM $MgCl_2$ and 2 mM $CaCl_2$ (pH 7.4, sterile filtered).

Targeted viral vector-based transduction of organotypic hippocampal slice cultures[88] was performed by pressure injecting (20 PSI/2–2.5 bar, 50-ms duration) rAAV particles encoding AAV2/9.CaMKIIα(0.4)-PdCO-mScarlet or AAV2/9.CaMKIIα(0.4)-LcPPO-mScarlet and AAV2/9.hSyn-somBiPOLES-mCerulean using a Picospritzer III (Parker) under visual control (oblique illumination) into the stratum pyramidale of CA3 between DIVs 2 and 5. Slice cultures

were then maintained in the incubator for 2–3 weeks allowing for virus payload expression.

### Confocal imaging, quantification membrane targeting and expression levels
Primary cultured hippocampal neurons (transfected as described above) coexpressing the different opsins (pAAV-CaMKIIα(0.4)-opsin-mScarlet) together with a cell-filling EYFP (pAAV-CaMKIIα(0.4)-EYFP) were fixed and permeabilized 4 days after transfection using 4% paraformaldehyde (PFA) for 15 min, washed three times with PBS and stained for 3 min with DAPI (5 mg ml$^{-1}$ solution diluted at a 1:30,000 ratio). Coverslips were mounted using PVA-DABCO (Sigma), and fluorescence images were acquired using Zeiss Zen 3.7 software on a Zeiss LSM 700 confocal microscope equipped with a Plan-Apochromat ×63/1.40 Oil DIC objective (all Carl Zeiss).

For quantification of opsin expression in the membrane and cytosol, respectively, binary masks for EYFP and mScarlet signals were generated using fixed thresholding in ImageJ[89] on a single equatorial z-slice per expressing neuron, identified visually with help of the nuclear DAPI stain. Expression analysis was restricted to the somatodendritic region by manual selection. The EYFP mask was subtracted from the mScarlet mask to generate a mask that restricts the analysis to the membrane only. Subsequently, the average pixel intensity was measured for the defined regions of interest. The expression index was calculated by subtraction of the whole-cell mScarlet signal by the EYFP signal, divided by the sum of both signals. Confocal images of brain sections from pupillometry experiments were acquired with an SP5 laser scanning confocal microscope (Leica) and a ×10 air/0.4 NA objective. Confocal images of brain sections from the cued reward delivery experiments were acquired with an FV3000 laser scanning microscope (Olympus IMS) using a ×20 objective.

### In vitro electrophysiology on neuronal samples
Qualitative measurements of optoGPCR functionality using primary cultured neurons coexpressing pAAV-CaMKIIα(0.4)-optoGPCR-mScarlet together with pcDNA3.1-CMV-GIRK2.1 transfected as described above were performed using the same setup as described for measurements on stably expressing GIRK2/1 HEK293 cells. Neurons were patched with microelectrodes (3.0–4.5 mΩ), filled with 2 mM NaCl, 4 mM KCl, 10 mM HEPES, 135 mM potassium gluconate, 4 mM $Na_2$ATP, 4 mM EGTA and 0.3 mM $Na_2$GTP, with 290 mOsm and pH adjusted to 7.3 with KOH. Electrophysiological recordings were obtained under continuous perfusion in Tyrode's medium: 150 mM NaCl, 4 mM KCl, 2 mM $MgCl_2$, 2 mM $CaCl_2$, 10 mM D-glucose, 10 mM HEPES; 320 mOsm; pH adjusted to 7.35 with NaOH.

EPSCs from autaptic primary neurons were recorded under visual guidance using an Olympus IX51 inverted microscope with an Olympus UPlanSApo ×20/0.75 UIS2 objective under far infrared light (>665 nm) widefield illumination. A CoolLED P4000 served as a light source to identify expressing cells and for light application to toggle optoGPCR activation. In the case of non-switchable optoGPCRs (AsOPN3 and OlTMT1A), electrophysiological recordings were performed first, and cells were investigated for expression after recordings. Acquired data were excluded in case cells were not expressing. Autaptic neurons were constantly perfused with extracellular solution: 140 mM NaCl, 2.4 mM KCl, 10 mM HEPES, 10 mM D-glucose, 2 mM $CaCl_2$ and 4 mM $MgCl_2$ (pH adjusted to 7.3 with NaOH, 300 mOsm). Cells were patched with microelectrodes pulled from quartz glass capillaries (3–4 mΩ), filled with 136 mM KCl, 17.8 mM HEPES, 1 mM EGTA, 0.6 mM $MgCl_2$, 4 mM MgATP, 0.3 mM $Na_2$GTP, 12 mM $Na_2$-phosphocreatine and 50 U ml$^{-1}$ phosphocreatine kinase (300 mOsm), with pH adjusted to 7.3 with KOH. A Multiclamp 700B (Molecular Devices) amplifier and NI USB-6343 digitizer (National Instruments) were used to control and acquire electrophysiological recordings and the application of light stimulation via WinWCP 5.7 software (https://github.com/johndempster/WinWCPXE/). Data

were acquired at 10 kHz and filtered at 3 kHz. Cells were kept at −70 mV, and series resistance and capacitance were compensated by 70%. EPSCs were elicited by a 1-ms depolarization to 0 mV (50-ms interstimulus interval, every 5 s) resulting in an unclamped axonal AP causing neurotransmitter release. SCH23390 (Tocris) was locally applied with a perfusion system (AutoMate Scientific ValveLink8.2). Pertussis toxin (0.5 mg ml$^{-1}$) was applied to the cultures 24 h before the recordings. For the initial comparison of EPSC reduction efficacy, the different optoGPCRs were activated with a 0.5-s, 390 ± 10-nm light pulse (FB390-10, Thorlabs), and potential recovery was induced using a 4.5-s, 560 ± 10-nm pulse (FB560-10, Thorlabs). For spectral sensitivity measurements, light from the CoolLED P4000 was filtered with narrow bandpass filters mounted on a FW212C filter wheel (Thorlabs). The following filters were used (center wavelength ± 10 nm; Edmund Optics catalog no.): 365 nm (65-069), 405 nm (65-072), 436 nm (65-077), 470 nm (65-083), 492 nm (65-087), 520 nm (65-093), 560 nm (87-887), 594 nm (86-733), 636 nm (65-106) and 660 nm (86-086). Light intensities for each wavelength were calibrated to the same photon flux corresponding to 1.1 mW mm$^{-2}$ at 520 nm. Light intensities were measured with a calibrated S130VC power sensor (Thorlabs). Spectral measurements were performed alternating from UV to red wavelengths or vice versa. Light titration experiments were performed from low to high light intensity or light pulse duration. Experiments were performed at RT.

Electrophysiological recordings in organotypic hippocampal slice cultures were performed using a BX51WI microscope (Olympus) equipped with a Multiclamp 700B amplifier (Molecular Devices) controlled by Ephus[90] R220 that was used for data acquisition. Alternatively, a second BX51WI microscope (Olympus) equipped with a Double IPA integrated patch amplifier controlled by SutterPatch software (Sutter Instrument) was used for electrophysiological measurements. Patch pipettes with a tip resistance of 3–5 MΩ were filled with 135 mM potassium gluconate, 4 mM MgCl$_2$, 4 mM Na$_2$ATP, 0.4 mM Na-GTP, 10 mM Na$_2$-phosphocreatine, 3 mM ascorbate, 0.2 mM EGTA and 10 mM HEPES (pH 7.2). Artificial cerebrospinal fluid consisted of 135 mM NaCl, 2.5 mM KCl, 4 mM CaCl$_2$, 4 mM MgCl$_2$, 10 mM Na-HEPES, 12.5 mM D-glucose and 1.25 mM NaH$_2$PO$_4$ (pH 7.4). All experiments were performed at RT (21–23 °C) except for the extracellular field stimulation experiments, which were performed at 33 ± 1 °C. For experiments measuring GIRK currents, synaptic blockers D-CPP-ene (10 μM), NBQX (10 μM) and picrotoxin (100 μM; Tocris) were added to the HEPES-buffered artificial cerebrospinal fluid, and patched optoGPCR-expressing CA3 neurons were held at −70 mV during the measurements. In synaptic stimulation experiments (optogenetic and electrical), postsynaptic non-transfected CA1 neurons were held at −60 mV while recording PSCs in voltage-clamp mode. Access resistance of the recorded CA1 neurons was continuously monitored, and recordings above 20 MΩ and/or with a drift >30% were discarded. A 16-channel pE4000 LED light engine (CoolLED) was used for optogenetic stimulation of the optoGPCRs. Light intensity was measured in the object plane with a 1918-R power meter equipped with a calibrated 818-ST2-UV/DB detector (Newport) and divided by the illuminated field of the Olympus LUMPLFLN 60XW objective (0.134 mm$^2$) or of the Olympus LUMPLFLN 40XW objective (0.322 mm$^2$). For presynaptic somBiPOLES stimulation, we used a fiber-coupled LED (400-μm fiber, NA 0.39, M118L02, Thorlabs) to deliver 5-ms red light pulses at 625 nm.

In extracellular electrical stimulation experiments, afferent Schaffer collateral axons were stimulated (0.2 ms, 20–70 μA every 10 s) with a monopolar glass electrode connected to a stimulus isolator (IS4 stimulator, Scientific Devices).

### One-photon spectral multiplexing in cultured neurons

For calcium imaging experiments paired with transient *Pd*CO activation, hippocampal neuronal cultures were co-transduced with rAAV2/1.hSyn-PdCO-EGFP and rAAV2/1.CaMKIIα(0.4)-FR-GECO1c as described above, and experiments were carried out between DIVs 14 and 21.

Imaging was performed with SliceScope II (Scientifica) using an Olympus LUMPlanFL N ×40/0.80 W objective, and data were acquired with an ORCA-Flash4.0 digital camera (C11440, Hamamatsu). Data acquisition was carried out with 4 × 4 binning, 80-ms exposure at 10 Hz and FR-GECO1c was excited at 586 ± 10 nm with an irradiance of 0.52 mW mm$^{-2}$. To transiently activate *Pd*CO, 445 ± 10-nm light (6.37 mW mm$^{-2}$) was applied at 10 Hz, 5-ms pulse width between the imaging intervals. Cells were constantly perfused with extracellular solution: 140 mM NaCl, 2.4 mM KCl, 10 mM HEPES, 10 mM D-glucose, 2 mM CaCl$_2$ and 4 mM MgCl$_2$ (pH adjusted to 7.3 with NaOH, 300 mOsm). To evoke AP-induced calcium transients, cells were patched in the tight-seal, cell-attached configuration as described by Perkins[91]. Briefly, cells were patched with low-resistance microelectrodes pulled from glass capillaries (1.5–2.5 mΩ), filled with 145 mM NaCl. A train of APs was evoked by a 40-ms square current injection of 0.6 to 1.0 μA at a frequency of 0.2 Hz through the intact membrane patch (>3 GΩ seal resistance). The change in the FR-GECO1c signal $\Delta F/F_0$ (where $F_0$ is the median fluorescence signal and $\Delta F = F(t) - F_0$; $F(t)$ is the fluorescence value at a given time) was calculated in ImageJ[89] and corrected for blue light-induced increase in the fluorescence signal (Extended Data Fig. 9). A Lumencor SpectraX light engine served as light source, and light intensities were measured with a calibrated S170C power sensor (Thorlabs). The spectra of light used to excite *Pd*CO and FR-GECO1c were measured with an Ocean QE pro spectrometer (Ocean Optics).

Experiments using CaMKIIα(0.4)-stChrimsonR-EGFP-P2A-PdCO-WPRE were performed on the setup described above using the same extracellular solution. Neurons were patched at −70 mV with microelectrodes (3.0–4.5 mΩ), filled with 2 mM NaCl, 4 mM KCl, 10 mM HEPES, 135 mM potassium gluconate, 4 mM Na$_2$ATP, 4 mM EGTA and 0.3 mM Na$_2$GTP, with 290 mOsm and pH adjusted to 7.3 with KOH. For action spectra recordings, light from the Lumencor SpectraX light engine was filtered with narrow bandpass filters mounted on a FW212C filter wheel (Thorlabs). The following filters were used (center wavelength ± 10 nm, Edmund Optics catalog no.): 394 nm (65-070), 422 nm (34-496), 450 nm (65-079), 473 nm (34-502), 500 nm (65-088), 520 nm (65-093), 546 nm (65-097), 568 nm (65-099), 594 nm (86-733), 620 nm (65-104) and 647 nm (65-108). The light intensity for each wavelength was calibrated to the same photon flux corresponding to 0.92 mW mm$^{-2}$ at 520 nm. Short light pulses were applied (2 ms) and photocurrents measured under TTX (1 μM), CNQX (20 μM) and AP-5 (50 μM), recorded in both directions per expressing cell (red to blue and blue to red) and then averaged. For postsynaptic recording in non-expressing cells, neurons were measured with 10 μM SCH23390 (10 μM) to eliminate the potential contribution of GIRK currents. The light from the Lumencor SpectraX light engine was filtered with bandpass filters and the following light intensities were used: 390 ± 9 nm (0.155 mW mm$^{-2}$), 512 ± 12 nm (0.327 mW mm$^{-2}$) and 632 ± 11 nm (7.015 mW mm$^{-2}$). A Multiclamp 700B amplifier and Digidata 1440A digitizer (both Molecular Devices) were used to control and acquire electrophysiological recordings. Data were acquired at 10 kHz and filtered at 3 kHz. Light intensities were measured with a calibrated S170C power sensor (Thorlabs). Electrophysiological data were recorded using Clampex 10.7, while imaging data were acquired using Micro-Manager (2.0)[92].

### Surgeries

DAT-Cre transgenic mice ($n = 27$) were anesthetized (with 2.5% isoflurane by volume for induction and 1–2% for maintenance) and placed in a stereotaxic frame. Surgery was performed under aseptic conditions. rAAV vectors encoding a Cre-dependent *Pd*CO-mScarlet transgene (rAAV2/1.hSyn-SIO-PdCO-mScarlet-WPRE; 3.8 × 10$^{12}$ vg per ml) or eYFP (rAAV2/1.EF1a-DIO-eYFP-WPRE; 2 × 10$^{13}$ vg per ml) were bilaterally injected into the substantia nigra (coordinates respective to bregma: anteroposterior (A-P), −3.5 mm; mediolateral (ML), ±1.4 mm; dorsoventral (DV), −4.25 mm; 500 nl per site, at a rate of 100 nl min$^{-1}$). Optical fibers (200-μm diameter, NA 0.5) were bilaterally implanted above

the dorsomedial striatum (A-P, +0.6 mm; ML, ±1.5 mm; DV, −2.1 mm). C&B Metabond cement and dental acrylic were used to fix the fibers to the skull. Mice were allowed to recover for 6–9 weeks to allow for viral expression. Analgesia was administered before the surgery and for at least 3 days following recovery.

For pupillometry experiments, mice ($n = 6$ $Pd$CO, $n = 5$ mCherry control) were anesthetized (with 3% isoflurane by volume for induction and 1.3% for maintenance) and placed in a stereotaxic frame. Surgery was performed under aseptic conditions. Under microscopic control, a craniotomy was made using a high-speed surgical drill. Two surgeries were performed on each mouse. In the first surgery, the viruses ($Pd$CO: rAAV2/5.hSyn-SIO-PdCO-mScarlet-WPRE or rAAV2/1.hSyn-SIO-PdCO-mScarlet-WPRE; $6.1 × 10^{12}$ or $3.8 × 10^{12}$ vg per ml, mCherry controls: rAAV2/5.hSyn1-DIO-mCherry-WPRE or hEF1α-DIO-mCherry-WPRE; $1.3 × 10^{13}$ or $7.8 × 10^{12}$ vg per ml) were injected into the LC using a UMP3 Microsyringe Injector, a Micro4 Controller pump and 33-gauge NanoFil needles (World Precision Instruments) unilaterally at a rate of 100 nl min$^{-1}$ for 8 min (two injections of 250 nl for a total volume of 500 nl). Coordinates used to target LC were (respective to bregma) A-P, −5.4 mm; ML, 1 mm; DV, −3.3, −3.5 mm relative to the brain surface. In the second surgery, optic fibers (MFC_200/240- 0.22_10 mm_MF1.25_FLT, Doric Lenses) were implanted above the BF (A-P, 0.7 mm; ML, 2.25 mm at 7.5°; DV, −4.75 mm from the brain surface) and above the EW (A-P, −3.8 mm; ML, 1.3 mm at 20°; DV, −3.2 mm relative to brain surface). In one animal, the caudate putamen was targeted instead of the BF (A-P, −1.1 mm; ML, 1.35 mm; DV, −4 mm). Dental acrylic was gently placed around the optic fibers, fixing them to the skull.

For the cue reward task, mice ($n = 6$ for $Pdyn$-Cre, 3 males and 3 females; $n = 4$ for WT, 2 males and 2 females) were anesthetized in an induction chamber (2–4% isoflurane) and placed into a stereotaxic frame (Kopf Instruments, 1900) where they were maintained at 1–2% isoflurane. Male and female mice were anesthetized, following which we performed a craniotomy and bilaterally injected, using a blunt neural syringe (65457-01, Hamilton Company), 400 nl of rAAV2/1.hSyn-SIO-PdCO-mScarlet-WPRE (viral titer $3 × 10^{12}$ vg per ml) into the NAc (stereotaxic coordinates from bregma: A-P, 1.0 mm; ML, ± 1.0 mm; DV, −4.5 to 4.0 mm), followed by bilateral fiber optic implantation 5 weeks later into the VTA (stereotaxic coordinates from bregma: A-P, −3.4 mm; ML, ± 1.5 mm; DV, −4.5 mm; 15° angle). Implants were secured using two bone screws and a dental cement head cap (Lang Dental).

## Pupillometry experiments

Mice were anesthetized with (0.7–1% isoflurane) and placed in a stereotaxic frame. Optic patch cords connected the implanted optic fiber ferrules to a 447-nm laser (Changchun New Industries Optoelectronics Technology). Pupillometry was performed by illuminating the eyes with infrared light (VGAC) and by continuous video monitoring (ten frames per second) synchronized with laser stimulation. We examined the effects of different stimulation frequencies (1, 5, 10, 20 and 40 Hz) in two separate locations (EW or forebrain) on ipsilateral or contralateral pupil size (relative to injection and fiber). Each stimulation trial was 10 s long, with 20-ms pulses and a long 2-min intertrial interval to allow pupils to return to baseline. Trial order was randomized for stimulation frequency and stimulation location within each experiment. Pupil video recordings were captured using USB web cameras (Logitech C615), with IR filter removed, connected to a TDT RZ2 processor (Tucker-Davis Technologies). The RZ2 was programmed through the TDT Synapse software and allowed automated control of laser activation through a MATLAB code, and synchronization with the camera feeds. Data analysis (quantification of pupil area in video frames) was performed as in previous studies using custom MATLAB code[93]. Briefly, we fit a circle to the pupil area in the video images and, for each trial separately, normalized pupil area relative to the 4-s pretrial baseline to compute the percentage change dynamics in the −5 to 30-s interval

around laser stimulation. Trials were then averaged for each animal, eye and condition separately (~8 trials per condition) to generate time courses as seen in Fig. 6j. Maximal pupil constriction (through in time course) was defined as the minimal value in this pupil time course.

## In vivo optogenetic silencing of the nigrostriatal pathway

Following recovery, mice underwent a single 15-min habituation session, to habituate to handling, bilateral patch cord attachment and the open field arena. In experimental sessions, we attached individual mice to a patch cord unilaterally to measure $Pd$CO-induced bias in locomotion. We recorded the free locomotion in an open field arena (50 × 50 × 50 cm) continuously for 30 min under near-infrared illumination. After a 10-min baseline no-light period ('baseline'), we delivered 500-ms light pulses (400 nm, 10 mW at the fiber tip), at 0.4 Hz for 10 min ('$Pd$CO activation'), to the right or left sides (on separate sessions). Then, we administered five light pulses (528 nm, 4-s duration at 0.1 Hz, 10 mW at the fiber tip; '$Pd$CO recovery') followed by no-light administration for the rest of the 10-min recovery period. The behavior was video recorded using IC Capture at 30 fps. (672 × 672 pixels). A MASTER-8 pulse generator (A.M.P.I.) triggered the activation of both wavelengths, delivered by the STSI-Optogenetics-LED-Violet-Green system (Prizmatix). The two LEDs had the following wavelength specifications: 400 nm/15 nm and 532 nm/38 nm (peak wavelength/full width at half maximum, respectively). A near-infrared LED placed at the video region of interest was used as a synchronization signal (100-ms pulse duration, 0.1 Hz, for the entire 30-min session). These digital signals were split and recorded using Open Ephys acquisition board and used to synchronize the video and LED operation. Offline video processing and mouse tracking were done using DeepLabCut[94]. Briefly, we trained DeepLabCut to detect seven features on the mouse body (nose, head center, left and right ears, left and right hips, tail base), the four corners of the arena and the ON or OFF states and coordinates of the video synchronization LED. $X–Y$ coordinates of each feature were then further processed to complete missing or noisy values (large and fast changes in $X$ or $Y$) using linear interpolation (interp1) of data from neighboring frames. This was followed by a low pass filtering of the signals (malowess, with 50-point span and of linear order). Finally, a pixel-to-centimeter conversion was done based on the video-detected arena features and its physical measurements. A linear fit to the nose, head, center and tail features defined the mouse angle with respect to the south arena wall at each frame. Following its dynamics over the session, we identified direction shifts as a direction change in angle that exceeds 20° and 1 s. To achieve a comparable measurement between right- and left-hemisphere sessions, we measured motion in the ipsilateral direction as positive and contralateral motion as negative from the cumulative track of angle. The net angle gain was calculated as the sum of ipsilateral and contralateral angles gained over each time bin (1-min or 10-min bins as indicated). Results from the left- and right-hemisphere sessions of each mouse were averaged and then used for statistical comparison between the $Pd$CO and control groups.

## Cued reward delivery and photoinhibition of dynorphin terminals

All tests took place in mouse operant chambers (17.8 cm × 15.2 cm × 18.4 cm; Med Associates). A rotating optical commutator (Doric) was located on the top of the operant chamber and connected to a 465-nm diode-pumped solid-state laser (OEM Laser Systems). Fibers were connected to the implants on the mouse for every training session. Sucrose pellets were delivered at a variable intertrial interval (VI, 90 s), preceded by a 5-s houselight cue, and animal head entries were recorded by outfitting the pellet receptacles with custom-made infrared sensors to record beam breaks during head entries. Upon training, animals were given free access to 0.25 g of sucrose for half an hour and underwent counterbalanced sessions with laser on (1, 20 or 40 Hz) versus off, where laser was delivered for 10 s at the start of the houselight cue. Laser power was

adjusted to obtain ~5 mW transmittance into the brain, and we used 20-ms pulse width for laser delivery at all frequencies.

## Histology

Following all pupil experiments, mice underwent deep isoflurane anesthesia (4%) combined with a ketamine–xylazine dose (100 mg per kg body weight ketamine, 1.33 mg per kg body weight xylazine) and were perfused intracardially with saline (0.9% NaCl; 1 ml per gram body weight) followed by 4% PFA (Merck). Brains were then extracted and fixed for 24–48 h in 4% PFA. Coronal brain sections were cut using a Leica VT1000 S vibrating blade microtome at 60 μm. Sections were either kept free floating in PBS for immunohistochemistry or mounted on glass slides and examined under brightfield microscopy to verify fiber placement.

The viral expression was evaluated histologically by double staining of free-floating sections. To this end, following all experiments, sections were washed three times in PBS (Hylabs) and then permeabilized in PBST (PBS containing 0.1% Triton X-100 (Merck)). Brain sections were blocked in PBST containing 20% NGS (Vector Laboratories) for 1 h at RT and incubated with primary antibodies in PBST (containing 2% NGS) at 4 °C for 24–36 h. Primary antibodies were against norepinephrine transporter (mouse anti-NET, 1:300 dilution, NET05-2 MAb Technologies), or against red fluorescent protein (guinea pig anti-RFP, 1:500 dilution, 390004 SYSY). After three washes in PBS, sections were incubated with secondary antibodies conjugated to fluorophores (AF488 goat anti-mouse, 1:500 dilution, ab150117, Abcam; donkey anti-guinea pig CyTM5, 1:300 dilution, 706-175-148, Jackson ImmunoResearch) in PBST containing 2% NGS for 1.5 h at RT. After three washes in PBST and once in PBS, sections were mounted onto glass slides and cover-slipped with aqueous mounting medium (Thermo Scientific, 9990412).

At the end of the cued reward delivery experiments, animals were perfused with 4% PFA followed by anatomical analysis for histology and placement validation. Brains were dissected and post-fixed in 4% PFA overnight and then transferred to 30% sucrose solution for cryoprotection. Brains were sectioned at 30 μM on a microtome and stored in a 0.01 M phosphate buffer at 4 °C before confocal imaging.

## Data analysis, quantification, statistics and reproducibility

Phylogenetic trees were generated with phylogeny.fr[95]. In vitro electrophysiological recordings in cultured cells were analyzed using Clampfit 10.7 (Molecular Devices) as well as IgorPro (WaveMetrics) and NeuroMatic[83] for two-photon experiments. Analysis of mEPSC data was performed using Easy Electrophysiology (v2.3.3b) with a 0.37 correlation cutoff and a 15-pA amplitude threshold due to artificial noise created by series resistance compensation. G-protein coupling assays (TRUPATH, GsX, GloSensor) were analyzed in Microsoft Excel. Confocal imaging and calcium imaging data were analyzed in ImageJ[89]. Data from organotypic slice recordings were analyzed with MATLAB. Atrial cardiomyocyte beating was analyzed using LabVIEW (National Instruments). In vivo experiments were analyzed using MATLAB, GraphPad Prism and DeepLabCut[94]. Statistical analysis was performed with MATLAB and GraphPad Prism 9 or 10, and estimation statistics were performed online[96]. Schematic brain slice representations were extracted from the enhanced and unified mouse brain atlas[97].

Sample sizes were similar to those commonly used in the field, and no statistical tests were run to predetermine sample size. Blinding was performed in confocal quantification of expression levels, autaptic benchmark experiments (Fig. 2e) and behavioral experiments silencing the nigrostriatal pathway. Randomization was performed in the nigrostriatal pathway silencing experiments, for biophysical characterization of optoGPCRs in autaptic neurons and for two-photon characterization. Further, the trial order was randomized for stimulation frequency and stimulation location within each experiment for pupillometry experiments. Automated analysis was used whenever possible. In electrophysiological experiments, cells were always patched randomly without any preselection by fluorescence intensity. For autaptic neuron recordings, cells were excluded from analysis if the first EPSC amplitude was below 100 pA, and from the analysis of the paired-pulse ratio if optoGPCR activation completely abolished the first EPSC. Further, cells were excluded from the analysis if the access resistance was above 20 MΩ or if the holding current exceeded 200 pA. For organotypic slice culture recordings, cells were additionally excluded from analysis if an EPSC amplitude drift >30% occurred. All data points represent measurements from biological replicates or animals. All in vitro data points originate from at least two independent biological samples (for example, neuronal cultures or organotypic slices from different batches of mice) or batches of cultured HEK cells.

The micrographs shown are representative of experiments that have been independently repeated with similar results, except for Extended Data Fig. 9a,b where expression was usually confirmed by one-photon fluorescence instead of the shown two-photon micrograph.

## Reporting summary

Further information on research design is available in the Nature Portfolio Reporting Summary linked to this article.

## Data availability

Due to the complexity of the data and the multiple participating groups, raw data for specific experiments will be provided by the corresponding authors upon request. Source data are provided with this paper.

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

## Acknowledgements

We thank all our laboratory members and collaborators for thoughtful discussions and comments on the manuscript. We thank A. Inoue (Tohoku University) for providing the delta-G7 HEK293 cell line as well as A. Tinker (Queen Mary's School of Medicine and Dentistry) and S. Herlitze (Ruhr-Universität Bochum) for providing HEK293 cells stably expressing GIRK1/2 subunits, P. Hegemann (Humboldt-Universität zu Berlin) for *Lc*PPO DNA, and E. Reuveny (Weizmann Institute of Science) for providing pcDNA3.1-CMV-GIRK2.1 plasmid, R. J. Lucas (University of Manchester) for GsX plasmids, and B. Roth (University of North Carolina at Chapel Hill) for the TRUPATH kit. We further thank E. Entcheva, H. Janoviak, M. Sheves, M. Shalev-Benami, S. Bitzenhofer and J. Vierock for fruitful discussions.

This work is supported by grants from the Deutsche Forschungsgemeinschaft (DFG; German Research Foundation) SPP 1926 (to B.R.R., O.Y., J.S.W., P. Sasse and P. Soba), EXC-2049 – 390688087, SPP 1665, SFB 958 (to D. Schmitz), SFB 1315 (to B.R.R. and D. Schmitz), the National Institutes of Health (1U01NS128537-01 to B.A.C., M.R.B. and O.Y.), the European Horizon 2020 Program (ERC CoG PrefrontalMap 819496 and H2020-RIA DEEPER 101016787 to O.Y., ERC SyG BrainPlay 810580 to D. Schmitz, ERC-2016-StG 714762 to J.S.W., ERC-2019-STG 850784 to M.O.-S. and ERC-2019-CoG 864353 to Y.N.) and the Israel Science Foundation (ISF 3131/20 to O.Y., ISF 961/21 to M.O.-S. and ISF 1557/2 to Y.N.). J.W. is supported by the EMBO ALTF 378-2019 and Amos de Shalit-Minerva fellowship. O.Y. is supported by the Joseph and Wolf Lebovic Charitable Foundation Chair for Research in Neuroscience. M.O.-S. is supported by the Dr. Barry Sherman Institute for Medicinal Chemistry, Sagol Weizmann-MIT Bridge Program and the Azrieli Foundation and is the incumbent of the Jenna and Julia Birnbach Family Career Development Chair. J.D. and I.S.-S. are incumbents of the Achar research fellow chair in electrophysiology.

## Author contributions

Conceptualization: J.W., J.S.W., O.Y. and I.S.-S. Data acquisition: J.W., A.N., M.P., I.S.-S., N.M., R.G., D.M., B.J.B., B.N.I., A.G., J.D., K.A., D. Summarli, R.L., K.S., E.B. and E.M.G. Analysis: J.W., A.N., M.P., I.S.-S., N.M., R.G., D.M., B.J.B., B.N.I., R.L., A.G., K.S., E.B., A.B., G.M. and E.M.G. Methodology: J.W., A.N., M.P., I.S.-S., N.M., R.G., D.M., B.J.B., A.L., R.L., K.S., M.R.B., E.B. and S.S. Supervision: O.Y., J.S.W., D. Schmitz, P. Soba, M.O.-S., P. Sasse, B.A.C., Y.N., M.R.B., B.R.R. and J.W. Funding acquisition: O.Y., J.S.W., M.R.B., D. Schmitz, P. Soba, M.O.-S., P. Sasse, B.A.C., Y.N., B.R.R. and J.W. Project administration: J.W. and O.Y. Writing: J.W. and O.Y. with help from all other authors.

## Competing interests

O.Y. is listed as an inventor on a patent application (US20210403518A1) filed with the US Patent Office regarding type II bistable opsins and serves as a consultant for Modulight.bio. The other authors declare no competing interests.

## Additional information

**Extended data** is available for this paper at https://doi.org/10.1038/s41592-024-02285-8.

**Correspondence and requests for materials** should be addressed to Jonas Wietek or Ofer Yizhar.

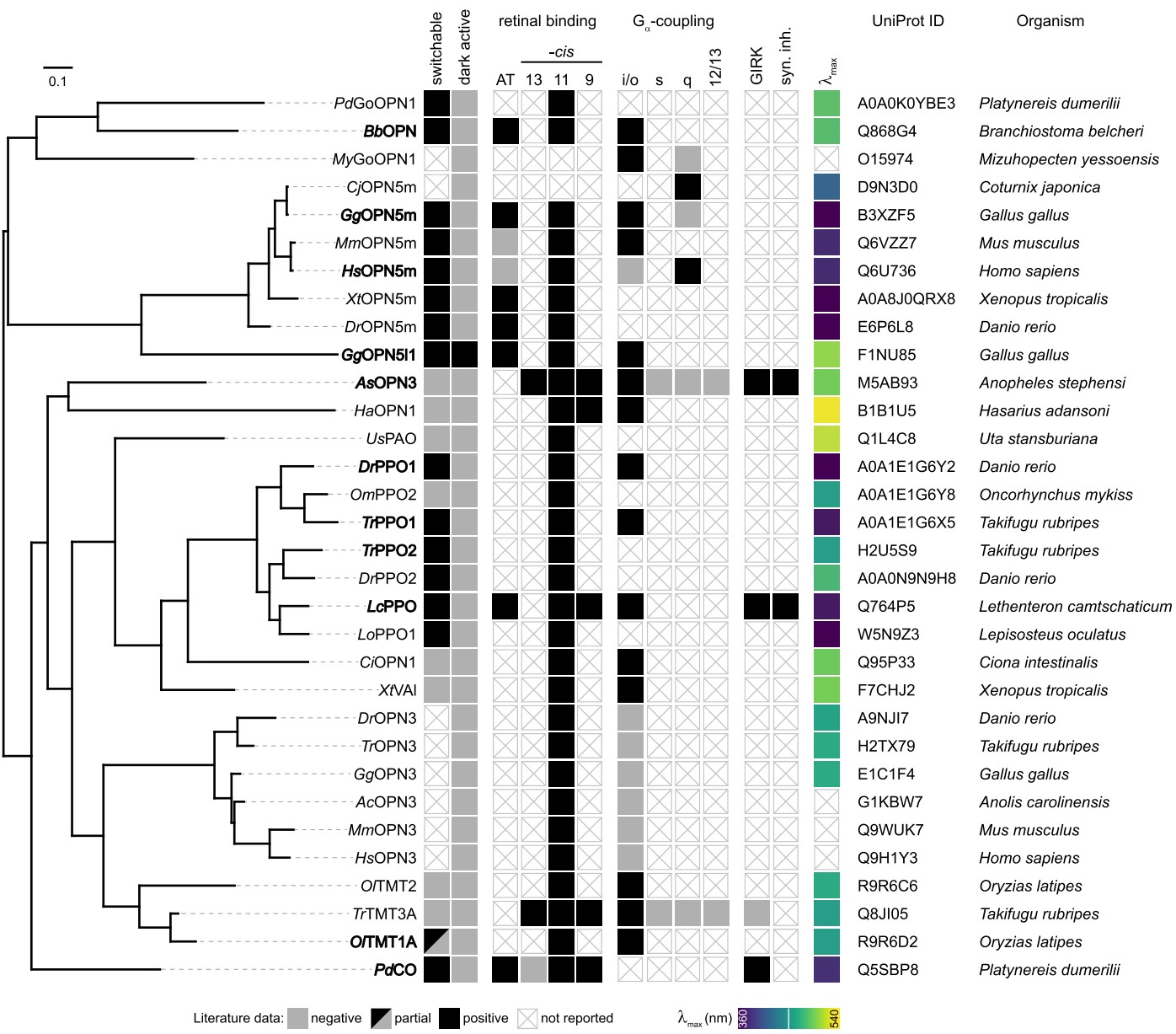

**Extended Data Fig. 1 | Curated literature data of optoGPCRs.** Overview of all optoGPCR and their properties from literature. Left: phylogenetic tree. Middle section (squares): optoGPCR properties. Right: UniProt identifiers and species of origin of optoGPCRs.

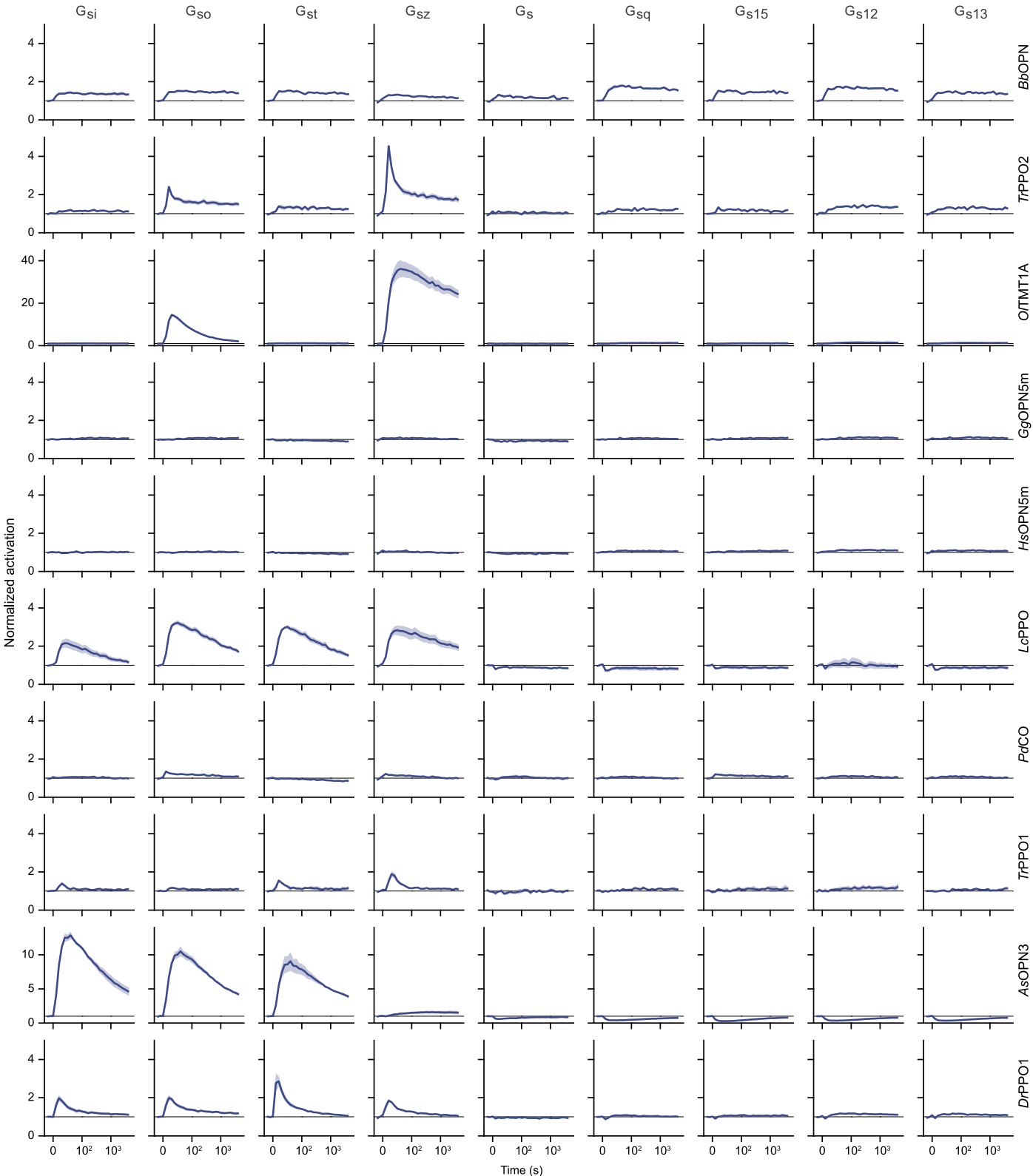

**Extended Data Fig. 2 | GsX assay of optoGPCR candidates.** Time course of averaged bioluminescence reads for Gs-protein chimeras (columns) and optoGPCRs (rows). For *Ol*TMT1A, *Tr*PPO2, and *Bb*OPN a 1s 470nm light pulse (Time = 0s) was used for optoGPCR activation, while all other optoGPCRs were activated with a 1s 365nm light pulse. Please note different scales for *As*OPN3 and *Ol*TMT1A. All data is shown as mean ± SEM (n = 2–6).

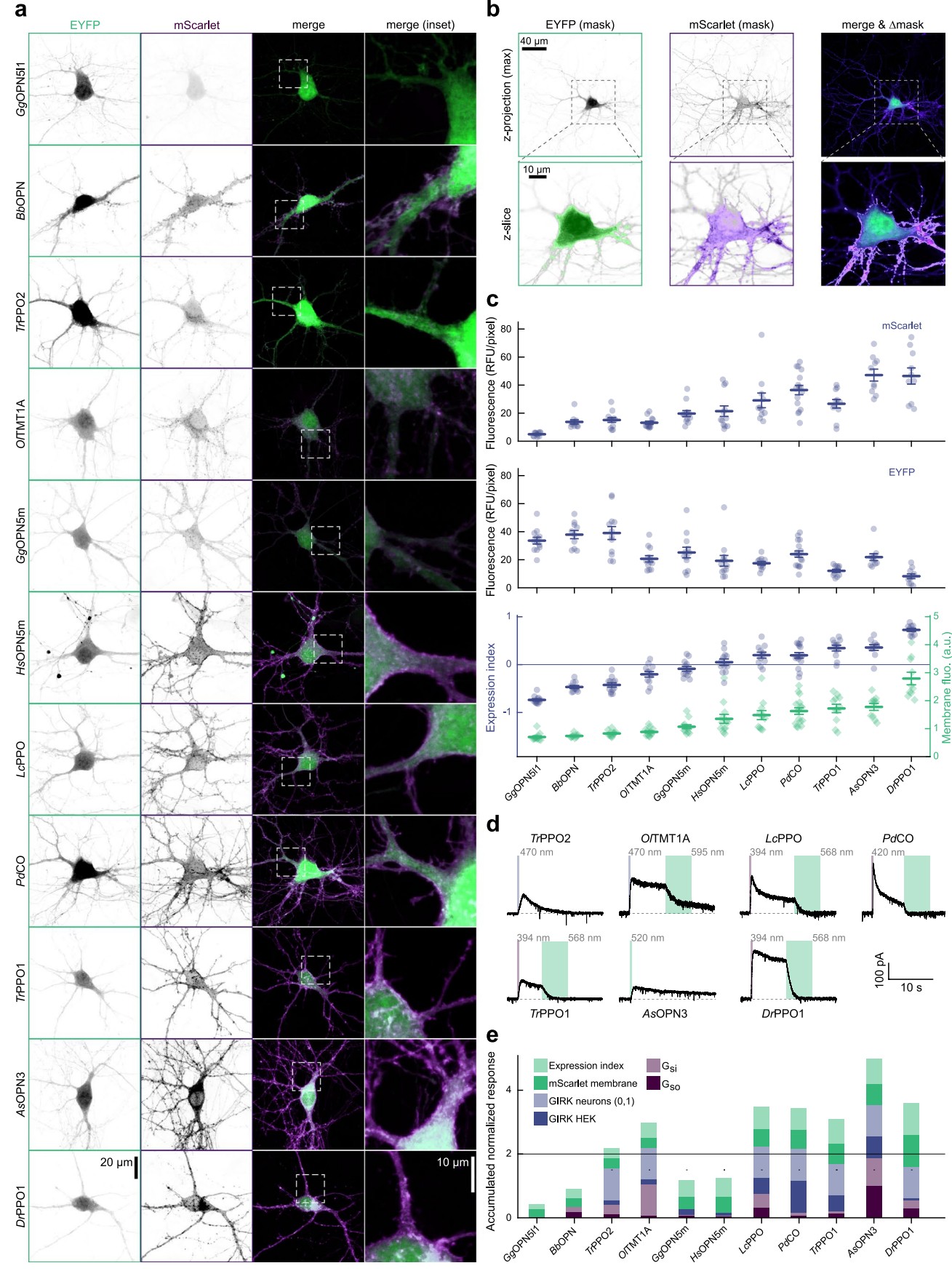

**Extended Data Fig. 3 | See next page for caption.**

**Extended Data Fig. 3 | optoGPCR expression, GIRK coupling in cultured neurons and benchmark summary. a**, Representative maximum intensity projection confocal images of neurons expressing the optoGPCRs together with a cytosolic EYFP. From left to right column: EYFP, mScarlet and merge of both signals. Inverted grayscale images are gamma corrected (1.25). The merged projections and magnified projections are false color coded by green (EYFP) and purple (mScarlet) lookup tables. **b**, *Pd*CO example for quantification of fluorescence measurements. Zoomed in single z-slices are shown for each channel in the lower row together with the calculated binary masks. To quantify the membrane expression only (bottom row, right), the calculated mask for the cytosolic EYFP channel was subtracted from the mScarlet channel mask. Color coding and lookup tables as in a, but gamma (1.75) for single z-slice images. **c**, Quantification of total mScarlet (top) and EYFP (middle) fluorescence mean gray intensities, determined from equatorial z-slices. (bottom) Quantification of membrane expression index (blue circles) and the membrane fluorescence (green diamonds) of each optoGPCR, determined from equatorial z-slices. **d**, Representative whole-cell patch-clamp recordings of cultured neurons co-transfected with the different optoGPCRs and GIRK2.1, respectively. Neurons were kept at −70mV holding potential and a 0.5s light pulse at indicated wavelengths was used for optoGPCR activation, while 5s light pulse at indicated wavelengths was used for optoGPCR inactivation. Illumination intensities were adjusted to equal photon flux density for all wavelengths. **e**, Accumulated normalized response across all assays used to benchmark optoGPCR candidates against each other. Within each assay, the mean value of each optoGPCR was normalized to the maximum mean value within the regarding assay. For positive GIRK-coupling shown in a, a value of 1 (0, in case no coupling detected) was assigned. Logarithmic GsX mean values were used. An arbitrary cutoff of 2 was chosen for the follow up benchmark in autaptic neurons. n = 10–16. All data is shown as mean ± SEM.

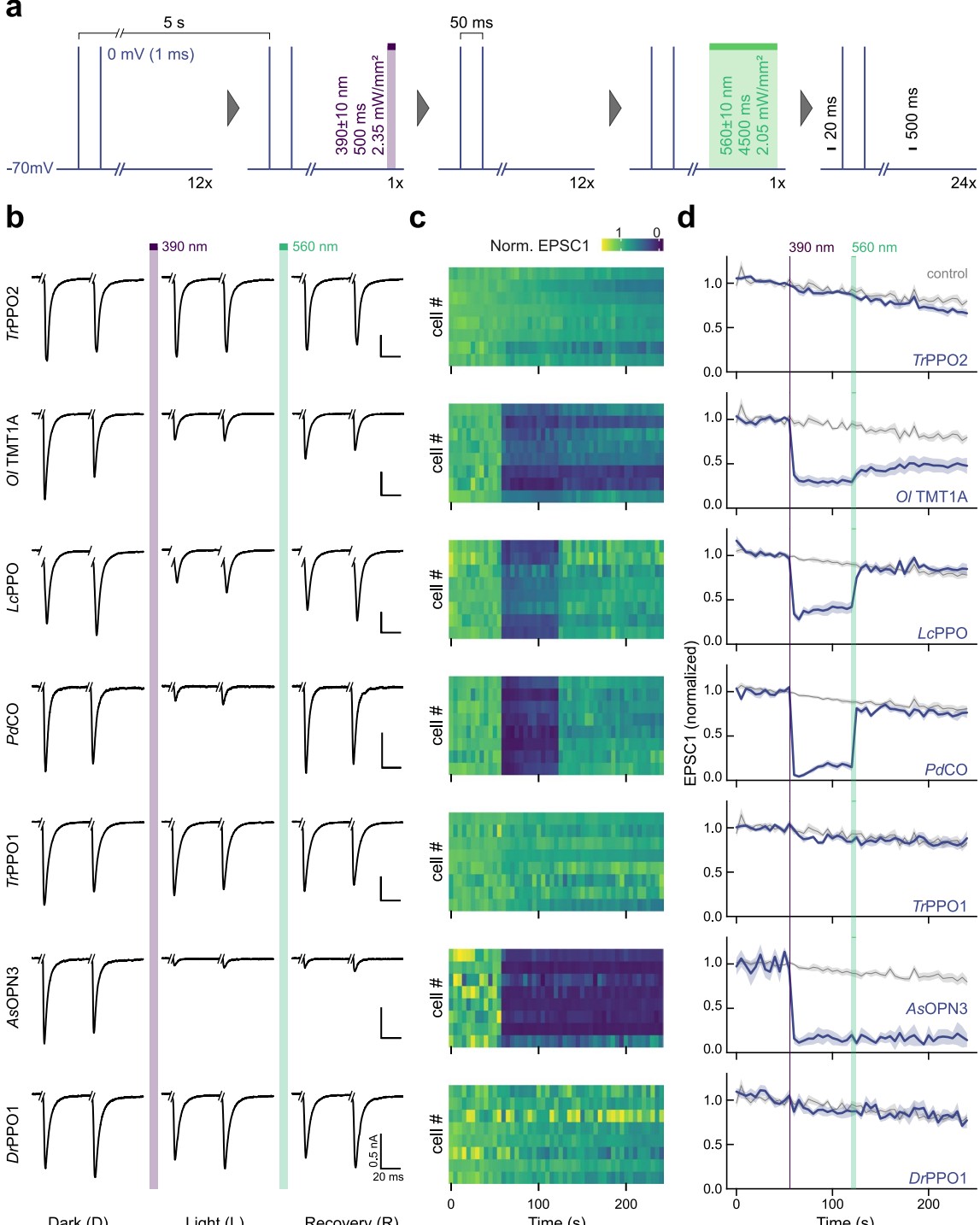

**Extended Data Fig. 4 | Inhibition of synaptic transmission in autaptic neurons. a,** Experimental scheme of EPSC recordings in autaptic neurons. **b,** Representative averaged EPSC traces for the 7 investigated optoGPCR candidates recorded as described in a. Traces show 5 averaged sweeps. Current injections were cut for representation. **c,** Contour plot of EPSC amplitudes for 8 biological replicates. Amplitudes were normalized to the average of 5 EPSC1s prior 390 nm illumination. **d,** Timeplot of averaged EPSC data (across replicates) as shown in c, together with non-expressing control cells from matching autaptic cultures measured with the same protocol. Data shown as mean ± SEM (n = 8).

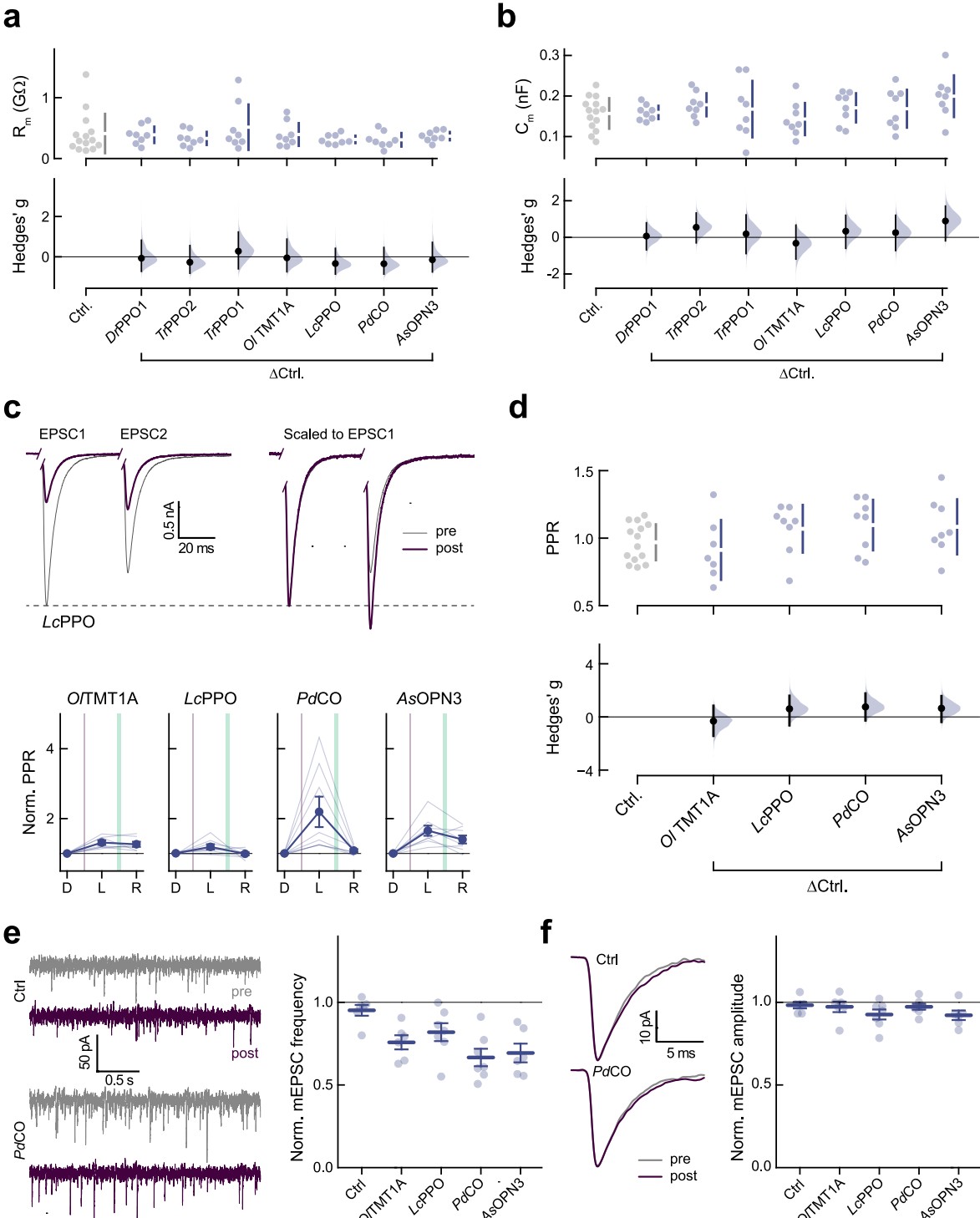

**Extended Data Fig. 5 | Intrinsic properties, paired-pulse recordings, and mEPSCs in autaptic neurons. a**, **b**, Intrinsic properties, including membrane resistance ($R_m$, a) and cell membrane capacitance ($C_m$, b) of non-expressing control (Ctrl.) and optoGPCR-expressing autaptic neurons. The Hedges' g for 7 comparisons against the shared control obtained from the same batch of neurons are shown as a Cumming estimation plot. The raw data is plotted on the upper axes. On the lower axes, mean differences are plotted as bootstrap sampling distributions. Each mean difference is depicted as a dot. Each 95% confidence interval is indicated by the ends of the vertical error bars. n = 8 (optoGPCRs), 14 (Ctrl.). **c**, (top) Paired-pulse recording of an *Lc*PPO expressing neuron (left), scaled to the first EPSC (right). (bottom) Quantification of paired-pulse ratio (EPSC2/EPSC1). n = 7–8. **d**, Baseline paired-pulse ratio of non-expressing control (Ctrl.) and optoGPCR expressing autaptic neurons.

The Hedges' g for 4 comparisons against the shared control obtained from the same batch of neurons are shown as a Cumming estimation plot. The raw data is plotted on the upper axes. On the lower axes, mean differences are plotted as bootstrap sampling distributions. Each mean difference is depicted as a dot. Each 95% confidence interval is indicated by the ends of the vertical error bars. n = 7-8 (optoGPCRs), 14 (Ctrl.). **e**, Representative traces of mEPSCs (left) in a non-transduced control neuron (top) and a *Pd*CO expressing autaptic cell (bottom). Right: Quantification of the mEPSC frequency after the light flash, normalized to the frequency prior light. n = 6-7. **f**, Quantal mEPSC amplitude in control (top trace) and *Pd*CO-expressing (bottom trace) neurons pre and post illumination. Right: Quantification of the mEPSC amplitudes after the light flash, normalized to pre illumination; n = 6-7. All data is shown as mean ± SEM.

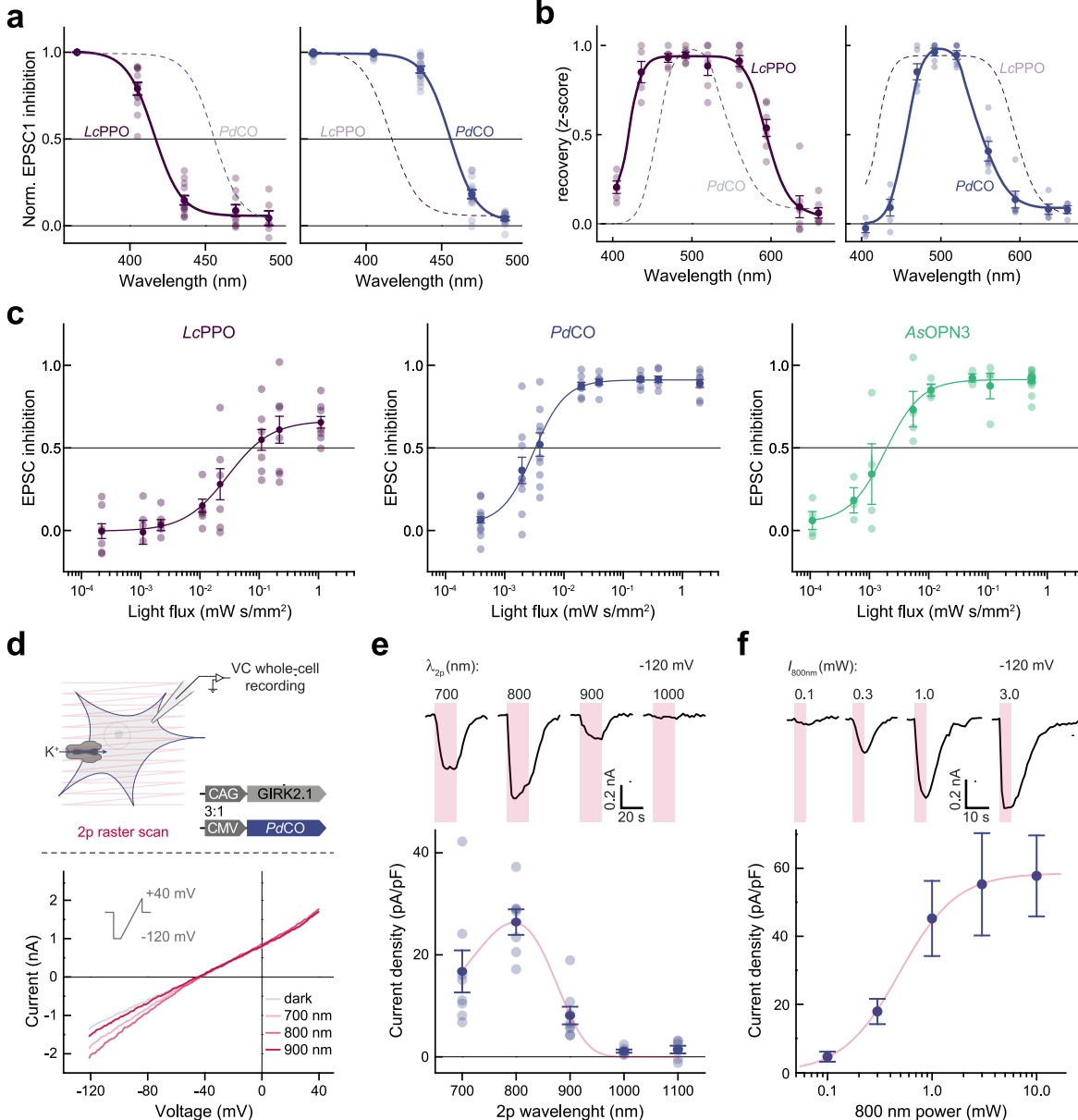

**Extended Data Fig. 6 | Biophysical properties in autaptic neurons and HEK293T cells. a**, Quantification of EPSC inhibition versus applied wavelengths for *Lc*PPO (right) and *Pd*CO (left) including single cell measurement data points. For each cell, the inhibition was normalized to the maximum inhibition. Lines show dose-response fits. n = 6–15. **b**, Wavelength sensitivity of light induced recovery including single cell measurement data for *Lc*PPO (right) and *Pd*CO (left). n = 4–7. **c**, Quantification of absolute inhibition for the first EPSC post sample illumination versus light flux for (from left to right) *Lc*PPO (365 nm), *Pd*CO (405 nm) and *As*OPN3 (520 nm) including single cell measurement data points. Lines depict sigmoidal fits to calculate half maximal inhibition light flux (EC$_{50}$) values. n = 3–17. **d**, Top: schematic of 2-photon (2p) recordings in HEK293T

cells co-transfected with *Pd*CO-mScarlet and GIRK2.1 in a 1:3 ratio. Whole-cell patch-clamp recordings were performed and *Pd*CO was activated with 2p raster scanning. Bottom: example recordings at various 2p wavelengths while applying a voltage ramp from −120 to +40mV. **e**, Top: example recordings of one representative HEK293T cell at −120mV holding potential for different scan wavelengths as indicated. Bottom: Quantification of 2p wavelength sensitivity; n = 6–8. Data was fitted with a 3-parametric Weibull distribution (red line) **f**, Top: example recordings of a representative HEK293T cell at −120mV for different 2p intensities of 800 nm. Bottom: Quantification of 2p sensitivity at 800 nm; n = 6–8. Data was fitted with a logistic dose response curve to determine the EC$_{50}$ value (red line). All data is shown as mean ± SEM.

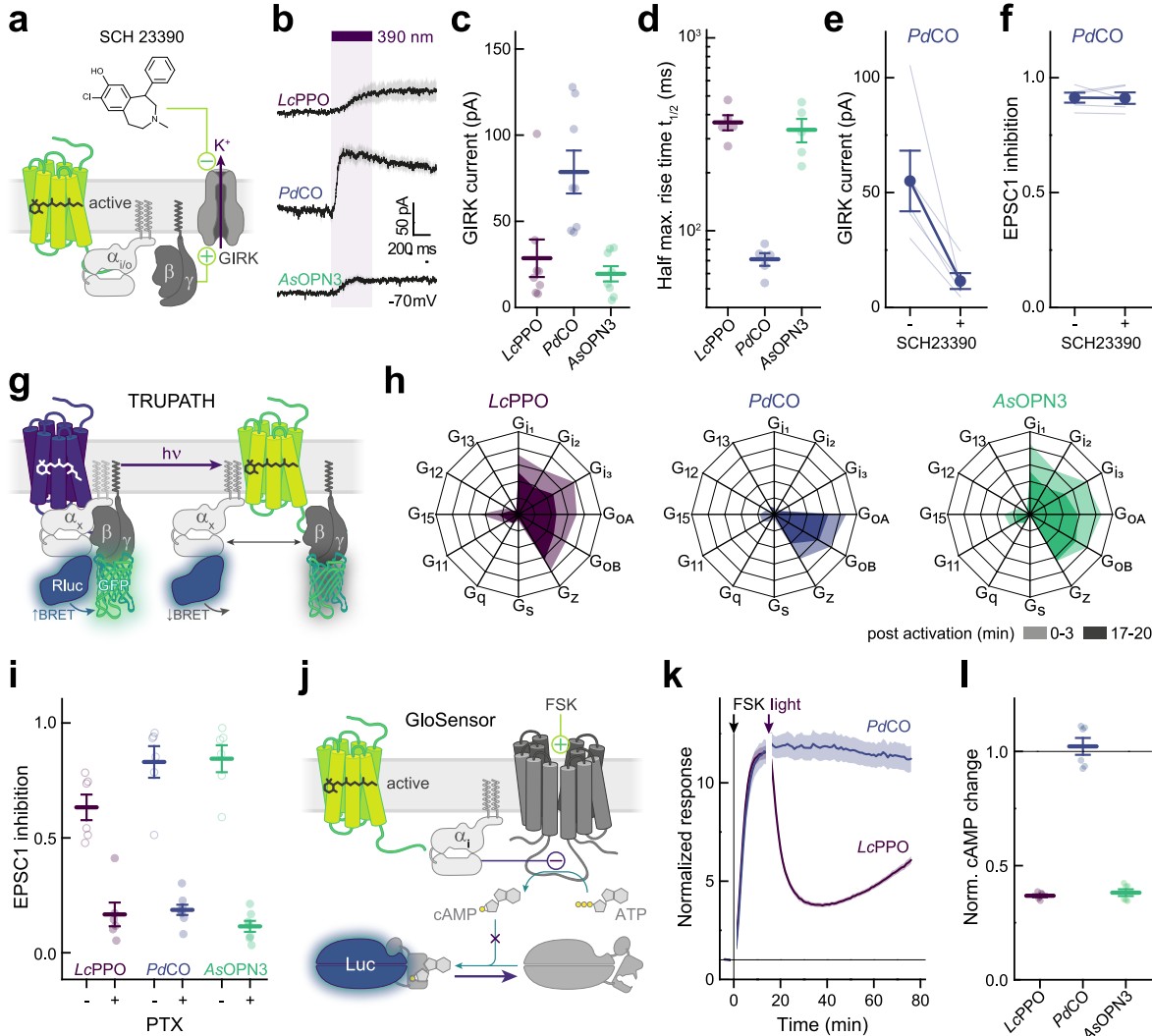

**Extended Data Fig. 7 | G-protein signaling in potent inhibitory optoGPCRs.**
**a**, Scheme of GIRK activation and inhibition. Gβγ-complex signaling by photoactivated optoGPCRs activates GIRK channels, which are inhibited by SCH23390. **b**, Average current traces from cells expressing the indicating optoGPCRs, showing GIRK channel activation induced by light application in autaptic neurons. **c**, Quantification of maximum GIRK channel currents (n = 8). **d**, Quantification of the rise time to the half maximum current determined from the neurons shown in b. **e**, GIRK channel currents activated by light and inhibited by local application of SCH23390 in autaptic neurons (n = 5). **f**, EPSC1 inhibition of the same neurons shown in e. Application of SCH23390 does not alter EPSC inhibition. **g**, TRUPATH assay. Activation of the optoGPCR reduces the bioluminescence energy transfer (BRET) between the luciferase-fused Gα and the GFP-fused Gβγ subunits. **h**, Relative -netBRET means (n = 4) for major Gα subtypes integrated for 0–3 min post light activation (1s 390 nm; light shaded

area) and integrated for 17–20 min post light activation (dark shaded area). **i**, EPSC1 inhibition as measured in Fig. 2a–e in autaptic neurons (open circles) and pertussis toxin (PTX) treated neurons from matched neuronal cultures. Cells were incubated with PTX for 12–16 h (n = 6–8). **j**, The GloSensor assay for inhibition of adenylyl cyclase (AC) activity by optoGPCRs. Forskolin (FSK) stimulates cAMP production by ACs, which can be inhibited by Gα$_i$ signaling induced by optoGPCRs. Changes in cAMP levels can be detected by cAMP-dependent Luciferase (Luc). **k**, Time course of cAMP response measured with the GloSensor assay for PdCO and LcPPO, after addition of FSK (t = 0 min; black arrow) and after activation of LcPPO and PdCO (purple arrow). Data is normalized to bioluminescence reads pre-FSK application (n = 6). **l**, Normalized cAMP changes after light application, calculated by division of minimum response post illumination by maximum pre-illumination response of data as shown in k (n = 6). All data is shown as mean ± SEM.

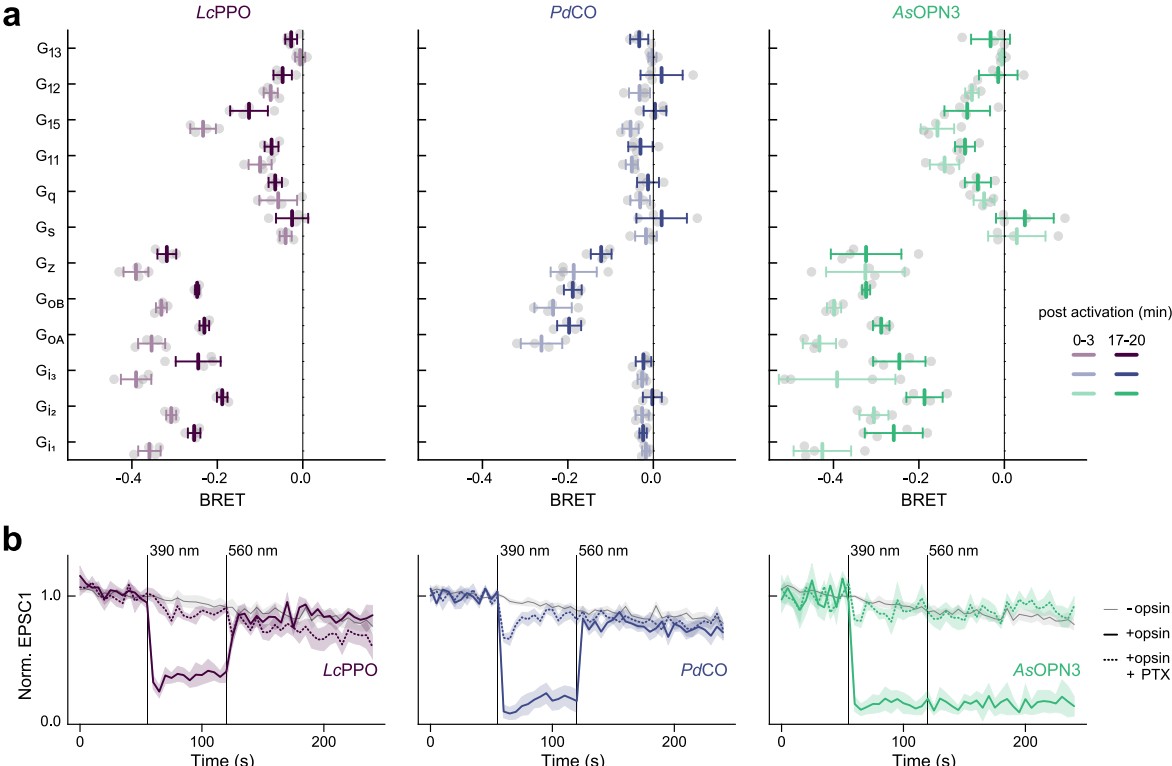

**Extended Data Fig. 8 | TRUPATH assay and pertussis toxin treatment in autaptic cultures. a**, TRUPATH netBRET data for all G-protein classes integrated over 3 minutes post optoGPCR activation (light colors) and 17–20 minutes post activation (saturated colors). n = 4. **b**, EPSC recordings in autaptic neurons as described for initial benchmark of optoGPCRs with additional pertussis toxin (PTX) treatment. PTX strongly reduces optoGPCR-mediated EPSC inhibition, and therefore indicates a $G_{i/o}$-dependent mechanism. n = 6–8. All data is shown as mean ± SEM.

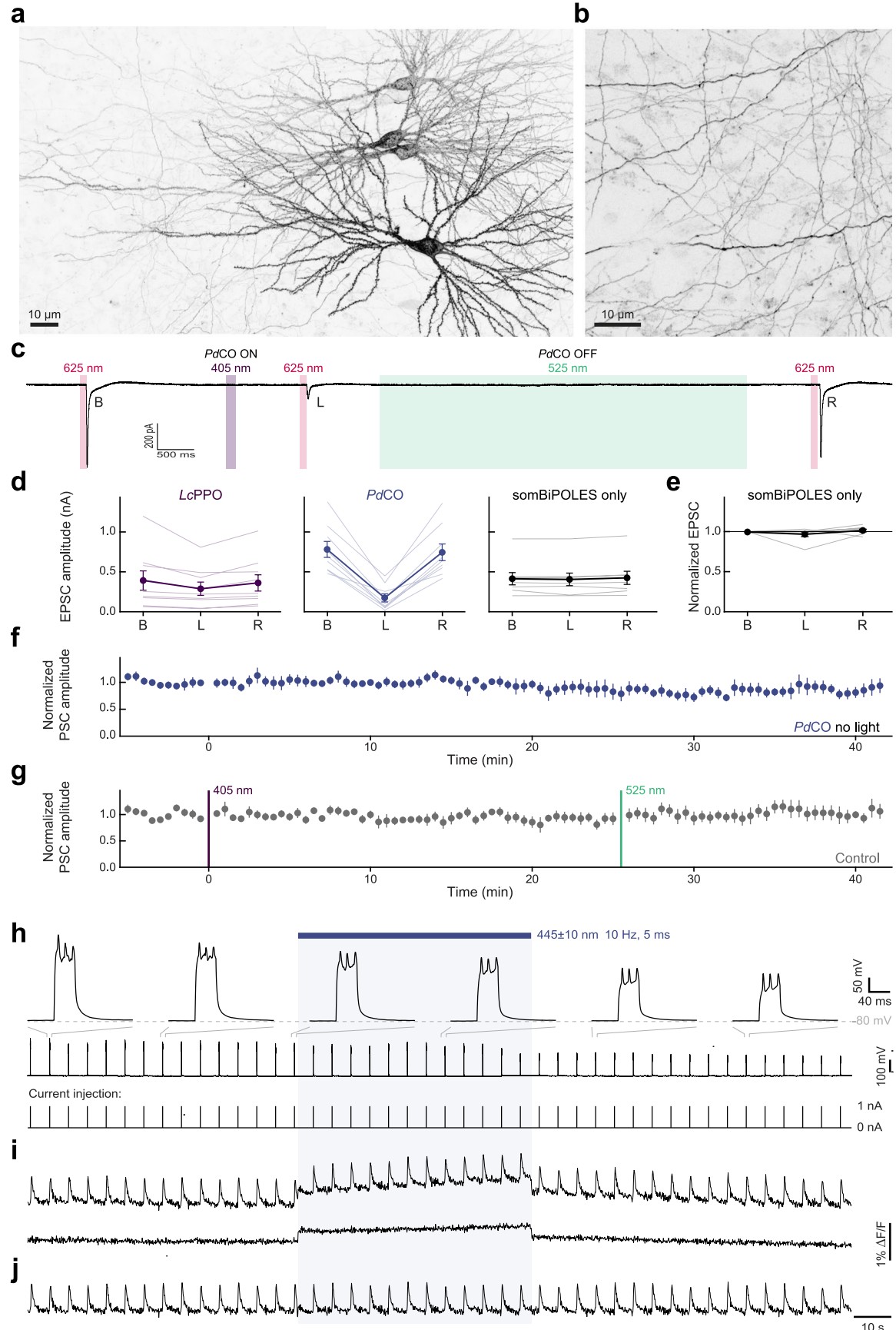

**Extended Data Fig. 9 | See next page for caption.**

**Extended Data Fig. 9 | Organotypic slices measurements and calcium imaging. a**, Two-photon maximum-intensity projections of CA3 neurons expressing *Pd*CO-mScarlet (delivered via electroporation). **b**, Axons (originating from CA3 neurons (shown in a) projecting into stratum radiatum of CA1. **c**, Experimental design, and representative current trace for bidirectional optogenetic control of synaptic transmission at Schaffer collateral synapses using somBiPOLES and *Pd*CO. **d**, Quantification of absolute PSC amplitudes from bidirectional optogenetic experiments. n = 8-9. **e**, Normalized PSC amplitudes from control cultures expressing somBiPOLES alone, same as right panel in b. n = 8. **f**, Time course of normalized PSC amplitudes from control cultures expressing *Pd*CO alone, without light stimulation. n = 4–6. **g**, Time course of normalized PSC amplitudes from control, non-expressing cultures, before and after light stimulation. n = 5-6. **h**, Voltage recording under tight-seal current-clamp conditions. 1 nA current pulses (40 ms duration) were injected through the intact membrane patch at 0.2 Hz to evoke a train of action potentials. Top: magnified voltage recordings at selected time points. Middle: voltage recording of the complete experiment. Bottom: current injection. **i**, Top: raw ΔF/F signal from the neuron shown in a. Bottom: raw ΔF/F signal from a neuron in the same field of view recorded simultaneously. **j**, manually corrected ΔF/F signal as shown in b, top. All data is shown as mean ± SEM.

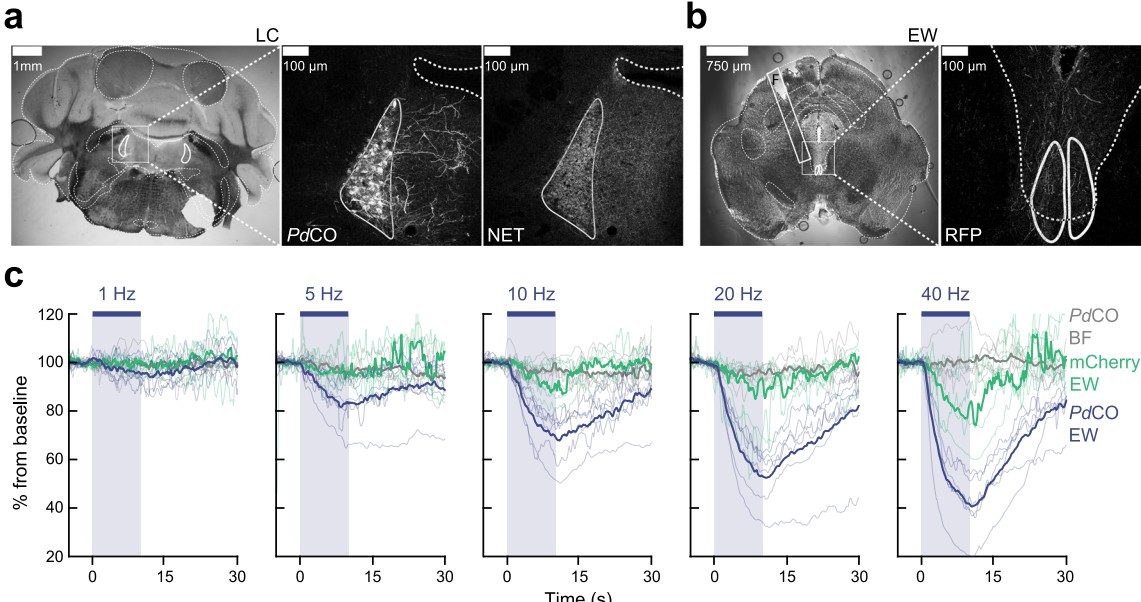

**Extended Data Fig. 10 | Histology and control pupil experiments. a**, Coronal brain section of the locus coeruleus (LC) injection site. Left: Bright field image. Center: confocal single-channel fluorescence of *Pd*CO-mScarlet at the LC. Right: confocal single-channel fluorescence image of norepinephrine transporter (NET) at the LC. **b**, Coronal brain section of the Edinger-Westphal (EW) projection site. Left: Bright field image including fiber placement (F). Right: confocal fluorescence expression of *Pd*CO-mScarlet stained with anti-RFP at the EW.

All fluorescence micrographs are gamma-enhanced (1.25). **c**, Time course of mean (bold lines) ipsilateral pupillometry traces for *Pd*CO expressing mice (n = 6) illuminated at the basal forebrain (BF, gray) or the EW (blue), compared to mCherry expressing control animals (n = 5) illuminated at the EW (green) at different laser stimulation frequencies as indicated (vertical blue shaded area). Thin traces denote individual mice.

# Reporting Summary

## Statistics

For all statistical analyses, confirm that the following items are present in the figure legend, table legend, main text, or Methods section.

| n/a | Confirmed | |
|---|---|---|
| ☐ | ☒ | The exact sample size (*n*) for each experimental group/condition, given as a discrete number and unit of measurement |
| ☐ | ☒ | A statement on whether measurements were taken from distinct samples or whether the same sample was measured repeatedly |
| ☐ | ☒ | The statistical test(s) used AND whether they are one- or two-sided *Only common tests should be described solely by name; describe more complex techniques in the Methods section.* |
| ☒ | ☐ | A description of all covariates tested |
| ☐ | ☒ | A description of any assumptions or corrections, such as tests of normality and adjustment for multiple comparisons |
| ☐ | ☒ | A full description of the statistical parameters including central tendency (e.g. means) or other basic estimates (e.g. regression coefficient) AND variation (e.g. standard deviation) or associated estimates of uncertainty (e.g. confidence intervals) |
| ☐ | ☒ | For null hypothesis testing, the test statistic (e.g. *F*, *t*, *r*) with confidence intervals, effect sizes, degrees of freedom and *P* value noted *Give P values as exact values whenever suitable.* |
| ☒ | ☐ | For Bayesian analysis, information on the choice of priors and Markov chain Monte Carlo settings |
| ☒ | ☐ | For hierarchical and complex designs, identification of the appropriate level for tests and full reporting of outcomes |
| ☒ | ☐ | Estimates of effect sizes (e.g. Cohen's *d*, Pearson's *r*), indicating how they were calculated |

*Our web collection on statistics for biologists contains articles on many of the points above.*

## Software and code

Policy information about availability of computer code

| Data collection | In vitro electrophysiological data was recorded with Clampex 10.7 (Molecular Devices), WinWCP 5.7 (https://github.com/johndempster/WinWCPXE), Ephus R220 (doi:10.3389/fncir.2010.00100), SutterPatch software (Sutter Instruments), or Patchmaster software (HEKA). Confocal images were acquired using Zeiss Zen 3.7 software (Carl Zeiss), Leica LAS AF software, or Olympus FluoView FV3000. Other imaging data was acquired using Micro-Manager 2.0 (doi:10.14440/jbm.2014.36). Video-microscopy data from cardiomyocytes was aquired with LabView (National Instruments). Bioluminescence data was acquired using MikroWin2010 (Mikrotek Laborsysteme GmbH). Pupil video recordings in anesthetized animals were captured using USB web cameras (Logitech C615) with removed IR filter connected to an TDT RZ2 processor (Tucker Davis Technologies). The RZ2 was programmed through the TDT Synapse software and allowed automated control of laser activation through a MATLAB code, and synchronization with the camera feeds. Nigrostriatal pathway inhibition behavior was video recorded using IC Capture, 30 FPS, 672x672 px. Master 8 controlled the timing of LED protocols and video synchronization signal that was recorded in the video. TTL signals (protocol onset, activation LED on and off, Recovery LED on and off, and video sync on and off, were recorded using OpenEPhys, with their GUI. C. elegans were tracked with WormLab (MBF Bio-science). |
|---|---|
| Data analysis | Phylogenetic trees were generated with phylogeny.fr (DOI: 10.1093/nar/gkn180). In vitro electrophysiological recordings in cultured cells were analyzed using Clampfit 10.7 (Molecular Devices) as well as IgorPro (Wavemetrics) and NeuroMatic (DOI: 10.3389/fninf.2018.00014) for two-photon experiments. Analysis of mEPSCs data was performed using Easy electrophysiology (v2.3.3b) with a 0.37 correlation cutoff and a 15 pA amplitude threshold due to artificial noise created by series resistance compensation. G-protein coupling assays (TRUPATH, GsX, GloSensor) were analyzed in Microsoft Excel. Confocal imaging and calcium imaging data were analyzed in ImageJ (DOI: 10.1038/nmeth.2019). Data from organotypic slice recordings was analyzed with MATLAB. Atrial cardiomyocyte beating was analyzed using LabView (National Instruments). C. elegans data was analyzed in Python. In vivo experiments were analyzed using Matlab, Graphpad Prism, and DeepLabCut (DOI: 10.1038/s41593-018-0209-y). Statistical analysis was performed with MATLAB, Graphpad Prism 9 or 10 and estimation |

statistics were performed online (DOI: 10.1038/s41592-019-0470-3). Schematic brain slice representations were extracted from the enhanced and unified mouse brain atlas (DOI: 10.1038/s41467-019-13057-w).

For manuscripts utilizing custom algorithms or software that are central to the research but not yet described in published literature, software must be made available to editors and reviewers. We strongly encourage code deposition in a community repository (e.g. GitHub). See the Nature Portfolio guidelines for submitting code & software for further information.

## Data

Policy information about availability of data

All manuscripts must include a data availability statement. This statement should provide the following information, where applicable:
- Accession codes, unique identifiers, or web links for publicly available datasets
- A description of any restrictions on data availability
- For clinical datasets or third party data, please ensure that the statement adheres to our policy

All data supporting the findings of this study are available as source data files. Raw data will be provided by the corresponding authors upon request.

## Human research participants

Policy information about studies involving human research participants and Sex and Gender in Research.

| | |
|---|---|
| Reporting on sex and gender | N/A |
| Population characteristics | N/A |
| Recruitment | N/A |
| Ethics oversight | N/A |

Note that full information on the approval of the study protocol must also be provided in the manuscript.

# Field-specific reporting

Please select the one below that is the best fit for your research. If you are not sure, read the appropriate sections before making your selection.

☒ Life sciences      ☐ Behavioural & social sciences      ☐ Ecological, evolutionary & environmental sciences

For a reference copy of the document with all sections, see nature.com/documents/nr-reporting-summary-flat.pdf

# Life sciences study design

All studies must disclose on these points even when the disclosure is negative.

| | |
|---|---|
| Sample size | Sample sizes were similar to those commonly used in the field (DOI: 10.1126/sciadv.add7729, DOI: 10.1016/j.neuron.2021.03.013, DOI: 10.1016/j.neuron.2021.04.026) and no statistical tests were run to predetermine sample size. |
| Data exclusions | For autaptic neuron recordings, cells were excluded from analysis if the first EPSC amplitude was below 100pA, and from the analysis of the paired-pulse ratio if optoGPCR activation completely abolished the first EPSC. Further, were cells excluded from analysis if the access resistance was above 20 MΩ or if the holding current exceeded 200 pA. For organotypic slice culture recordings cells were additionally excluded from analysis if a EPSC amplitude drift > 30% occurred. |
| Replication | All datapoints represent measurements from biological replicates. All in vitro data points originate at least from two independent biological samples (e.g. neuronal cultures or organotypic slices from different batches of mice) or batches of cultured HEK cells. All replication attemps were successful. |
| Randomization | Randomization was performed in the nigrostriatal pathway silencing experiments, for biophysical characterization of optoGPCRs in autaptic neurons, and for two-photon characterization. Further, the trial order was randomized for stimulation frequency and stimulation location within each experiment for pupillometry experiments. In electrophysiological experiments, cells were always patched randomly without any preselection by fluorescence intensity. |
| Blinding | Blinding was performed in confocal quantification of expression levels, autaptic benchmark experiments (Fig. 2e) and behavioral experiments silencing the nigrostriatal pathway. Blinding during acquisition was not possible in cases where the cellular expression patterns were clearly different between constructs. |

# Reporting for specific materials, systems and methods

We require information from authors about some types of materials, experimental systems and methods used in many studies. Here, indicate whether each material, system or method listed is relevant to your study. If you are not sure if a list item applies to your research, read the appropriate section before selecting a response.

## Materials & experimental systems

| n/a | Involved in the study |
|---|---|
| ☐ | ☒ Antibodies |
| ☐ | ☒ Eukaryotic cell lines |
| ☒ | ☐ Palaeontology and archaeology |
| ☐ | ☒ Animals and other organisms |
| ☒ | ☐ Clinical data |
| ☒ | ☐ Dual use research of concern |

## Methods

| n/a | Involved in the study |
|---|---|
| ☒ | ☐ ChIP-seq |
| ☒ | ☐ Flow cytometry |
| ☒ | ☐ MRI-based neuroimaging |

# Antibodies

| | |
|---|---|
| Antibodies used | Primary antibodies:<br>anti-NET 1:300, NET05-2 (MAb Technologies)<br>guinea pig anti RFP 1:500, 390004 (SYSY)<br><br>Secondary antibodies:<br>AF488 goat anti-mouse 1:500, ab150117 (Abcam)<br>donkey anti guinea pig CyTM5 1:300, 706-175-148 (Jackson ImmunoResearch) |
| Validation | Primary antibodies were previously tested in mice (see references at https://www.sysy.com/product-factsheet/SySy_390004 and https://mabtechnologies.com/categories/product/7-norepinephrine-transporter-mouse-net05-2) |

# Eukaryotic cell lines

Policy information about cell lines and Sex and Gender in Research

| | |
|---|---|
| Cell line source(s) | HEK293T cells (HEK293T/17, ATCC, #CRL-1573), HEK293 cells stably expressing GIRK1/2 subunits (Dr. A. Tinker, Queen Mary's School of Medicine and Dentistry). HEK293ΔG7 (lacking GNAS/GNAL/GNAQ/GNA11/GNA12/GNA13/GNAZ; A. Inoue, Tohoku University, Japan) |
| Authentication | Cell lines were authenticated as follows: HEK293T/17 (authenticated by STR profiling by vendor), GIRK1/2 cell line (authenticated by patch-clamp), HEK293ΔG7 (functually tested for the absence of endogenous GS in GsX assay). |
| Mycoplasma contamination | Cell lines tested negative for mycoplasma contamination. |
| Commonly misidentified lines (See ICLAC register) | No commonly misidentified cell lines were used in the study. |

# Animals and other research organisms

Policy information about studies involving animals; ARRIVE guidelines recommended for reporting animal research, and Sex and Gender in Research

| | |
|---|---|
| Laboratory animals | In-vivo experiments: DAT-IRES-Cre, age: 8-27 weeks (The Jackson Laboratory, Strain #006660), DBH-Cre, age: 8-12 weeks (B6.FVB(Cg)-Tg(Dbh-cre)KH212Gsat/Mmucd; 036778-UCD-HEMI), pdyn-Cre, age 12-16 weeks (PDYN-IRES-Cre, The Jackson Laboratory, Strain #027958), and WT animals, age 12:-16 weeks (The Jackson Laboratory, Strain #000664).<br>In-vitro experiments (all age P0): Primary neuronal cultures from Sprague-Dawley rat pups (Envigo), autaptic neuronal cultures from WT mice (C57BL/6NHsd; Envigo, Cat#044), and organotypic slice cultures from Wistar rat pups.<br>C. elegans (age: larval stage 4): MOS704 etyEx248 [pAG10 (rab-3p::PdCO::SL2::2XNLS::tagRFP), lite-1(ce314) X. MOS772 syb7630 [etyEx248 integrated], lite-1(ce314) X, outcrossed X2.<br>Drosophila melanogaster (age: staged 3d instar larvae, 96h after egg laying): cha-Gal4>UAS-PdCO |
| Wild animals | Na wild animals were used in the study. |
| Reporting on sex | Animals of either sex have been used for all experiments. Sex differences were not considered in study design. Sex based analysis were not performed due to the general molecular nature of GPCR-mediated inhibition of synaptic transmission. |
| Field-collected samples | No field collected samples were used in the study. |
| Ethics oversight | All experiments involving animals were carried out according to the guidelines stated in directive 2010/63/EU of the European Parliament on the protection of animals used for scientific purposes. Animal experiments at the Weizmann Institute of Science were approved by the Institutional Animal Care and Use Committee (IACUC) of the Weizmann Institute; experiments in Berlin were approved by the Berlin local authorities and the animal welfare committee of the Charité-Universitätsmedizin Berlin, Germany. Experiments in Bonn and Hamburg were performed in accordance with the guidelines of local authorities. Experiments in Tel Aviv were approved by the Institutional Animal Care and Use Committee (IACUC) of Tel Aviv University (approval 01-19-037). Experiments |

performed at the University of Washington, Seattle were approved by the Animal Care and Use Committee of the University of Washington and conformed to US National Institutes of Health guidelines.

Note that full information on the approval of the study protocol must also be provided in the manuscript.

