## [Peer Review File · Nature Methods]

Peer Review Information

Manuscript Title: A bistable inhibitory OptoGPCR for multiplexed optogenetic control of neural circuits

Corresponding author name(s): Jonas Wietek, Ofer Yizhar

Editorial Notes: None

Reviewer Comments & Decisions:

Decision Letter, initial version:

Dear Ofer,

Thanks for your patience. Your Article, "A bistable inhibitory OptoGPCR for multiplexed optogenetic control of neural circuits", has now been seen by two reviewers. As you will see from their comments below, although the reviewers find your work of considerable potential interest, they have raised a number of concerns. We are interested in the possibility of publishing your paper in Nature Methods, but would like to consider your response to these concerns before we reach a final decision on publication. We therefore invite you to revise your manuscript to address these concerns.

[Redacted]

We hope to receive your revised paper within 2-3 months. If you cannot send it within this time, please let us know. In this event, we will still be happy to reconsider your paper at a later date so long as nothing similar has been accepted for publication at Nature Methods or published elsewhere.

OPEN SCIENCE REQUIREMENTS

REPORTING SUMMARY AND EDITORIAL POLICY CHECKLISTS

DATA AVAILABILITY

We strongly encourage you to deposit all new data associated with the paper in a persistent repository where they can be freely and enduringly accessed. We recommend submitting the data to discipline-specific and community-recognized repositories; a list of repositories is provided here:

<http://www.nature.com/sdata/policies/repositories>

All novel DNA and RNA sequencing data, protein sequences, genetic polymorphisms, linked genotype

and phenotype data, gene expression data, macromolecular structures, and proteomics data must be deposited in a publicly accessible database, and accession codes and associated hyperlinks must be provided in the "Data Availability" section.

MATERIALS AVAILABILITY

ORCID

Nature Methods is committed to improving transparency in authorship. As part of our efforts in this direction, we are now requesting that all authors identified as 'corresponding author' on published papers create and link their Open Researcher and Contributor Identifier (ORCID) with their account on the Manuscript Tracking System (MTS), prior to acceptance. This applies to primary research papers only. ORCID helps the scientific community achieve unambiguous attribution of all scholarly contributions. You can create and link your ORCID from the home page of the MTS by clicking on

'Modify my Springer Nature account'. For more information please visit please visit www.springernature.com/orcid.

Best regards,
Nina

Nina Vogt, PhD
Senior Editor
Nature Methods

Reviewers' Comments:

Reviewer #1:

Remarks to the Author:

In this work the authors described a new bistable rhodopsin that seems to have specific selectivity to Go which, as the authors argued, may attribute to the stronger presynaptic inhibition and activation of inhibitory pathways than rhodopsins targeting Gi/o. The main claim to the novelty is the spectrally narrower and red-shifted response and better performance of PdCO versus a previous bistable rhodopsin from Lamprey published by the Bruchas group. The manuscript is generally well-done in the amount of works and characterization and there are no major contradictory results that would make the main finding invalid, however, there are still some minor concerns that should be addressed.

1. Given the main claim for the manuscript is that PdCO performs better than LcPPO, there is no mention of how experimenter's bias is controlled or not controlled in the experiments comparing the performance of LcPPO and PdCO or even other variants. Are there ways to show the cells chosen for recordings have similar level of expression. How is the comparison controlled for bias?
2. Despite the LcPPO performs well in microisland recordings at disrupting release, the performance is inconsistently much worse in organotypic slice in Fig 4. The effect of wavelength (between 365 and 405nm) should not be such a big factor in penetration or scattering in organotypic slice. Are there reasons for this, are there additional measurement to show this comparison is not biased or screwed?
3. There is no quantification of basic cell properties (e.g., capacitance and membrane resistance) and whether these are changed when expressed in neurons. It will be important to measure a high number of expressing and ideally non-expressing nearby neurons (preferably in acute slices) especially comparing membrane resistance as the author did find dark activation in some opsin. It will be critical to see whether this is the case with the pdCO as well.
4. I have trouble following the expression images in fig 1 and ext fig 3 since the examples are barely visible (especially in Fig 1). Preferably something just showing the soma where the quantification takes place. I also can't see the validity of the quantitative approach as this may be screwed heavily

by the relative expression level of EYFP and mScarlet. It will be nice to show in supplemental figures to see how this quantification method performs with something this is poorly but partly membrane bound such as Chrimson-mScarlet and something very membrane restricted such as mScarlet-CaaX or Chronos-mScarlet as a standard.

5. The experiments in Figure 6 don't have sufficient controls other than no expression? It will still be good to see the comparison to AsOPN3 and FP only (or one of the non-functional opsin) as controls.

6. The pair-pulse recordings in autaptic neurons sometimes show PPF and sometimes show PPD, is this due to the cell variability or the expression of specific construct? If this is cell variability, how is this taken into account in the comparison? It will be good to have no-expression control with sufficient 'n' here.

7. In this manuscript there is limited data on repetitive stimulation, most of the data is based on one 'on' light stimulation and one 'off' light stimulation. How does the rhodopsin do with repetitive on/off cycles? Theoretical switching fatigue of isomerization based tool using dual wavelengths (especially one near UV) can be a concern in the implementation of such tool. It would be nice to see some attempts to test this.

8. Small typos, Fig 1 legend 'longer wavelength (h lambda) is presented with incorrect symbol.

Reviewer #2:

Remarks to the Author:

This study systematically evaluated multiple bistable opsins for optogenetic applications. The authors found that PdCO stands out as an effective and versatile light-activated bistable GPCR. PdCO suppresses synaptic transmission in mammalian neurons independently of GIRK channel activity. This optogenetic GPCR holds promise for achieving efficient presynaptic inhibition with excellent temporal precision. Its spectral characteristics also render it suitable for optical multiplexing. This provides an excellent new tool for precise in vivo manipulation of neural circuits. The experiments are well executed. There are several suggestions to improve the manuscript:

1. In the initial characterization of the optoGPCR, the authors only provided the design in Fig. 1c. However, the main text does not include a comprehensive description of this construct, nor does the figure legend clarify its composition. A few sentences detailing the design of the construct, alongside an explanation of the roles of 1D4 and TS within the DNA construct depicted in Fig. 1c should be included.

2. The main conclusion regarding the optoGPCR emphasizes its impact on excitatory neurotransmission. It is important to consider the feasibility of applying this tool to modulate inhibitory neurotransmission as well.

3. The observed inhibition of EPSC by PdCO through 470nm light in Figure 2f appears to be 10-20%. However, in Fig 2i it is ~50%. Can the authors clear this confusion?

4. With sustained 470nm light stimulation, the PdCO-induced inhibition exhibits rapid recovery without an additional green pulse for deactivation (Fig. 2i). However, in organotypic hippocampal slices, the local application of a brief 500 ms light pulse in the CA1 region reduced evoked PSCs by $71 \pm 0.3\%$ with no spontaneous recovery over 25 minutes (Fig. 4h). This suggests that the recovery time after photostimulation might vary across neuron types or experimental contexts. The authors propose in their discussion that such variations could potentially stem from different recruitment number of

activated G-proteins. It would greatly enhance the strength of their argument if the authors could provide some evidence. Additionally, a brief comment on possible reasons why PSC amplitude is still lower than the original state after the first 525nm illumination (Fig. 4h) could be considered.

5. Potential side effect of PdCO should be discussed, specifically in terms of inducing morphological and functional changes within the infected cells. It is important to address whether this optoGPCRs transfection would result in any discernible behavioral modifications in the animals.

6. In the experiment to identify the peak activation wavelength and the best light pulse duration for optoGPCRs (Fig. 4), the authors used GIRK current as the indicator, which has been shown not to be related to synaptic inhibition via PdCO. Why not using EPSCs as the indicator? A clear explanation would be very helpful.

Author Rebuttal to Initial comments

Reviewers' Comments:

Reviewer #1:

Remarks to the Author:

In this work the authors described a new bistable rhodopsin that seems to have specific selectivity to Go which, as the authors argued, may attribute to the stronger presynaptic inhibition and activation of inhibitory pathways than rhodopsins targeting Gi/o. The main claim to the novelty is the spectrally narrower and red-shifted response and better performance of PdCO versus a previous bistable rhodopsin from Lamprey published by the Bruchas group. The manuscript is generally well-done in the amount of works and characterization and there are no major contradictory results that would make the main finding invalid, however, there are still some minor concerns that should be addressed.

We thank the reviewer for the evaluation of our work, and for the constructive criticism that helped to improve our manuscript. Please find our response to the raised concerns and provided suggestions below.

1. Given the main claim for the manuscript is that PdCO performs better than LcPPO, there is no mention of how experimenter's bias is controlled or not controlled in the experiments comparing the performance of LcPPO and PdCO or even other variants. Are there ways to show the cells chosen for recordings have similar level of expression. How is the comparison controlled for bias?

We agree that an unbiased approach should be used when different tools with a similar function are compared against each other. We carried out blinded experiments and analyses whenever this was possible, as detailed below:

- To facilitate the comparison between opsins, all of our opsin-expressing constructs were designed identically, differing only in the opsin itself.*
- As described in the Methods section and the "nature portfolio reporting summary", we performed the comparison of inhibition efficiency in autaptic neurons in a **blinded fashion** and applied the same illumination protocol to all measured cells, including non-expressing controls (Fig. 2e). We also transduced all autaptic neurons with matched AAV titers, to ensure the same viral delivery of each optoGPCR. While the latter does not guarantee similar levels of expression for each optoGPCR, it does ensure quantification of the inhibition efficiency at a matched AAV titer, which is the factor of interest for optoGPCR applications. Quantification of expression was also performed in a blinded manner, and analysis was carried out using an unbiased automated method.*
- In all other experiments performed to compare optoGPCRs, the same amounts of DNA were used for transfection of optoGPCRs (and EYFP for cell filling) in autaptic neurons (Fig. 1) and single-cell electroporation in organotypic slices (Fig. 4).*
- In the case of viral transduction of organotypic slices, high titers of rAAVs were used to ensure maximal expression of both optoGPCRs. The inhibitory effect of PdCO was stronger than that of LcPPO, even when latter was activated at 10 times higher light intensity. Due to these apparent differences, but also due to different spectral properties of LcPPO and PdCO (e.g. Fig. 4a-c) and the requirement to compare the two optoGPCRs under ideal illumination conditions, blinded experiments were impossible in this case. Nonetheless, cells were always patched randomly without any preselection by fluorescence intensity (now included in the methods, previously missing).*
- In experiments with matched DNA concentrations, we did indeed observe different expression levels and variable membrane localization between the optoGPCRs. Obtaining the same level of expression and targeting for different opsins is practically impossible, since expression levels and membrane targeting efficiency are inherent properties of rhodopsin proteins and are very difficult to alter without substantial modification to the opsin gene, including transmembrane domains, intracellular loops and C-terminal sequences (which can also change the functional properties of the opsin). However, expression levels and*

membrane targeting for LcPPO and PdCO were still comparable (e.g. Fig. 1j,k). We did not directly quantify expression levels of individual recorded cells and therefore cannot correlate GIRK activation or inhibitory effects to expression levels of both optoGPCRs on a cell-by-cell basis.

Taken together, we have done our best to compare optoGPCRs in different preparations and cell types, and to make these comparisons as unbiased as possible. Given these circumstances and the reported results, we are convinced that PdCO outperforms LcPPO in the presented experiments.

2. Despite the LcPPO performs well in microisland recordings at disrupting release, the performance is inconsistently much worse in organotypic slice in Fig 4. The effect of wavelength (between 365 and 405nm) should not be such a big factor in penetration or scattering in organotypic slice. Are there reasons for this, are there additional measurement to show this comparison is not biased or screwed?

Although LcPPO attenuated release in autaptic neurons, the average inhibition was $61 \pm 5\%$, whereas PdCO achieved $89 \pm 3\%$ (Fig. 2e-h). Consistent with this lower inhibition efficiency of LcPPO, the effects of paired-pulse increase as well as reduction of mEPSCs were less pronounced for LcPPO compared to PdCO (EDF 5). In addition, LcPPO showed lower inhibition efficiency in all further autaptic neuron experiments, when activated at maximum responsive wavelengths (Fig. 2h,i, Fig 3i and EDF 6c, EDF 7b).

In neurons of organotypic slice cultures, coupling to GIRK channels was similarly less effective for LcPPO compared to PdCO (Fig. 4a-c). In comparison to autaptic recordings, both LcPPO and PdCO showed reduced inhibition in virally transduced slice cultures (78/71 vs 89 % [PdCO], 27 vs 61 % [LcPPO]) at optimized illumination properties. However, in both preparations, PdCO was consistently more effective compared to LcPPO.

We agree that the choice of optimal wavelengths should not be a crucial factor in slice recordings and are convinced that LcPPO inhibited release less efficiently than PdCO under the tested circumstances. As explained in the previous response, we have done our best to faithfully compare both (and other) optoGPCRs even when using 10 times higher light intensity for LcPPO transduced in organotypic slices and showed that LcPPO performed consistently less well across neuronal preparations and experimental conditions.

3. There is no quantification of basic cell properties (e.g., capacitance and membrane resistance) and whether these are changed when expressed in neurons. It will be important to measure a high number of expressing and ideally non-expressing nearby neurons (preferably in acute slices) especially comparing membrane resistance as the author did find dark activation in some opsin. It will be critical to see whether this is the case with the PdCO as well.

We agree that optoGPCR expression should not alter basic cell properties such as cell capacitance and membrane resistance. As the reviewer correctly notes, some opsins (e.g. OPN511 and Opn7b) showed dark activity, and these opsins were not further tested in neurons. For those opsins that did not show dark activity, we conducted recordings of membrane capacitance and resistance from autaptic neurons as shown below (now added to EDF 5 and respective source data files):

We did not observe any statistically significant deviation from control neurons. We agree that data from acute slice recordings would have been useful as further validation. However, performing these measurements in cultured neurons is common practice in the field.

Accordingly, we added the following sentence to the main text:

"Expression of the different optoGPCRs did not alter the intrinsic properties of expressing neurons (membrane resistance and cell capacitance) when compared to non-expressing control cells (Extended Data Fig. 5)."

4. I have trouble following the expression images in fig 1 and ext fig 3 since the examples are barely visible (especially in Fig 1). Preferably something just showing the soma where the quantification takes place. I also can't see the validity of the quantitative approach as this may be screwed heavily by the relative expression level of EYFP and mScarlet. It will be nice to show in supplemental figures to see how this quantification method performs with something this is poorly but partly membrane bound such as Chrimson-mScarlet and something very membrane restricted such as mScarlet-CaaX or Chronos-mScarlet as a standard.

We agree that the expression images in Fig. 1 were not ideally presented, and expression was barely visible especially for the mScarlet signal. As suggested, we therefore now show 2-times magnified single-channel images, centered on the somata, where the quantification was performed. In addition, we now provide a further magnified 2-channel overlay zoom on the somatic membrane, while the 2-channel overlay on full size images can be found in Extended Data File 3.

We initially assumed that the co-transfection with EYFP would result in equal amounts of detected EYFP, but instead discovered that in case of well-expressing optoGPCRs, EYFP expression was down-regulated, probably because the same promoters were used. This makes a direct normalization to intracellular EYFP impossible. We therefore used cytosolic EYFP only as a region marker for the cytosolic region to measure the mScarlet signal in the cellular membrane only (Fig. 1k). An EYFP-based normalization of optoGPCR expression was only performed by using the sum of both fluorophores (expression index in Fig. 1k). This ensures that variations in total protein expression across different cells have a minimal effect on the output value by internal normalization.

With regard to the suggestion of comparing to other known opsins, we agree that this would be interesting. However, as our measurements are internally-consistent, we feel that this would only serve to validate the initial benchmark. Furthermore, due to the current situation in Israel, we could not complete this experiment in time, and it might take months before we can conduct any additional primary cell culture work.

5. The experiments in Figure 6 don't have sufficient controls other than no expression? It will still be good to see the comparison to AsOPN3 and FP only (or one of the non-functional opsin) as controls.

The rotation behavior experiments were performed with EYFP-expressing mice as a control group. Because this was only mentioned in the figure legend and the main text, we now updated the figure and replaced "Ctrl." by EYFP. For a comparison with AsOPN3, please see Mahn et al. 2021 (<https://doi.org/10.1016/j.neuron.2021.03.013>).

In the experiment targeting the LC→EW projection, we so far only used internal controls where illumination was applied at the basal forebrain (BF). We conducted additional control experiments (mice expressing mCherry) and now directly compare pupil constriction between the two groups at stimulation frequencies between 5 and 40 Hz. While there is no statistically significant difference in pupil constriction of the contralateral eye, high-frequency stimulation (10–40 Hz) leads to a pronounced and significant difference in pupil constriction in PdCO vs. control mice. The data shown below is now included in Fig 6h (and respective source data files).

For further validation of PdCO's utility *in vivo*, we now additionally performed experiments in *C. elegans* that express PdCO under a pan-neuronal promoter. While all-trans retinal fed animals (+ATR) showed strong reduction of locomotion, worms not fed with retinal (-ATR; non-functional opsin) served as control and did not show light-induced reduction of locomotion. A comparison of baseline locomotion showed no difference between +ATR and -ATR animals. The data shown below is now included in Supplemental Figure 2. Further details can be found in the supplementary material.

In addition, we expressed PdCO in cholinergic neurons of *Drosophila* larvae. Activation of PdCO in retinal fed larvae led to a 30 % reduction in aversive stop and turn behavior, which could be reversed by subsequent inactivation of PdCO by green light, while no difference in behavior was observed in larvae that were not supplied with retinal (see below, Supplementary Fig 3). Further details can be found in the supplementary material.

Of note, the baseline behavior differed between *Drosophila* larvae fed with 9-cis retinal and non-retinal controls. We therefore included the following section in our discussion:

"For PdCO expression across various preparations, we did not observe any discernible modifications of intrinsic neuronal cell parameters or effects on baseline behavior compared to vertebrate control cells or animals and *C. elegans*. In *Drosophila* larvae, an increased behavioral response was noted for functionally expressed PdCO compared to control animals. However, it should be noted that high-level overexpression of any exogenous protein can lead to impairment in neuronal cell health. We therefore recommend that users test for such alterations at the cellular, circuit and behavioral level and adhere to the lowest possible expression levels that allow an adequate inhibitory effect of PdCO."

6. The pair-pulse recordings in autaptic neurons sometimes show PPF and sometimes show PPD, is this due to the cell variability or the expression of specific construct? If this is cell variability, how is this taken into account in the comparison? It will be good to have no-expression control with sufficient 'n' here.

As correctly pointed out by Reviewer #1, the autaptic neurons sometimes show PPF or PPD. Because of this cell variability, paired-pulse ratios (PPRs) were normalized to the PPR in the dark of each cell in the previously presented data and only relative changes of PPRs upon light stimulation were shown. For clarification, we now additionally show the absolute PPR values in the dark for each recorded cell and for all constructs studied, including non-expressing control cells that were measured from the same batches of autaptic neurons (experiments done blindly, see below; Extended Data Figure 6 and related source data file). There is no statistically significant change in baseline PPR between cells expressing the optoGPCRs and control cells.

7. In this manuscript there is limited data on repetitive stimulation, most of the data is based on one 'on' light stimulation and one 'off' light stimulation. How does the rhodopsin do with repetitive on/off cycles? Theoretical switching fatigue of isomerization based tool using dual wavelengths (especially one near UV) can be a concern in the implementation of such tool. It would be nice to see some attempts to test this.

We agree that switching fatigue could be a potential issue regarding repetitive activation and deactivation of rhodopsins. We therefore repeated the experiment using our bicistronic ChrimsonR-PdCO construct and performed at least 20 cycles of on- and off-switching. As shown below, we do not observe any switching-fatigue in this experiment. The new data is now available in Figure 5 (and respective source data files), replacing the previously presented data.

In (j), we show the full time course of the experiment. Panel (k) shows higher time resolution traces at repetition cycles 1 (top traces) and 20 (middle traces), as well as the average across all repetitions (bottom traces), without PdCO activation (left, black) and with PdCO activation (right, blue). In panel (l) we show the average PSC traces from the 2 x 10 ChrimsonR activations with and without PdCO activated, respectively, as shown in panel (k). Panel (m) shows the average PSCs for 8 biological replicates, normalized to the average PSC amplitude evoked across the experiment in the "PdCO OFF" state.

8. Small typos, Fig 1 legend 'longer wavelength (h lambda) is presented with incorrect symbol.

Fixed.

Reviewer #2:

Remarks to the Author:

This study systematically evaluated multiple bistable opsins for optogenetic applications. The authors found that PdCO stands out as an effective and versatile light-activated bistable GPCR. PdCO suppresses synaptic transmission in mammalian neurons independently of GIRK channel activity. This optogenetic GPCR holds promise for achieving efficient presynaptic inhibition with excellent temporal precision. Its spectral characteristics also render it suitable for optical multiplexing. This provides an excellent new tool for precise in vivo manipulation of neural circuits. The experiments are well executed. There are several suggestions to improve the manuscript:

We thank the reviewer for the positive assessment of our work. Please find our point-by-point answers regarding the specific comments below.

1. In the initial characterization of the optoGPCR, the authors only provided the design in Fig. 1c. However, the main text does not include a comprehensive description of this construct, nor does the figure legend clarify its composition. A few sentences detailing the design of the construct, alongside an explanation of the roles of 1D4 and TS within the DNA construct depicted in Fig. 1c should be included.

We now added a description of the construct design in the main text which was previously only detailed in the methods section (Molecular biology and DNA constructs):

"We designed optoGPCR constructs as previously reported (Mahn et al. 2021) with a c-terminal rhodopsin 1d4 epitope tag, a Golgi trafficking signal (TS) and an ER export signal (ER) to enhance axonal localization (Mahn et al 2018) (Fig. 1c)."

Furthermore, we have submitted all of the PdCO constructs to Addgene, where readers can find the complete map and sequence for each plasmid.

2. The main conclusion regarding the optoGPCR emphasizes its impact on excitatory neurotransmission. It is important to consider the feasibility of applying this tool to modulate inhibitory neurotransmission as well.

*We demonstrated PdCO activity in excitatory, dopaminergic and noradrenergic neurons, cardiomyocytes and now also pan-neuronal in *C. elegans*, as well as in cholinergic neurons in *Drosophila* larvae. In addition, we now provide a third in-vivo application where PdCO is utilized to silence peptidergic neurotransmission from dynorphin-releasing neurons as shown below (now included in the revised manuscript)*

The following paragraph was added to the manuscript accordingly:

"We next tested how PdCO-mediated inhibition of synapses *in vivo* affects motivated behavior. Prior studies have shown that photostimulation of NAc-VTA D1/dyn terminals with channelrhodopsin negatively impacts feeding behavior, whereas photoinhibition of D1/dyn neurons in the NAc projecting to the VTA enhances it (<https://www.jneurosci.org/content/40/24/4727>). Hence, to demonstrate the utility of PdCO to inhibit peptidergic terminals and impact behavior, we used PdCO to silence dynorphin (dyn) terminals projecting from the nucleus accumbens (NAc) to the ventral tegmental area (VTA) during cued reward consumption behavior. Following injection of PdCO in the NAc and fiber implantation in the VTA in either *pdyn-Cre* or WT mice (Fig. 6i,j), we trained food restricted animals on a cued reward delivery task, where they learned to associate a cue with delivery of sucrose pellets. Once the mice consistently consumed all the reward pellets, mice received 0.25g of sucrose prior to the session to ensure that we are able to bidirectionally modulate behavior, and received stimulation of 465 nm light at varying frequencies (off, 1, 20, 40 Hz, 20ms pulse-width) time-locked to cue presentation for 10 s across counterbalanced sessions (Fig. 6k,l). Our results show that *pdyn-Cre* animals that received 20 or 40 Hz light pulses to the VTA increased their food consumption, relative to the sessions where no light was delivered (Fig. 6m,n). In contrast, in WT controls, light delivery at any frequency did not alter consumption (Fig. 6o,p)."

Our previously published findings showed that eOPN3 can be used to silence inhibitory neurotransmission (Mahn et al. 2021). Similarly, chemogenetic inhibitory GPCRs can attenuate inhibitory neurotransmission as demonstrated in multiple papers (e.g.: <https://doi.org/10.7554/eLife.68760>, <https://doi.org/10.1038/s41386-023-01620-5>, <https://doi.org/10.1093/cercor/bhac245> or <https://doi.org/10.1038/nature12485>). While this is potentially feasible for PdCO as well, it was technically impossible for us to perform this additional experiment at this time. We therefore refined our summary in the discussion to include a reference to "excitatory and neuromodulatory" neurotransmission:

"Taken together, our results demonstrate that PdCO is a rapid, reversible, and versatile optoGPCR that mediates efficient silencing of glutamatergic and neuromodulatory synaptic transmission in diverse cell types *in vitro* and *in vivo*."

3. The observed inhibition of EPSC by PdCO through 470nm light in Figure 2f appears to be 10-20%. However, in Fig 2i it is ~50%. Can the authors clear this confusion

Thank you for pointing this out. The apparent discrepancy results from the fact that the average EPSC traces in Fig. 2f are averaged from 7 consecutive EPSCs recorded over a 35-s period before (gray) and after (blue) a 500ms pulse. At 470nm, PdCO recovers to its baseline state spontaneously (as can be seen from the time course in Fig. 2i), the average EPSC seen in Fig. 2f shows a lower attenuation than the peak shown in Fig. 2i. To avoid confusion, we added the following sentence in the Results section describing this experiment: "...we varied the wavelength of the activating 500 ms light pulse to generate action spectra for opsin activation, quantified from the average EPSC inhibition over 35 s post-illumination (Fig. 2f)."

4. With sustained 470nm light stimulation, the PdCO-induced inhibition exhibits rapid recovery without an additional green pulse for deactivation (Fig. 2i). However, in organotypic hippocampal slices, the local application of a brief 500 ms light pulse in the CA1 region reduced evoked PSCs by $71 \pm 0.3\%$ with no spontaneous recovery over 25 minutes (Fig. 4h). This suggests that the recovery time after photostimulation might vary across neuron types or experimental

contexts. The authors propose in their discussion that such variations could potentially stem from different recruitment number of activated G-proteins. It would greatly enhance the strength of their argument if the authors could provide some evidence. Additionally, a brief comment on possible reasons why PSC amplitude is still lower than the original state after the first 525nm illumination (Fig. 4h) could be considered.

The organotypic slice experiments were carried out with a 405nm light pulse, not 470nm. Therefore no recovery of PdCO is expected as no recovery is induced by 405nm light (see also Fig. 5 a,b). However, this comment drew our attention to the fact that our color scheme used for representing activation wavelengths was not consistent across figures. We therefore changed all illumination colors to:

*Black / gray: <385nm
Purple: 385-405nm
Blue: 440-470nm*

The recruitment of a different number of G-proteins could presumably be inferred from the presented spectral data discussed above (Fig. 2e, pulsed activation), where only transient inhibition is visible for 470 nm activation. To test this hypothesis, the number of activated G-proteins needs to be determined.. This seems to be a very complex task that is not feasible to us in combination with electrophysiological experiments.

The lower EPSC amplitude after 525 nm illumination is most likely explained by a slight rundown over the long recording period. This could result from incomplete neurotransmitter recycling or from changes of the access resistance over time. Notably, the reduction in EPSC amplitude in this experiment was not statistically significant.

Figure description: The paired Hedges' g between normalized EPSC amplitudes pre-405 nm and post-525nm is shown in the above Gardner-Altman estimation plot. Both groups are plotted on the left axis as a slopegraph: each paired set of observations is connected by a line. The paired mean difference is plotted on a floating axis on the right as a bootstrap sampling distribution. The mean difference is depicted as a dot; the 95% confidence interval is indicated by the ends of the vertical error bar. The paired Hedges' g between pre 405 nm and post 525 nm is -1.04 [95.0%CI -2.45, 0.262]. The P value of the two-sided permutation t-test is 0.114.

5. Potential side effect of PdCO should be discussed, specifically in terms of inducing morphological and functional changes within the infected cells. It is important to address whether this optoGPCRs transfection would result in any discernible behavioral modifications in the animals.

We agree that the use of any exogenously expressed protein (utilized for optogenetic purposes) should have minimal to no side effects. As mentioned in our answers to Reviewer #1, questions 3,5,6, we did not observe any change in

intrinsic cell parameters like membrane resistance or cell capacitance for PdCO expressing cells compared to control neurons, nor did we observe a change in baseline paired-pulse ratios between PdCO and non-expressing cells. Furthermore, we did not observe any change in baseline behavior between PdCO-expressing mice and mice expressing a control fluorophore, as well as in the newly added C. elegans dataset (Supplementary Material). As mentioned above (Reviewer 1, comment 5), we observed a baseline discrepancy in the newly added Drosophila larvae experiments (Supplementary Material).

We therefore conclude that PdCO expression across different preparations and species, except Drosophila larvae does not result in any discernible modifications. It is important to note, however, that any optogenetic actuator can cause changes in cell health when strongly overexpressed, which calls for rigorous testing of new AAVs and expression vectors when initiating a new optogenetic experiment. We have added a paragraph regarding this point to our Discussion section as follows:

"For PdCO expression across various preparations, we did not observe any discernible modifications of intrinsic neuronal cell parameters or effects on baseline behavior compared to vertebrate control cells or animals and C. elegans. In Drosophila larvae, an increased behavioral response was noted for functionally expressed PdCO compared to control animals. However, it should be noted that high-level overexpression of any exogenous protein can lead to impairment in neuronal cell health. We therefore recommend that users test for such alterations at the cellular, circuit and behavioral level and adhere to the lowest possible expression levels that allow an adequate inhibitory effect of PdCO."

6. In the experiment to identify the peak activation wavelength and the best light pulse duration for optoGPCRs (Fig. 4), the authors used GIRK current as the indicator, which has been shown not to be related to synaptic inhibition via PdCO. Why not using EPSCs as the indicator? A clear explanation would be very helpful.

The experiments in Fig. 4a-c, using GIRK currents as the indicator were used as an additional measure to prove similar behavior of PdCO and LcPPO upon light stimulation in organotypic slices. These results confirm the detailed spectral characterization based on EPSCs measured in autaptic neurons (Fig. 2f,g and Fig. 5a,b).

Decision Letter, first revision:

Dear Ofer,

Thank you for submitting your revised manuscript "A bistable inhibitory OptoGPCR for multiplexed optogenetic control of neural circuits" (NMETH-A53193A). It has now been seen by the original referees and their comments are below. The reviewers find that the paper has improved in revision, and therefore we'll be happy in principle to publish it in Nature Methods, pending minor revisions to satisfy the referees' final requests and to comply with our editorial and formatting guidelines.

TRANSPARENT PEER REVIEW

Please note: we allow redactions to authors' rebuttal and reviewer comments in the interest of confidentiality. If you are concerned about the release of confidential data, please let us know specifically what information you would like to have removed. Please note that we cannot incorporate redactions for any other reasons. Reviewer names will be published in the peer review files if the reviewer signed the comments to authors, or if reviewers explicitly agree to release their name. For more information, please refer to our FAQ page.

ORCID

Best regards,
Nina

Nina Vogt, PhD
Senior Editor
Nature Methods

Reviewer #1 (Remarks to the Author):

The revised manuscript is much improved and most of points are adequately addressed.

Reviewer #2 (Remarks to the Author):

The authors have addressed my suggestions and the paper is improved.

Final Decision Letter:

Dear Ofer,

I am pleased to inform you that your Article, "A bistable inhibitory OptoGPCR for multiplexed optogenetic control of neural circuits", has now been accepted for publication in Nature Methods. The received and accepted dates will be July 17th, 2023 and April 18th, 2024. This note is intended to let you know what to expect from us over the next month or so, and to let you know where to address any further questions.

Over the next few weeks, your paper will be copyedited to ensure that it conforms to Nature Methods style. Once your paper is typeset, you will receive an email with a link to choose the appropriate publishing options for your paper and our Author Services team will be in touch regarding any additional information that may be required. It is extremely important that you let us know now whether you will be difficult to contact over the next month. If this is the case, we ask that you send us the contact information (email, phone and fax) of someone who will be able to check the proofs and deal with any last-minute problems.

Please note that *Nature Methods* is a Transformative Journal (TJ). Authors may publish their research with us through the traditional subscription access route or make their paper immediately open access through payment of an article-processing charge (APC). Authors will not be required to make a final decision about access to their article until it has been accepted. Find out more about Transformative Journals

Authors may need to take specific actions to achieve compliance with funder and institutional open access mandates. If your research is supported by a funder that requires immediate open access (e.g. according to Plan S principles) then you should select the gold OA route, and we will direct you to the compliant route where possible. For authors selecting the subscription publication route, the journal's standard licensing terms will need to be accepted, including self-

archiving policies. Those licensing terms will supersede any other terms that the author or any third party may assert apply to any version of the manuscript.

Best regards,
Nina

Nina Vogt, PhD
Senior Editor
Nature Methods